# Pre-marking chromatin with H3K4 methylation is required for accurate zygotic genome activation and development

Meghana S. Oak [1,10] ✉, Marco Stock[1,2,3,4,10], Ana Janeva[1], Matthias Mezes[1], Antony M. Hynes-Allen[5], Tobias Straub [6], Ignasi Forné[7], Andreas Ettinger [1], Stephan Hamperl [1], Axel Imhof[7,8], Jelle van den Ameele [5], Antonio Scialdone[1,2,3] & Eva Hörmanseder [1,9] ✉

In vertebrate embryos, gene expression is first initiated at zygotic genome activation (ZGA). Maternally expressed transcription factors are essential for this process. However, it is unknown whether active chromatin modifications established in gametes are present in early embryos and contribute to ZGA and embryonic development. Here, we show that in *Xenopus laevis*, H3K4me3 occurs at common genomic loci in gametes, in transcriptionally quiescent pre-ZGA embryos, and in transcriptionally active ZGA embryos. These loci exhibit high H3K4me3 intensities and breadth, DNA hypomethylation, and elevated CpG content. We show that H3K4 methylation pre-marking is required for successful ZGA and development, including expression of the key ZGA transcription factor Pou5f3.2. We demonstrate that the H3K4 methyltransferase Cxxc1 ensures establishment of H3K4me3 and proper ZGA. These findings reveal a role for H3K4 methylation in defining active chromatin states in *Xenopus laevis* embryos and highlight its importance for accurate ZGA and embryonic development.

Cell fates are established through the activity of developmental cues, transcription factors (TFs) and epigenetic mechanisms that either permit or repress transcriptional programs. In vertebrates, the development of zygotes is supported by maternal transcripts and TFs until zygotic genome activation (ZGA), when gene expression patterns are established. The extent to which this depends on TFs or epigenetic mechanisms is not fully understood. It is unclear how parental chromatin modifications contribute to embryonic development via epigenetic memory.

Epigenetic memory refers to the maintenance of chromatin states across cell divisions independently of the developmental signal that induced the state[1]. Most studies of epigenetic inheritance have confirmed the transmission of repressive histone modifications around transcriptionally inactive chromatin. However, the role of active histone modifications, traditionally associated with actively transcribed genes, in epigenetic memory is under debate[2]. The maintenance of active chromatin states is primarily dependent on TF-coupled gene

[1]Institute of Epigenetics and Stem Cells (IES), Helmholtz Zentrum München, München, Germany. [2]Institute of Functional Epigenetics, Helmholtz Zentrum München, German Research Center for Environmental Health, Neuherberg, Germany. [3]Institute of Computational Biology, Helmholtz Zentrum München, German Research Center for Environmental Health, Neuherberg, Germany. [4]TUM School of Life Sciences Weihenstephan, Technical University of Munich, Munich, Germany. [5]Department of Clinical Neurosciences and MRC Mitochondrial Biology Unit, University of Cambridge, Cambridge, UK. [6]Core Facility Bioinformatics, Biomedical Center Munich (BMC), Faculty of Medicine, Ludwig-Maximilians-Universität in Munich, Martinsried, Germany. [7]Protein Analysis Unit, Biomedical Center (BMC), Faculty of Medicine, Ludwig-Maximilians-University (LMU) Munich, Martinsried, Germany. [8]Department of Molecular Biology, Biomedical Center Munich, Ludwig-Maximilians University, Planegg-Martinsried, Germany. [9]Research Institute of Molecular Pathology (IMP), Vienna Bio-Center, Vienna, Austria. [10]These authors contributed equally: Meghana S. Oak, Marco Stock. ✉e-mail: meghana.oak@helmholtz-munich.de; eva.hoermanseder@imp.ac.at

expression[3]. Propagation of active histone marks is thought to be dependent on TF binding and continuous gene transcription, and not via direct transmission using autonomously operating read-and-write mechanisms. Therefore, the field does not unequivocally consider active histone marks as an epigenetic mechanism. Moreover, the tight link between active chromatin states and transcription complicates efforts to discern whether active histone marks are causal or consequential, and whether they reflect epigenetic mechanisms[4,5].

Embryos of species such as *Xenopus laevis* and zebrafish undergo a transcription-free window of at least 8 rapid cell divisions from fertilization to ZGA, challenging chromatin-modification propagation[6]. In zebrafish, the active chromatin mark H3K4me3 is present before ZGA at developmental gene promoters, and placeholder mechanisms driven by other chromatin modifications, H2A.Z and DNAme, were shown to set the stage for the first transcriptional program[7,8]. In *Xenopus*, technical limitations hamper histone mark analyses of this dynamic, early embryonic chromatin, and thus, the presence of H3K4me3 is questioned[9]. Instead, it has been suggested that H3K4me3 is rapidly acquired de novo later around ZGA[9–11]. This would exclude a role of H3K4me3 in transmitting active chromatin states from gametes to embryos, as well as overlook the contribution of such a mechanism to embryonic development.

Previous experiments revealed that H3K4me3 decorates specific genomic regions in *Xenopus* sperm nuclei and in blastula embryos prior to ZGA[12]. Thus, chromatin marks could be present on the dynamic chromatin of pre-ZGA embryos. Cell-fate reprogramming studies using nuclear transfer to *Xenopus* eggs suggest that H3K4me3 may play a role in maintaining active transcriptional states[13,14]. Demethylation of H3K4 in somatic donor nuclei rendered them more permissive to reprogramming, and transcriptional memory of active gene expression states was lost[13]. These experiments indicate that nuclear transfer embryos retain information encoded by H3K4me3 via epigenetic memory. It raises the question of whether this mechanism also propagates information from gametes to embryos during physiological development. Together, these observations support the involvement of H3K4me3 in maintaining active transcriptional states between fertilization and ZGA, but evidence is currently missing. It has not been tested whether maintenance of active chromatin states marked by H3K4 methylation is important for embryonic development.

Here, we made use of the transcriptionally silent pre-ZGA developmental period in *Xenopus laevis* embryos[15–17] to investigate the dynamics of the active histone modification H3K4me3 and its role in embryonic development. We profiled histone modifications on chromatin in pre-ZGA embryos using mass spectrometry. We detected H3K4me3 together with other histone modifications associated with active chromatin on pre-ZGA chromatin, despite multiple rapid cell divisions in the absence of transcription. We traced the H3K4me3 mark on a large group of genes that are transcribed in the parental gametes, as well as the ZGA embryo, separated by transcriptional dormancy in cleavage-stage embryos. We show that H3K4me3 peaks are present around these genes at these selected time points prior to and during ZGA stages of development. Promoters of these genes are furthermore characterized by high CpG density and DNA accessibility, as well as DNA hypomethylation. Knockdown of the transcription-independent H3K4 methyltransferase Cxxc1 results in a reduction of H3K4me3 levels, improper gene expression of zygotic genes, and embryonic defects. H3K4 demethylation specifically during pre-ZGA cleavages shows defects of ZGA as well as increased embryonic defects. This suggests that H3K4me3 marking of chromatin during early embryonic cell cycles is essential for subsequent faithful ZGA. Together, this study demonstrates that H3K4 methylation is required for regulating embryonic transcription and development. The presence of H3K4me3 at ommon loci in gametes, prior to ZGA and during ZGA, is consistent with a potential role in epigenetic memory, suggesting that this modification contributes to information about chromatin states during early embryogenesis.

## Results

### Histone modifications linked with active chromatin are present in early embryos prior to ZGA

After fertilization, the *Xenopus* zygote divides rapidly 8 times before gene transcription is fully activated at ZGA. During these cleavages, the genome remains globally transcriptionally silent due to histone-mediated repression and inhibitory chromatin hindering TF binding until mid-blastula transition[15–17]. We first tested whether chromatin marks normally associated with transcriptionally active chromatin states are present on this fast-replicating chromatin in the absence of gene transcription.

We established a method enriching chromatin proteins from yolk-rich embryos for chromatin profiling experiments. Using mass spectrometry analysis of histone tail post-translational modifications (see "methods", Fig.1a and Fig. S1a), we detected a wide range of active and repressive histone modifications on pre-ZGA embryo chromatin (256-cell stage). These included the active marks H3K4me1/2/3, H3K9ac and H3K27ac, but not H3K36me3, as well as the repressive marks H3K9me3 and H3K27me3 and the replication-associated histone modification H4K20me1 (Fig.1b). As expected, we also observed these histone marks on transcriptionally active mid-ZGA embryo chromatin (4000-cell stage; Fig.1b). Together, this reveals that highly dynamic chromatin of rapidly dividing pre-ZGA embryos is decorated by both repressive and active chromatin marks.

Western Blotting confirmed these results. We detected H3K4me3-modified chromatin in pre-ZGA embryos from as early as the 32-cell stage in development (Fig. 1c), and previous studies detected H3K4me3 in whole protein extracts from *Xenopus laevis* 1-cell zygotes and 4-cell embryos, at levels comparable to those observed in 256-cell embryos[18]. H3K4me3 levels relative to total H3 on chromatin remained constant between 32-cell and 256-cell stages, followed by a pronounced increase by mid-ZGA, consistent with our mass spectrometry data (Fig. S1b). Despite global transcriptional quiescence in embryos before ZGA, the active histone marks H3K4me3, H3K4me1, H3K9ac and H3K27ac were detectable in 256-cell stage embryos, as well as in post-ZGA stage embryos (Fig. 1d). Conversely, we did not detect the transcription-associated H3K36me3 and H3K79me3 modifications in pre-ZGA chromatin. These results support that active histone marks H3K4me1/3, H3K9ac and H3K27ac are present on chromatin prior to ZGA.

In *Xenopus*, as well as in *Drosophila*, zebrafish and mouse, a defined cluster of microRNAs and β-catenin target genes is transcribed before the major wave of ZGA[19–25]. We hypothesized that this early transcription may reinforce H3K4 promoter methylation through transcription-coupled mechanisms[4,26] mediated by RNA Polymerase II and III, as RNA Polymerase I is absent in pre-ZGA embryos[15,27]. We tested whether transcription inhibition via the RNA Polymerase II and III inhibitor α-amanitin results in acute depletion of H3K4me3 in embryos during ZGA (Fig. S1c). At such high doses, α-amanitin is shown to not only inhibit elongation, but also destabilize the RPB1 subunit, causing rapid RNA Polymerase II loss[28,29]. We co-injected the uridine analog 5-EU, which is incorporated into nascent RNAs[30] to reveal effective inhibition of transcription (Fig. S1d). We observed that both H3K4me3 and H4 are detected in transcription-inhibited, 5-EU negative embryos and transcriptionally active, 5-EU positive embryos at developmental stages corresponding to mid-ZGA (Fig. 1e), in line with previous reports[11]. We noted that H3K4me3 levels are only reduced by 26% in α-amanitin-injected embryos (Fig. 1e), indicating that while a subset of H3K4me3 is transcription-dependent, the majority of H3K4me3 marks on mid-ZGA embryo chromatin is independent of RNA Pol II and III-mediated transcription.

This demonstrates that the dynamic chromatin of rapidly dividing, transcription-inhibited *Xenopus laevis* embryos is decorated by H3K4me3, together with several other histone modifications. It also indicates that during developmental stages preceding ZGA, the presence of active chromatin modifications may be supported by mechanisms operating independently of feedback from ongoing gene transcription.

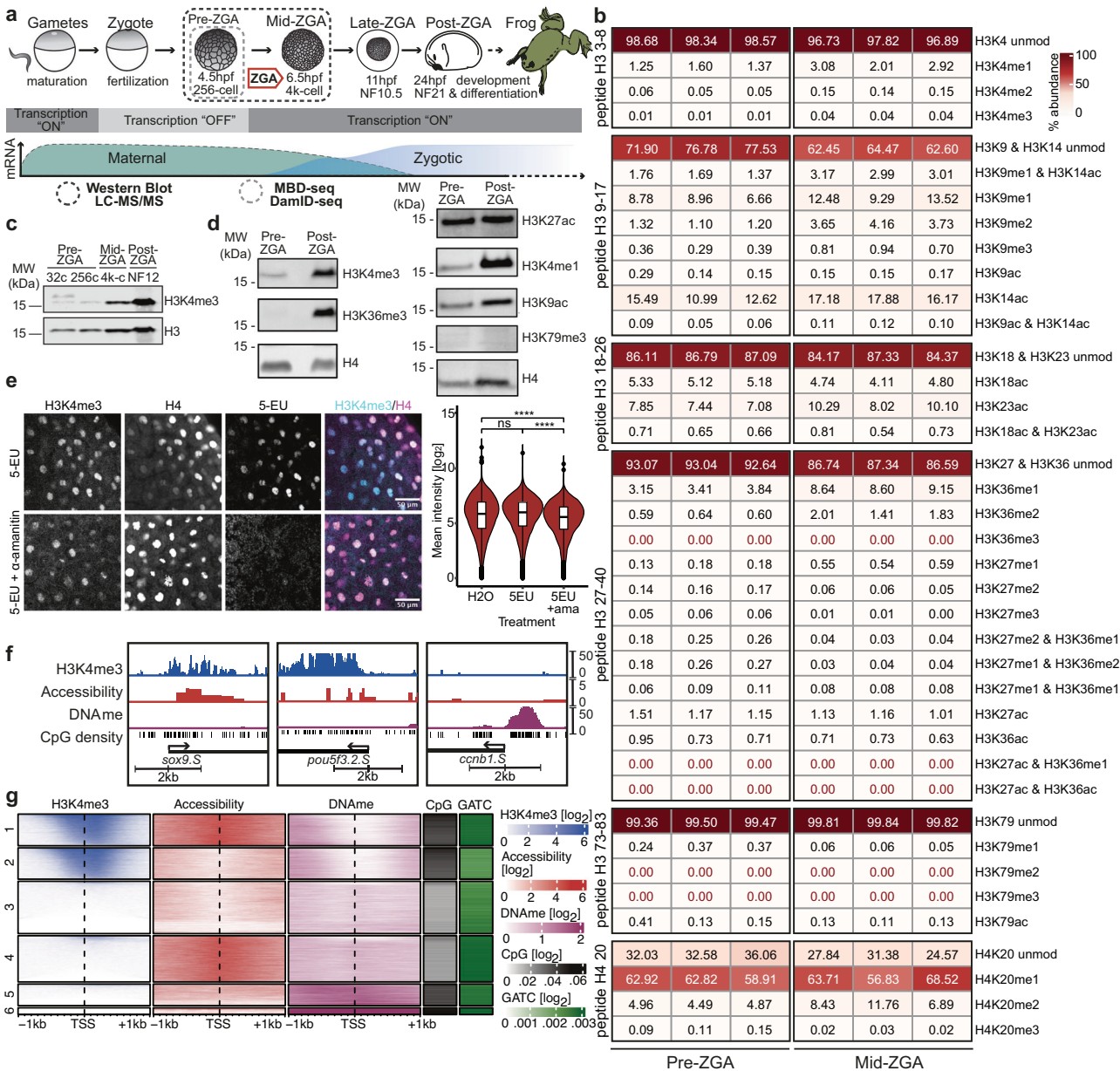

**Fig. 1 | Profiling histone modifications and epigenetic factors present on pre-ZGA embryo chromatin. a** Schematic highlighting early developmental stages of interest in this study and their transcriptional states with a focus on ZGA and the methods used in this figure. **b** Heatmap representing the percent abundance of histone modifications on each respective peptide relative to the total abundance of the corresponding peptide, obtained by PTM analysis of LC-MS/MS. Undetectable modifications are highlighted in red text. Each column represents a single biological replicate of pre-ZGA or mid-ZGA embryos. **c** Time-series Western blot showing the occurrence of H3K4me3 and H3 in embryos. **d** Western blots of histone modifications in pre-ZGA and post-ZGA embryos using chromatin-bound protein isolation. Western blots were independently repeated 2–3 times with consistent results. **e** Representative images of H3K4me3 levels in control and transcription-inhibited mid-ZGA embryos. Nascent transcripts are labeled using 5-EU. Overlay of H3K4me3 (cyan) and H4 (magenta) is represented. Scale bars: 50 μm. Nuclei are segmented using H4, and

H3K4me3 values are measured. (Right) Quantification of H3K4me3 immunofluorescence in H4-segmented nuclei across Z-stacks ($N = 3$ embryos per condition, $n = 2$ independent experiments). Box plots show the median (center line), 25–75th percentiles (box), whiskers extending to 1.5× the interquartile range, and outliers as individual points. Statistical test: one-sided Wilcoxon rank-sum test with the alternative hypothesis that the values from the amanitin-injected condition are lower than controls; $p$: (****) ≤ 0.0001,; n.s. are $p > 0.05$. Exact $p$: $H_2O$ vs 5EU: 0.2; $H_2O$ vs 5EU +Ama: $2.2 \times 10^{-16}$; 5EU+Ama vs Ama: $2.2 \times 10^{-16}$. **f** IGV tracks displaying H3K4me3 ChIP-seq, CATaDa, and MBD-seq data for pre-ZGA embryos with CpG regions at representative promoters (TSS ± 1kb). **g** Heatmap showing H3K4me3, accessibility, DNAme, CpG and GATC density across promoters (TSS ± 1kb) in pre-ZGA embryos. Accessibility enrichment is normalized to GATC frequency within the promoter region. Clusters are identified by k-means clustering on the z-scored data, excluding GATC density ($k = 6$; Hartigan-Wong algorithm).

## H3K4me3-marked promoters in pre-ZGA embryos are in distinct chromatin configurations

After establishing that pre-ZGA chromatin is marked with active histone modifications, we addressed the genomic context of H3K4me3. In transcriptionally active cells, H3K4me3 is enriched at promoters of expressed genes and associated with DNA hypomethylation, high CpG

density and accessible chromatin[31–35]. We tested whether these associations also apply to *Xenopus laevis* embryos prior to ZGA.

We profiled H3K4me3, DNA methylation (DNAme) and chromatin accessibility in pre-ZGA embryos using ChIP-seq, MBD-seq and CATaDa (Chromatin Accessibility profiling using Targeted Dam-ID), respectively (Fig.1a, f)[36,37] and calculated their occurrence at promoters

(1 kb ± TSS), normalizing accessibility to local GATC motif density (Fig. S1e). In pre-ZGA embryos, DNAme was depleted at ~95% of H3K4me3-marked promoters. Similarly, a prior study in *Xenopus tropicalis* similarly found that at mid-ZGA, ~90% H3K4me3-marked promoters were depleted of DNAme[38] (Fig. S1e). Promoters with high H3K4me3 levels and DNA hypomethylation also showed high CpG densities (Fig.1g clusters 1 and 2). Furthermore, approximately a third of the H3K4me3-marked promoters were in an accessible configuration, DNA hypomethylated and CpG-rich (Fig. 1g cluster 1), resembling promoters of actively transcribed genes. The remaining H3K4me3-marked promoters were less accessible, but DNA hypomethylated and CpG-rich (Fig. 1g cluster 2). Promoters lacking H3K4me3 were in three different configurations: DNA methylated, CpG-rich with low accessibility, suggesting a repressive chromatin state (Fig.1g clusters 5 and 6), DNA hypomethylated, CpG-poor and relatively accessible (Fig.1g cluster 4), or DNA hypomethylated, CpG-poor and less accessible (Fig.1g cluster 3).

This shows that in pre-ZGA *Xenopus laevis* embryos, characterized by widespread transcriptional quiescence, H3K4me3-marked promoters exhibit distinct chromatin configurations considering CpG content, DNAme and chromatin accessibility.

## Genes with high H3K4me3 intensity and breadth are shared across gametes and early embryos

We then investigated the origins and fate of promoter-associated H3K4me3 identified in pre-ZGA embryos by comparing its genomic landscape from the last transcriptionally active stage in gametes to the next transcriptionally active stage at ZGA (Fig.2a). For this, we utilized published H3K4me3 ChIP-seq datasets for spermatid, sperm, pre- and post-ZGA embryonic stages[12,39,40].

We found that H3K4me3 peaks predominantly associated with promoters in the gametes and early embryos (Fig. S2a). Of all identified H3K4me3 promoter peaks, 75% were shared across all four time points and made up 39% of all promoters ("SHARED" group; Fig. 2b, c). This suggests that H3K4me3 is established during the spermatid stage and subsequently present at the promoters of these genes until ZGA. In contrast, for 48% of all promoters, H3K4me3 peaks were not detected at any of the four time points ("ABSENT" group, Fig. 2c, d). Only the remaining 13% of promoters displayed dynamic H3K4me3 peak changes across stages (Figs.2b and S2b) and were subclassified by pattern (Fig. S2c). We refer to genes with a promoter H3K4me3 peak in spermatids that is absent at any subsequent stage as the "LOST" group (Figs. 2b and S2d), and those acquiring a peak after the spermatid stage as the "GAINED" group (Figs. 2b and S2d). These results suggest that most promoters marked by H3K4me3 in pre-ZGA embryos are already marked in the male gametes and also in ZGA stage embryos, although whether this reflects a continuous transmission remains unclear.

Next, we characterized the genes within these H3K4me3 dynamics groups. Gene ontology (GO) analyses revealed an enrichment of terms associated with housekeeping functions in the SHARED group, whereas terms related to embryo development, gastrulation and pattern specification were enriched in the GAINED group, and to a lesser extent in the SHARED group (Fig. S2e). No significant enrichment for GO-terms was found in the LOST group. Thus, in embryos, promoter H3K4me3 present before ZGA may support housekeeping functions, whereas acquiring H3K4me3 at ZGA may support developmental and differentiation.

We then evaluated H3K4me3 peak characteristics of the H3K4me3 dynamics groups. We found that the SHARED group showed significantly higher H3K4me3 peak intensities and breadths around the promoter regions than all other groups across all analyzed time points (Figs. 2e and S2f). Relative H3K4me3 promoter enrichment and breadth in the spermatid indicated how long H3K4me3 remains detectable during subsequent developmental stages: promoters of the SHARED group had the highest H3K4me3 enrichment and breadth

scores in spermatid (Fig. S2g), followed by promoters of the LOST subgroup with progressively lower scores. At ZGA, promoters that gained H3K4me3 peaks at pre-ZGA show greater H3K4me3 enrichment and breadth compared to peaks gained later at the ZGA stage (Fig. S2h). Consistent with reports of H3K4me3 extending into the coding region of actively transcribed genes in somatic cells[41], we observed an asymmetrical extension of the peak into the gene body for GAINED genes during ZGA onset (Fig. S2f). These analyses revealed that H3K4me3 peak intensities and breadth closely correlate with how long H3K4me3 marking is detected in early development.

Given the connections between CpG content, DNA hypomethylation and H3K4me3 deposition in embryonic stem cells (ESCs)[42], we explored how these features relate to H3K4me3 dynamics in *Xenopus*. We found that promoters of the SHARED group had significantly increased DNA accessibility (Fig. 2f), lower DNAme (Fig. 2h), and higher CpG content (Fig. 2h) than all other groups in pre-ZGA embryos. Interestingly, promoters gaining H3K4me3 during ZGA had lower chromatin accessibility (Fig. 2f), CpG content (Fig. 2h), and higher DNAme levels (Fig. 2g) than SHARED gene promoters in pre-ZGA embryos. This indicates that H3K4me3 presence around CpG-rich and accessible promoters during early development anticorrelates with DNAme before ZGA (Fig. 2i).

Together, we observe that H3K4me3 domains around promoters displaying high intensities and breadth are present at the same genomic loci in the last transcriptionally active stages in gametes, across transcriptionally dormant pre-ZGA stages, and in the next transcriptionally active ZGA stage in embryos.

## H3K4me3 marks genes transcribed in gametes and at ZGA during the interjacent silent window

We next asked whether H3K4me3 promoter enrichment across a window of transcriptional silence contributes to inheriting memory of past active transcriptional states. We first hypothesized that promoters marked by H3K4me3 in gametes and embryos were once actively transcribed in the gametes and then again during ZGA.

We used datasets of nascent RNA-seq spanning pre- and mid-ZGA stages (5–9 hpf), and total RNA-seq datasets from egg and spermatid, to characterize gene expression groups based on their transcriptional dynamics[39,40,43]. Genes whose transcripts were detected in either maternal or paternal gametes, and in embryos during these ZGA time points, were defined as "Gamete+Zygotic" genes (GZ-genes, Fig. 3a, b, c). Additionally, we grouped genes exclusively expressed in the maternal and paternal gametes ("Gamete-Specific", GS-genes), and genes expressed during ZGA and not detected in either gamete ("Zygote-Specific", ZS-genes; Fig. 3a, b, c. Genes with undetectable transcripts at all specified time points were named "Not-Detected" (ND) genes. GO analyses revealed that "embryo development" was associated with both GZ- and ZS-genes (Fig. S3a). In agreement with our analysis of H3K4me3 dynamics, terms related to housekeeping functions were primarily linked to GZ-genes, while terms associated with gastrulation, morphogenesis and symmetry formation were specifically enriched in ZS-genes (Fig. S3b). Together, this suggests that GZ-genes may propagate their active transcriptional state from the gametes to embryos and have potential housekeeping or developmental functions.

To address whether GZ gene promoters are marked by H3K4me3 across developmental stages, we compared H3K4me3 peak characteristics of GS-, ZS- and ND- genes at four time points—spermatid, sperm, pre-ZGA and ZGA (Fig. 3d). We observed that GZ genes had significantly higher H3K4me3 peak intensities and breadth than all other gene groups at all time points, notably also during the transcriptionally quiescent stages (Figs. 3d and S3c). GS-genes initially showed lower mean peak intensities than GZ-genes, dropping below those of ZS-genes during ZGA (Fig. 3d). Genes expressed only in male gamete progenitors (Spermatid-Specific or SS) displayed even lower average H3K4me3 levels than the overarching GS group (Fig. S3d).

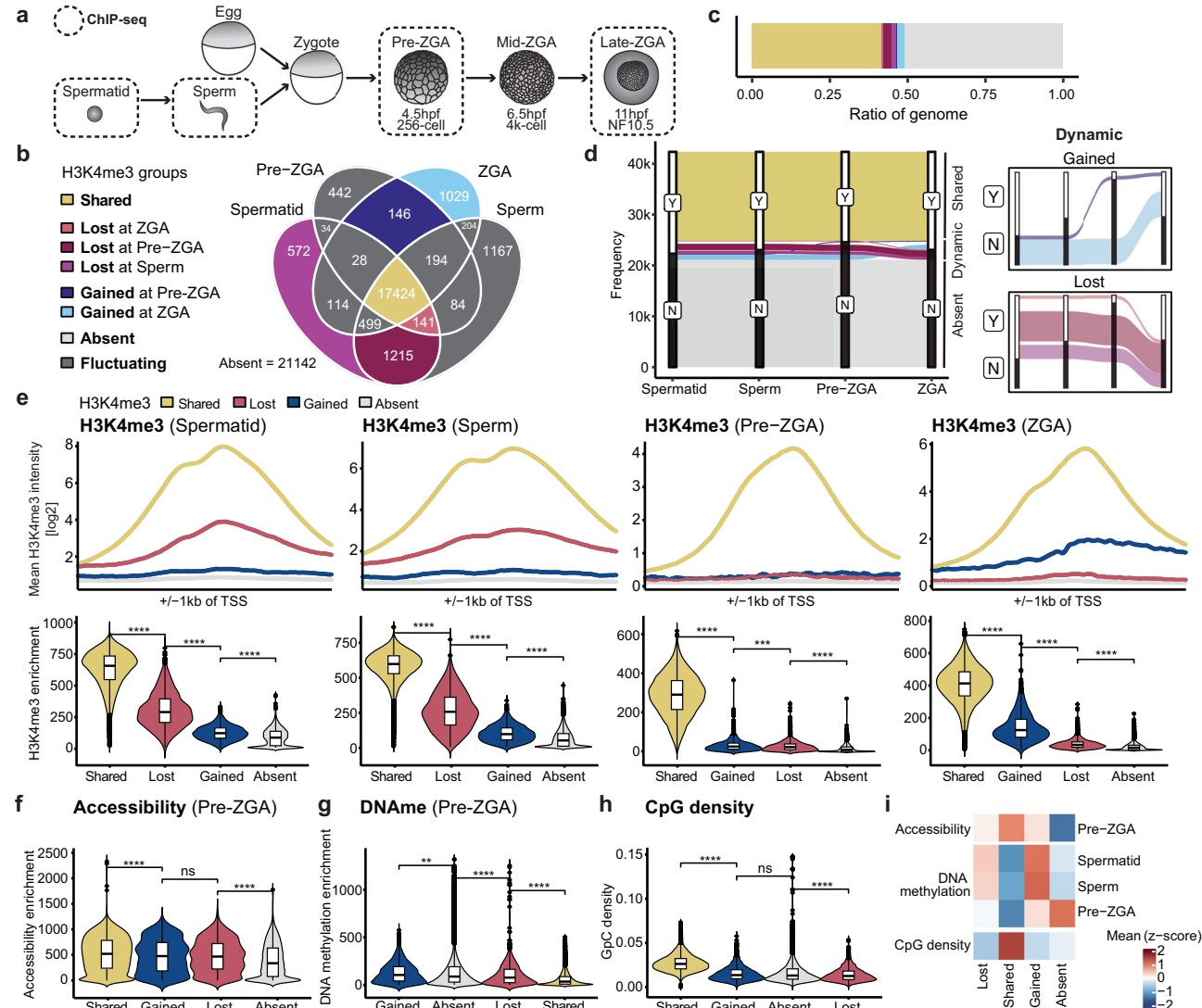

**Fig. 2 | H3K4me3 peak intensity closely correlates with the dynamics of H3K4me3 localization. a** Illustration highlighting stages of interest for H3K4me3 ChIP-seq. **b** Venn diagram showing the binarized overlap of H3K4me3 peaks at all promoter regions (TSS ± 1kb) across spermatid, sperm, pre-ZGA and post-ZGA timepoints. Colors are based on the dynamics of H3K4me3 across the stages: Yellow: genes that "share" H3K4me3 ("SHARED", $n = 17424$), blue: genes that "gain" H3K4me3 ("GAINED", $n = 1175$), purple: genes that "lose" H3K4me3 peaks across timepoints ("LOST", $n = 1928$). Promoters with fluctuating H3K4me3 dynamics are excluded from further analysis and marked in dark gray. **c** Ratio of genes that fall into each group of H3K4me3 dynamics. **d** Alluvial plots representing H3K4me3 dynamics of each group. **e** (Top) H3K4me3 around TSS (± 1kb) for H3K4me3 dynamics groups at each time point-spermatid, sperm, pre-ZGA, post-ZGA,

respectively. (Bottom) H3K4me3 promoter enrichment in each group for each time point. Statistical test: one-sided Wilcoxon rank-sum test with alternative hypothesis that values of left distribution tend to be larger than right distribution; $p$: (****)≤ 0.0001, (***)≤0.001, (**) ≤ 0.01, (*)≤ 0.05; n.s. are $p > 0.05$. Box plots show the median (center line), 25–75th percentiles (box), whiskers extending to 1.5× the interquartile range, and outliers as individual points. **f** Accessibility promoter enrichment around TSS (± 1kb) in each group for pre-ZGA stage. **g** DNAme promoter enrichment around TSS (± 1kb) in each group for pre-ZGA stage. **h** Comparison of promoter CpG density around TSS (± 1kb) in each group for pre-ZGA stage. **i** Heatmap showing the z-scored mean of accessibility, CpG density and DNAme at promoters for available timepoints for the major H3K4me3 dynamics groups.

Groups with zygotically expressed genes, i.e., GZ and ZS groups, gained asymmetrically broader peaks during ZGA, likely due to active transcription[11,41] (Fig. S3c). Together, this indicates that, in agreement with our hypothesis, genes expressed in the gametes and embryos (GZ group) display high intensities and breadth of H3K4me3 around their promoters across the interjacent transcriptionally silent phase.

Furthermore, H3K4me3-enriched GZ-gene promoters also showed high DNA accessibility (Fig. S3f), DNA hypomethylation (Fig. S3g) and high CpG content (Fig. S3h) at pre-ZGA compared to other groups (Fig.3e). ZS-gene promoters showed lower chromatin accessibility (Fig. S3f), CpG content (Fig. S3h) and increased DNAme (Fig. S3g) compared to GZ-gene promoters at pre-ZGA (Fig. 3e).

Since GZ-genes, expressed in both gametes and embryos, showed similar characteristics to the SHARED group, we addressed their overlap (Fig. 3f). Indeed, over 80% of GZ-genes overlapped and were significantly associated with the SHARED group, but not with any of the other groups (Fig. 3g, h and Table S1). GS-genes were significantly associated with the LOST group, fitting with their loss of gene expression during development, while ZS-genes were associated with the GAINED group of genes, matching their de novo activation at ZGA stage (Fig. 3h). Consistently, genes characterized as both SHARED and GZ had higher levels of accessibility, CpG density and displayed low levels of DNAme also in spermatid and sperm (Fig. S3i).

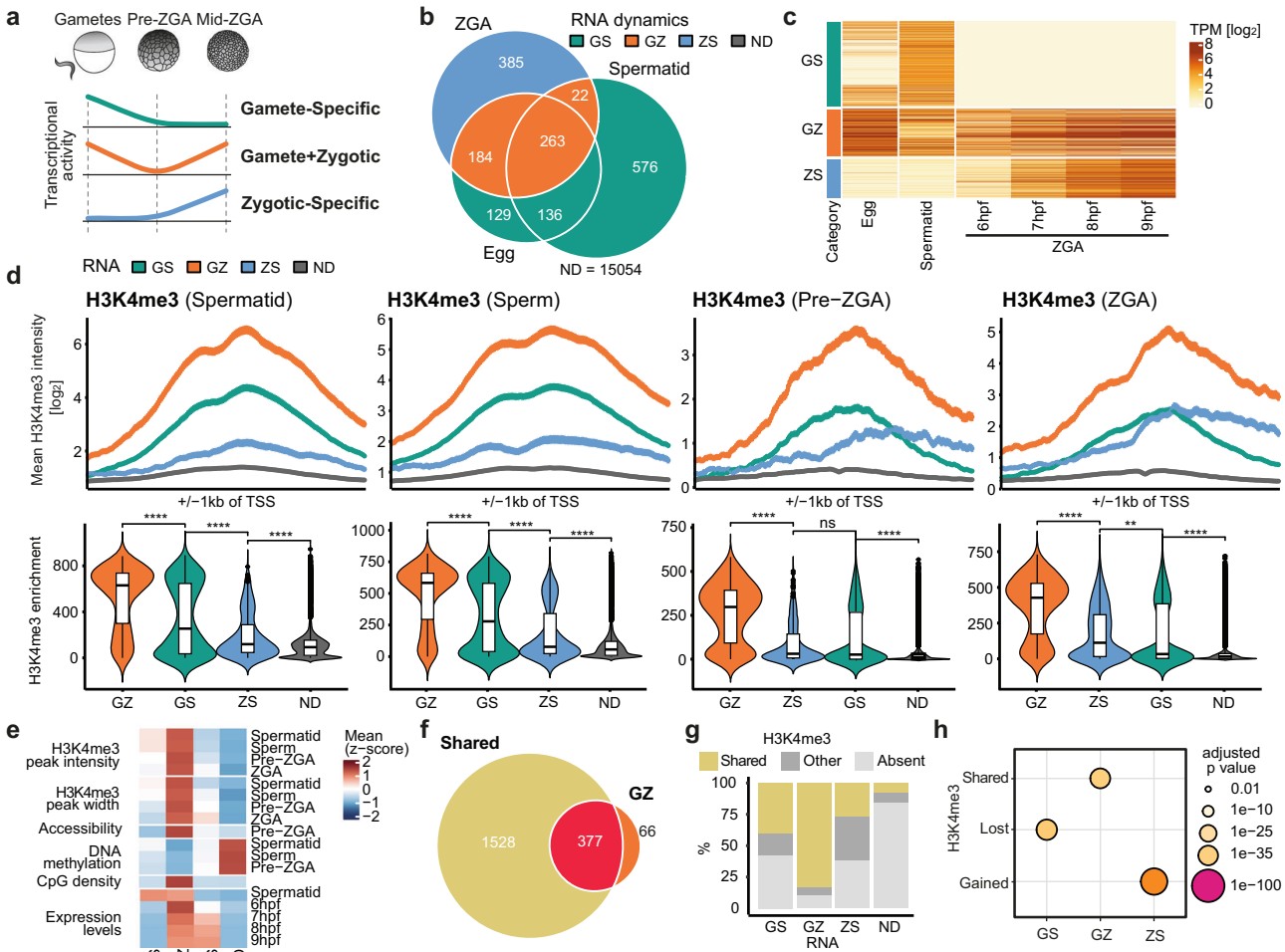

**Fig. 3 | H3K4me3 marks gene promoters independently of transcription between two transcriptionally active time points. a** Schematic representing transcriptional activity in gametes, pre-ZGA and ZGA embryos in each RNA group. Gamete-Specific (GS, $n = 841$): genes that are only expressed in the gametes, Gamete-Zygotic (GZ, $n = 469$): genes that are expressed both in the gametes and zygotically, Zygotic-Specific (ZS, $n = 385$): genes that are expressed only zygotically. **b** Venn diagram denoting overlap of genes detected in the egg, spermatid (total RNA) and at ZGA stages (nascent). Colors are based on the dynamics of RNA expression across stages. **c** Heatmap of RNA expression (log-scaled TPM) of the different RNA dynamics groups in the egg, spermatid and different timepoints during ZGA (6-9hpf). **d** (Top) H3K4me3 around TSS (±1kb) for RNA dynamics groups at each timepoint - spermatid, sperm, pre-ZGA, post-ZGA, respectively. Values displayed as mean ± SE. (Bottom) H3K4me3 promoter enrichment in each group for each time point. Statistical test: one-sided Wilcoxon rank-sum test with

alternative hypothesis that values of left distribution tend to be larger than right distribution; $p$: (****) ≤ 0.0001, (***) ≤ 0.001, (**) ≤ 0.01, (*) ≤ 0.05; n.s. are $p > 0.05$. Box plots show the median (center line), 25-75th percentiles (box), whiskers extending to 1.5× the interquartile range, and outliers as individual points. **e** Heatmap showing the z-scored mean of H3K4me3 peak intensity and peak width, accessibility, CpG density and DNAme at promoters and expression levels for available timepoints for the different RNA dynamics groups. **f** Overlap of SHARED H3K4me3 dynamics group and the GZ RNA dynamics group. **g** Composition of each RNA dynamics group with respect to the H3K4me3 dynamics groups. LOST and GAINED are summarized as OTHER. **h** Results of statistical tests for association between the H3K4me3 dynamics groups and RNA dynamics groups. *P*-values are calculated with one-sided Fisher's Exact Test for positive association and FDR adjusted.

While almost all GZ-genes also belong to the SHARED group of genes, the reverse is not true—not all genes of the SHARED group are also GZ-genes (Fig. 3f, g). This suggests that promoter H3K4me3 at pre-ZGA is not strictly indicative of gene expression at ZGA. SHARED genes whose transcripts were not detected at ZGA (Fig. 3f, g, GS and ND groups) may either be expressed below the detection limit of nascent RNA-seq or later in development. This suggests that promoters of genes actively expressed in gametes and embryos were also marked by H3K4me3.

Our results reveal a close correlation between H3K4me3 dynamics and gene expression dynamics in the early embryo. We uncover an association between genes that are pre-marked by H3K4me3 during early development and genes that are transcribed in both spermatid and ZGA, a finding compatible with H3K4me3 driving the maintenance of active chromatin states.

## Genes pre-marked with H3K4me3 are expressed early during ZGA

H3K4me3 at promoters and in gene bodies is reported to be important for high processivity of RNA pol II and, consequently, for fast, efficient gene transcription[26]. Thus, we hypothesized that pre-marking of genes with H3K4me3 prior to ZGA could facilitate their early and fast expression during ZGA.

We analyzed nascent RNA-seq data[43] and classified all zygotically expressed genes into four groups based on the timepoint at which their transcripts were first detected (Fig. 4a). Indeed, the earliest identified transcripts at 6 hours post-fertilization (hpf) were from genes significantly associated with the SHARED group, whereas later-expressed genes (7-9 hpf) were associated with both SHARED and GAINED groups (Fig. 4b). Similarly, the earliest expressed genes (6 hpf) were associated with the GZ group, and the significance of association

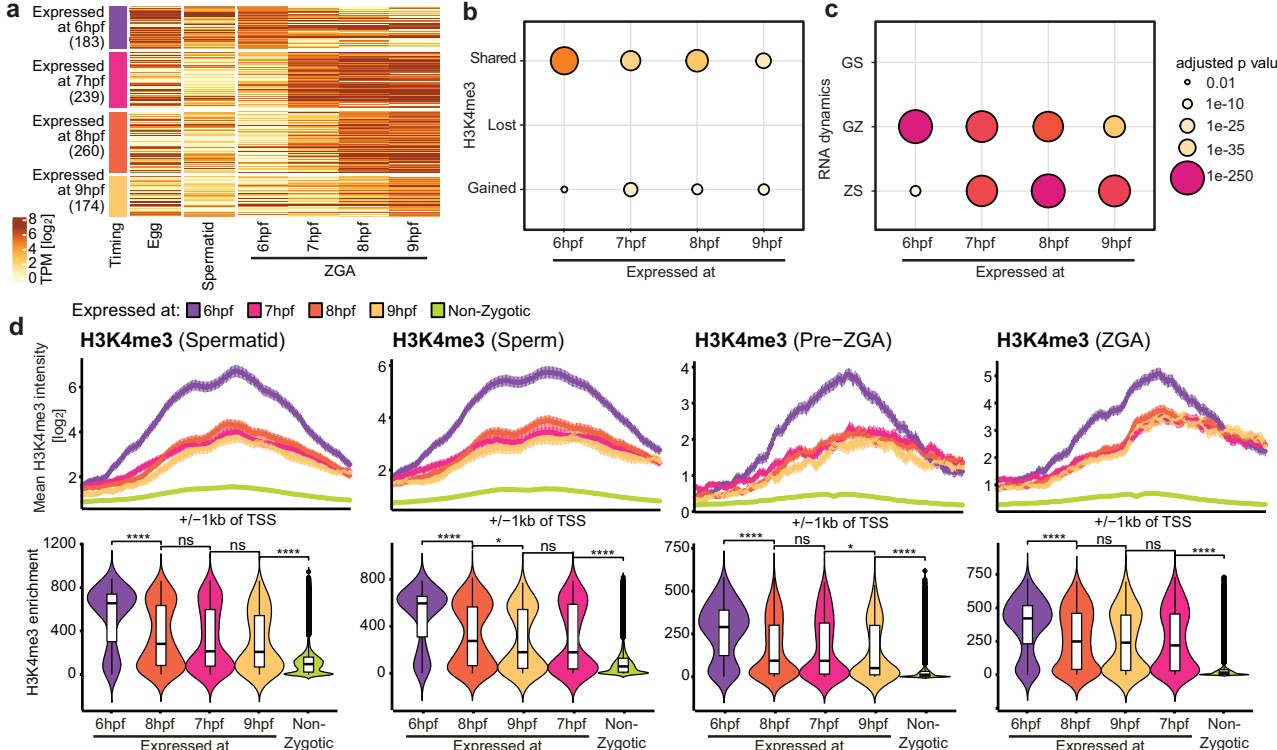

**Fig. 4 | Genes pre-marked by H3K4me3 are expressed early during zygotic genome activation. a** Heatmap of expression of zygotic genes in egg, spermatid, and different timepoints during ZGA grouped by their first detected expression timepoint during ZGA (6 hpf: $n = 183$, 7 hpf: $n = 239$, 8 hpf: $n = 260$, 9 hpf: $n = 174$). Zygotic genes = Gamete+Zygotic (GZ) + Zygotic-Specific (ZS). **b** Results of statistical tests for association between H3K4me3 dynamics groups and gene groups based on their first-detected expression timepoint during ZGA (6-9 hpf). *P*-values are calculated with one-sided Fisher's Exact Test for positive association and FDR adjusted. **c** Results of statistical tests for association between RNA dynamics groups and gene groups based on their first-detected expression timepoint during ZGA (6-9 hpf). **d** (Top) H3K4me3 around TSS (±1kb) for expression timing gene groups at each timepoint - spermatid, sperm, pre-ZGA, post-ZGA. Values displayed as mean ± SE. (Bottom) H3K4me3 promoter enrichment in each group for each time point. Statistical test: one-sided Wilcoxon rank-sum test with alternative hypothesis that values of left distribution tend to be larger than right distribution; *p*: (****) ≤ 0.0001, (***) ≤ 0.001, (**) ≤ 0.01, (*) ≤ 0.05; n.s. are *p* > 0.05. Box plots show the median (center line), 25–75th percentiles (box), whiskers extending to 1.5× the interquartile range, and outliers as individual points.

progressively decreased over time (Fig. 4c). Conversely, genes with later-detected transcripts (7–9 hpf) were associated with the ZS group, with the significance of association increasing with time (Fig. 4c).

We further found that genes expressed at 6 hpf had stronger promoter H3K4me3 peak intensities than other groups at gamete, pre-ZGA and ZGA stages (Fig.4d). No significant differences in peak intensities were observed among later-expressed genes, while non-zygotic genes remained significantly depleted (Fig.4d). Later-expressed genes (7–9 hpf) gained H3K4me3 asymmetrically from their promoter region towards the 3′ end as ZGA began, with peak breadth increasing in parallel (Fig. S4a), reminiscent of the trend seen for gene promoters in the GAINED group and reported previously[11] (Fig. S2f).

Promoters of early-expressed genes had high CpG density, in contrast to the low enrichment at non-zygotic promoters (Fig. S4b). At pre-ZGA, DNAme was reduced at early-expressed gene promoters but remained high at non-zygotic and later-expressed genes, suggesting that methylation is present at loci not yet poised for expression (Fig. S4c). This was consistent with the negative correlation between DNAme, nascent expression levels, and H3K4me3 abundance at pre-ZGA (Fig. S4d). Chromatin accessibility distinguished between expressed from non-zygotic genes, but did not correlate with gene expression timing (Fig. S4e). Together, these features highlight that early-expressed genes largely overlap with genes from the SHARED group, and are associated with CpG-rich, accessible, hypomethylated promoters marked by H3K4me3, facilitating timely transcriptional onset (Fig. S4f).

In summary, we uncovered a correlation between H3K4me3 dynamics and gene expression dynamics in early embryos. We find that H3K4me3 marks promoter regions of genes transcribed in the gametes, silenced across multiple cell divisions, and reactivated early during ZGA.

## H3K4me3 marking of chromatin during early embryonic divisions is required for proper ZGA

Having established that H3K4me3 is detectable at promoter regions of genes after multiple cell divisions when transcription is globally paused, we next investigated whether this pre-marking is also functionally linked to gene expression dynamics at ZGA. We aimed to determine whether disrupting H3K4 methylation in the cell cycles preceding ZGA, but not during ZGA itself, affects the proper activation of gene expression at ZGA.

We established an assay to reduce H3K4 methylation levels in embryos between fertilization and ZGA. We used a combination of (1) overexpressing the H3K4-specific demethylase Kdm5b$^{wt}$ to remove the methyl residues and (2) expression of the H3.3 variant with a lysine-to-methionine mutation H3.3$^{K4M}$, sequestering H3K4 methyltransferases and preventing deposition of new methyl residues in a dominant negative manner[44–46]. The catalytically inactive demethylase Kdm5bci, combined with wild-type H3.3wt, served as a control treatment. All mRNA transcripts were equipped with a C-terminal 3xHA tag and an auxin-inducible degron (AID) sequence to enable their degradation via TIR1 prior to the ZGA stage.

A combination of Kdm5b[wt] and H3.3[K4M] mRNAs, or a combination of their catalytic inactive counterparts Kdm5b[ci] and H3.3 [wt] as the control, was co-injected into one-cell embryos together with TIR1 (Fig. 5a). We confirmed their expression before ZGA (Fig. 5b), resulting in a decrease of H3K4me3 in pre-ZGA embryos expressing Kdm5b[wt] and H3.3[K4M] (Fig. 5c) compared to the control conditions. After 8 cell divisions, auxin-induced degradation of overexpressed proteins restricted the disruption of H3K4 methylation within a "window" from the zygote to the pre-ZGA time-point (Fig. 5a). We observed complete degradation of the ectopic proteins within 2 h (Fig.5b).

To validate this system, we tested whether H3K4me3 restoration in the "window" treatment condition occurs in a genome-wide localized manner. We performed CUT&RUN against H3K4me3 at 6.5 hpf in window-treated and control embryos, alongside wild-type embryos (Fig. 5d). To compare the genome-wide H3K4me3 signal in control and treated embryos, we defined a consensus peak set as regions present in at least two of three independent CUT&RUN replicates in wild-type embryos. Principal component analysis of spike-in-normalized H3K4me3 counts at these peaks revealed that window-treated samples clustered more closely with their respective controls and with wild-type samples than with the persistently treated samples in the PC space (Fig. S5a). This indicates that H3K4me3 levels in the window-treated samples have recovered to those of wild-type and control samples at mid-ZGA. To test this, we performed differential peak analysis within the consensus peak set and found very few differentially marked promoters between window-treated and control embryos (67 up, 59 down; $p < 0.05$) (Fig. 5e). In contrast, differential analysis revealed substantially more altered peaks in persistent-treated embryos (393 up, 358 down; $p < 0.05$) (Fig. 5e). Taken together, these analyses suggest a widespread recovery of promoter-associated H3K4me3 in the window- compared to the persistent-treatment condition

Next, we tested whether temporary disruption of H3K4me3 during early embryonic cell division affects the onset of ZGA. We treated embryos as described above (Fig. 5a) and analyzed them at three stages spanning the onset of ZGA (5.5 hpf, 6.5 hpf, 7.5 hpf) by RNA-seq. When comparing embryos with perturbed H3K4me3 propagation against control embryos, we detected significant up- and down-regulation of transcripts, with the number of differentially expressed genes (DEGs) increasing with time (Fig. S5b). Downregulated genes from all timepoints overlapped well between biological replicates (Fig. S5c). Using our CUT&RUN data, we confirmed that at 6.5 hpf, downregulated zygotic genes in the window condition had recovered their H3K4me3 levels, whereas those in the persistent condition remained significantly depleted, while unaffected genes showed no significant change in promoter H3K4me3 enrichment (Fig. S5d). Notably, we observed a strong bias of GZ- and ZS-genes among these downregulated genes in both "window" and "persistent" depletion conditions (Figs.5f and S5e). Hence, we focused further analysis on all genes found to be expressed at ZGA (GZ+ZS groups of genes).

Although none of our injection conditions fully disrupted the overall dynamics of global transcriptional activation during ZGA, as indicated by the comparable increase in mean transcript levels of ZGA genes between control-injected and non-injected wild-type embryos (Fig. S5f), we observed that, on average, 25% of these zygotic genes were significantly downregulated at 7.5 hpf. In contrast, very few zygotic genes were upregulated in treatments compared to controls (Fig. S5g). Interestingly, downregulated genes from all timepoints together consistently overlapped across "window" and "persistent" depletion conditions (Fig. S5h). The expression of zygotic genes displaying promoter H3K4me3 in normal embryos at all investigated timepoints (SHARED group) and genes gaining promoter H3K4me3 during ZGA (GAINED group) were strongly affected by both treatments, with expression reductions approaching those observed upon transcriptional inhibition with triptolide[47] (Figs.5g and S5i). Mean

expression levels of the SHARED group were significantly reduced in treated embryos of the "window" depletion at 7.5 hpf compared to controls, suggesting a direct link between H3K4me3 pre-marking prior to ZGA and accurate gene activation during ZGA (Figs.5h and S5j). Also, genes gaining promoter H3K4me3 around ZGA (GAINED group) were affected by both "persistent" and "window" treatments. Although the mean expression levels of these genes at 7.5 hpf remained significantly lower in treated embryos compared to controls (Figs.5h and S5j), H3K4me3 levels at promoters were largely restored in the window condition (Fig. S5k). In contrast, promoter H3K4me3 levels remained significantly depleted in the persistent condition (Fig. S5k), consistent with the observed downregulation of zygotic gene expression (Figs.5h and S5j). Together, this indicates that a temporary disruption of H3K4 methylation affects the reliable expression of genes accompanied by de novo H3K4me3 deposition at promoters during ZGA. We also observed some downregulated zygotic genes from the ABSENT group, possibly due to secondary effects of the treatment (Figs.5g and S5i).

Interestingly, among the downregulated genes in the SHARED group under the window treatment is *pou5f3.2*, a TF that, together with Sox3, facilitates ZGA in *Xenopus laevis* (Fig.5i)[47,48]. *Pou5f3.2* downregulation by the window treatment could potentially affect the expression of zygotic genes that are normally activated by this TF. In contrast, we found that the fold change in *sox3* expression at 7.5 hpf is only mildly reduced in window-treated embryos relative to controls, which likely stems from *sox3.S* and *sox3.L* being GZ genes with strong maternal contribution (Fig.5h). Overall, persistent reduction of H3K4me3 produced only a slightly stronger phenotype than that of the shorter window treatment, suggesting that presence of pre-ZGA H3K4 methylation, in addition to H3K4me3 acquisition during ZGA, is important for accurate activation of many zygotic genes.

We then tested whether H3K4 methylation in pre-ZGA embryos is crucial for embryonic development. Injection of mRNA at the concentrations determined above (Fig. S6a, b; 1x mRNA dosage) led to the survival failure of both treated and control embryos (Fig. S6c, d), likely due to mRNA overload or roles of the injected proteins other than H3K4 demethylation. Upon halving the dosage (0.5x mRNA dosage), we still observed a reduction in H3K4me3 in both treatment conditions compared to controls (Fig. S6a, b). Importantly, under these conditions, control embryos showed improved survival, while embryos with either window or persistent H3K4me3 reduction continued to exhibit developmental failure (Fig. S6c, d). Together, these results indicate that loss of H3K4 methylation contributes to the observed developmental defects. Notably, even transient reduction of H3K4me3 during the pre-ZGA window led to increased embryonic lethality, indicating that H3K4me3 marking of chromatin prior to ZGA contributes to successful development (Fig. S6e).

In summary, our experiments show that interfering with H3K4 methylation in the globally transcriptionally silent window affects proper ZGA. Additionally, we show that H3K4 methylation in pre-ZGA stages supports successful embryonic development.

## Cxxc1 assists transcription-independent deposition of H3K4me3 and zygotic gene expression

We next examined the mechanisms underlying H3K4me3 marking of promoters during the transcriptionally quiescent pre-ZGA window, focusing on enzymes reported to act independently of transcription. In mouse ESCs and oocytes, CFP1 and MLL2 function as transcription-independent H3K4 methyltransferases[42,49–51] and their *Xenopus* orthologs, *Cxxc1* and *Kmt2b*, are expressed during pre-ZGA[39]. Both enzymes contain a CXXC domain that recognizes unmodified CpGs and recruits the methyltransferase. Consistently, promoters displaying H3K4me3 at all investigated stages are characterized by high CpG density and low DNAme (Figs.2i and 3e). These observations support a model in which Cxxc1 and Kmt2b mediate transcription-independent H3K4

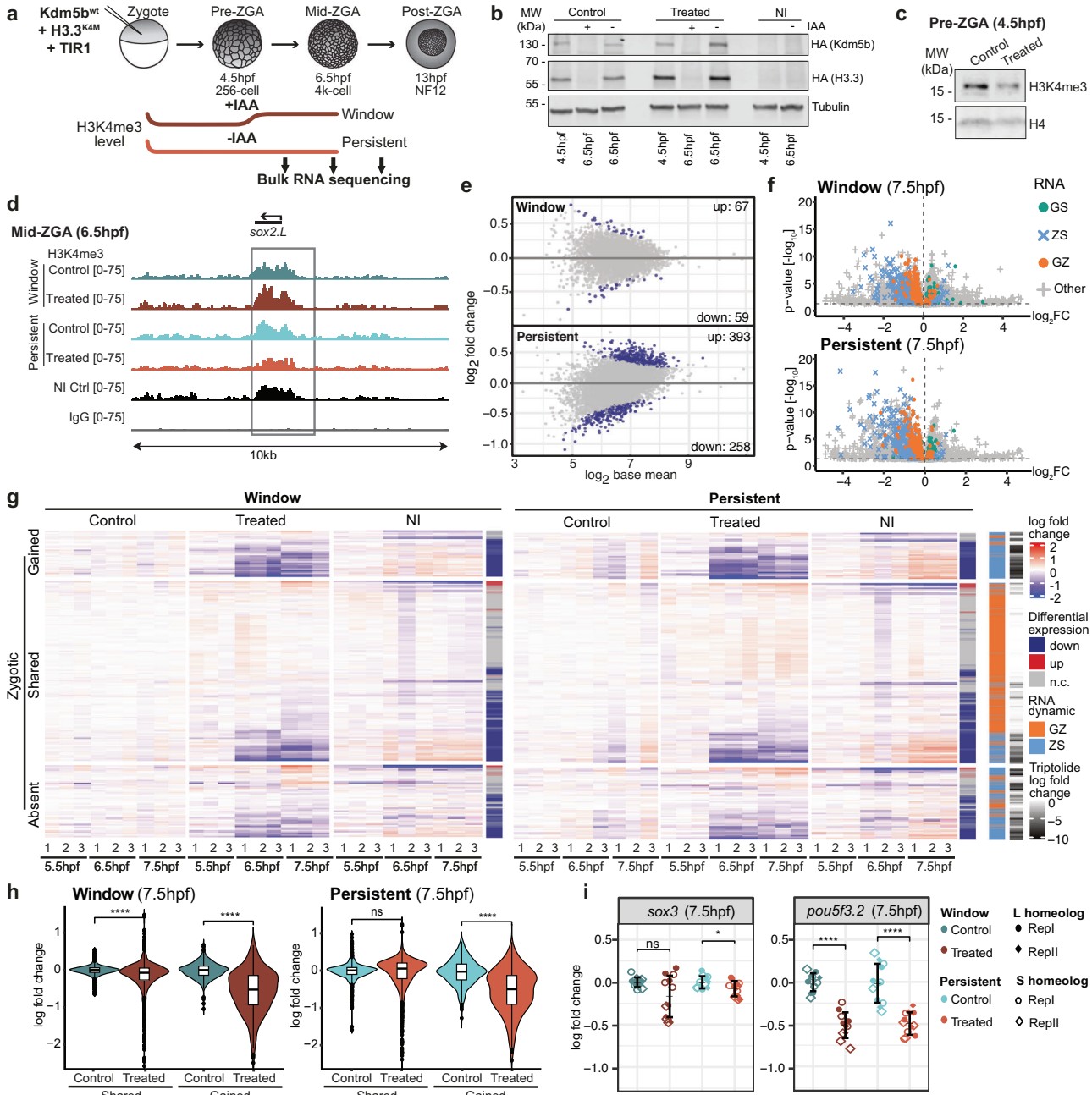

**Fig. 5 | Promoter H3K4me3 during early cell divisions is required for proper ZGA. a** Experimental setup for window-specific and persistent H3K4me3 depletion. **b** Western blot of cytoplasmic fractions from pre- and mid-ZGA embryos showing ectopic HA-tagged Kdm5bwt + H3.3K4M (treatment) or Kdm5bci + H3.3 wt (control) at pre-ZGA, and their auxin-induced (IAA) degradation at mid-ZGA. Tubulin represents the loading control; "NI" denotes non-injected. Blots were independently repeated 2–3 times with consistent results. **c** H3K4me3 intensity relative to H4 in treated and control pre-ZGA embryos after chromatin-bound protein isolation. **d** Genome browser snapshots for *sox2.L* showing *E. coli* spike-in normalized coverage for control and treated samples under window and persistent conditions, and wildtype and IgG controls (biological replicate 2). **e** MA-plot comparing H3K4me3 levels in treated versus control samples at promoter peaks for window (top) and persistent (bottom) conditions. Gray: all promoter peaks subset from the consensus peak set in (**b**). Blue: significantly changed ($p < 0.05$). **f** Volcano plots displaying log2 fold change and adj. *p*-value (cutoff: 0.05) at 7.5 hpf, colored by RNA dynamics groups for (top) window and (bottom) persistent conditions (biological replicate 1). **g** Zygotic gene expression (three technical replicates per

sample), clustered by H3K4me3 dynamics groups (SHARED, GAINED, ABSENT) (biological replicate 1). Log2 fold changes are relative to the mean control TPM at each time point. Genes are annotated for differential expression in each condition and by RNA dynamics group. The rightmost column displays triptolide-induced log2 fold changes at 7 hpf (adapted from Phelps et al., 2022), representing the expected maximal transcriptional downregulation at this stage. **h** Log2 fold change of zygotic SHARED ($n = 473$) and GAINED ($n = 121$) genes over the mean TPM of respective control technical replicates in window (left) and persistent (right) conditions at 7.5 hpf in biological replicate 1. One-sided Wilcoxon rank-sum test with alternative hypothesis that values of the left distribution tend to be larger than the right distribution; *p*: (****) ≤ 0.0001, (***) ≤ 0.001, (**) ≤ 0.01, (*) ≤ 0.05; n.s. >0.05. Box plots show the median (center line), 25–75th percentiles (box), whiskers extending to 1.5× the interquartile range, and outliers as individual points. **i** Log2 fold change expression of *pou5f3.2* and *sox3* homeologs over mean TPM of respective controls at 7.5 hpf ($n = 12$); both biological replicates shown (distinct point styles). Error bars indicate mean ± SD.

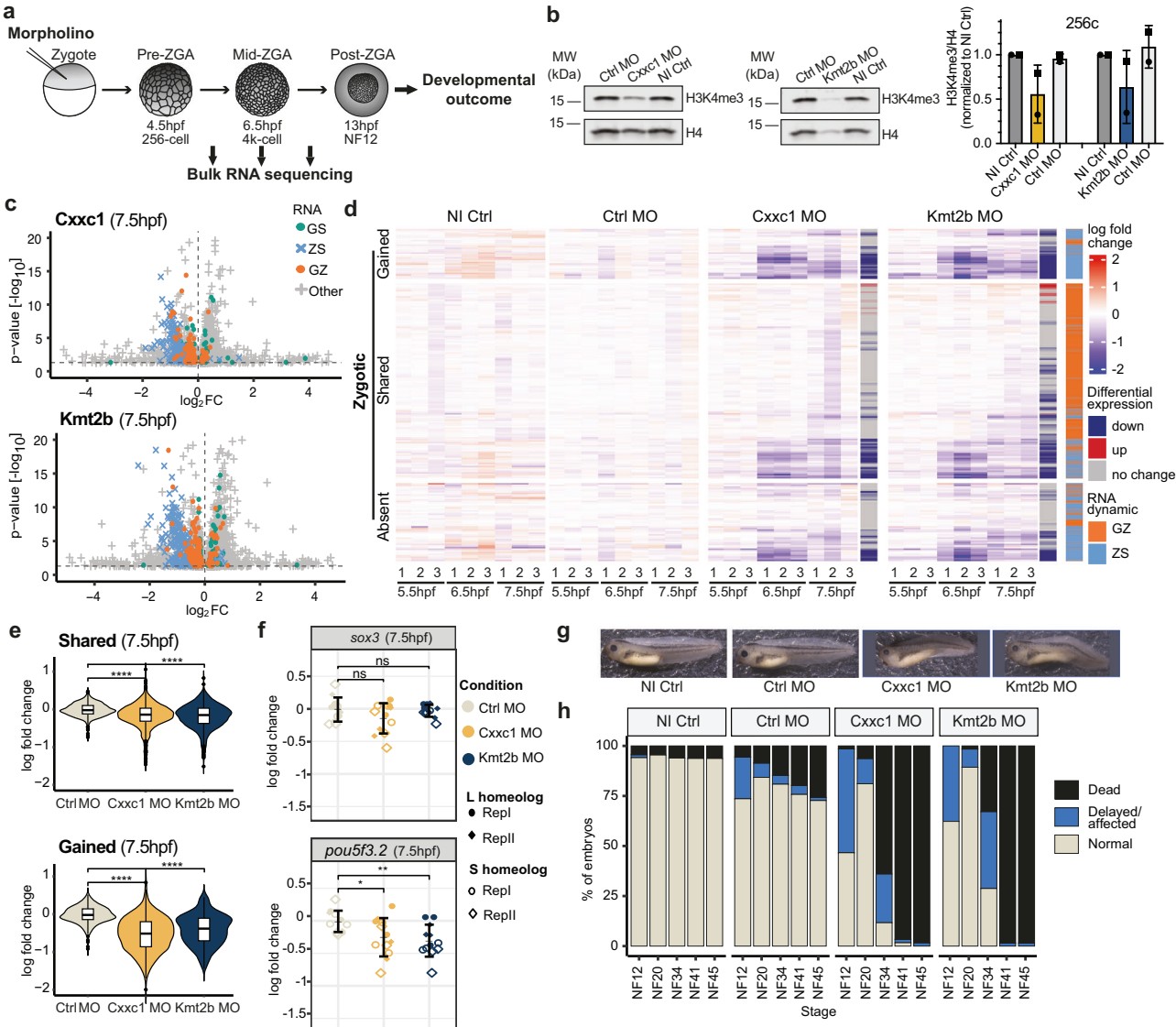

**Fig. 6 | Cxxc1 and Kmt2b are required for proper ZGA and embryonic development. a** Illustration of experimental setup. **b** (Left) Western blot showing H3K4me3 levels in Cxxc1 and Kmt2b knockdown embryos compared to non-injected control and control morpholino-injected embryos at pre-ZGA. (Right) Quantification of H3K4me3 signal intensity relative to H4 for each condition for three independent experiments. Data are presented as mean values ± SD. **c** Volcano plots displaying expression log2 fold change and adj. *p*-value (cutoff: 0.05) in knockdown conditions compared to control morpholino for biological replicate 1 at 7.5 hpf, colored by RNA dynamics groups. **d** Expression of zygotic genes in 3 technical replicates of every sample for biological replicate 1, clustered by H3K4me3 dynamics group (SHARED, GAINED and ABSENT). Log2 fold change of every gene is calculated over the mean TPM of three technical replicates of the control morpholino of the respective time point. Genes are annotated with differential expression status separately for the Cxxc1 and Kmt2b knockdown conditions, as well as by RNA dynamics group. **e** Log2 fold change of gene expression

levels of all zygotic genes calculated over mean TPM of control morpholino technical replicates for biological replicate 1 for SHARED (top, *n* = 473) and GAINED (bottom, *n* = 121) groups at 7.5 hpf. Statistical test: one-sided Wilcoxon rank-sum test with alternative hypothesis that values of left distribution tend to be larger than right distribution; *p*: (****) ≤ 0.0001, (***) ≤ 0.001, (**)≤0.01, (*) ≤ 0.05; n.s. are *p* > 0.05. Box plots show the median (center line), 25–75th percentiles (box), whiskers extending to 1.5× the interquartile range, and outliers as individual points. **f** Log2 fold change expression of *pou5f3.2* and *sox3* calculated over mean TPM of control morpholino replicates at 7.5 hpf (*n* = 12). Both biological replicates are represented, each by a different point style. Error bars indicate mean ± SD. **g** Representative phenotype images of embryos for each condition at NF41. *n* = 2 experiments, *N* > 35 per condition. **h** Quantification of survival phenotype. *n* = 2 experiments, *N* > 35 embryos per condition. Source data are provided as a Source Data file.

trimethylation via sequence-specific hypomethylation-related reader-writer mechanisms.

We tested whether Cxxc1 and Kmt2b contribute to H3K4 methylation in early *Xenopus* embryos and proper gene activation during ZGA. We injected translation-blocking antisense morpholinos (asMOs) against both homeologs of *kmt2b* and *cxxc1*, or control morpholinos (ctrlMO), into one-cell embryos and allowed them to develop (Fig.6a). We observed global depletion of Cxxc1 and Kmt2b protein levels (Fig. S7a) and correspondingly reduced H3K4me3 levels in asMO-

injected embryos compared to ctrlMO-injected embryos at both pre-ZGA (Fig.6b) and post-ZGA (Fig. S7b).

We then investigated the effect of this treatment on ZGA. We performed RNA-seq on Cxxc1 and Kmt2b knockdown embryos at three time points spanning the onset of ZGA (Fig.6a). Differential gene expression analyses revealed significant up- and downregulation of transcripts in both Cxxc1 and Kmt2b knockdown embryos, compared to the controls, with the number of DEGs increasing with developmental time (Fig. S7c). We identified overlap of downregulated genes

across biological replicates in both Cxxc1 and Kmt2b knockdown conditions (Fig. S7d). We observed a bias of GZ- and ZS-genes among the downregulated genes (Figs.6c and S7e) and focused further analysis on all zygotically expressed genes.

We found that an average of 10% and 16% of zygotic genes were downregulated in Cxxc1 and Kmt2b knockdown embryos, respectively, at 7.5 hpf (Fig. S7f), with an overlap across both conditions (Fig. S7g). Out of the downregulated zygotic genes, those belonging to the groups that keep H3K4me3 around their promoters during early embryonic development (SHARED) and those that gain H3K4me3 during ZGA stage (GAINED) were affected (Figs. 6d and S7h). At 7.5 hpf, we observed significantly decreased mean expression levels of genes in the SHARED and GAINED groups in Cxxc1 or Kmt2b knockdown versus control embryos (Figs. 6e and S7i). Both conditions affected the expression of genes involved in development or in transcription regulation (Fig. S7j). In both Cxxc1 and Kmt2b asMO-injected embryos, we observed a significantly reduced fold change in *pou5f3.2* expression compared to controls, but only a slight reduction in *sox3* expression at 7.5 hpf, explained by its maternal contribution (Fig. 6f). These observations link Cxxc1 and Kmt2b to H3K4me3 propagation and acquisition during *Xenopus* development and demonstrate a role for these enzymes in proper ZGA. To link early chromatin regulation to later transcriptional changes, we compared the genes affected upon Cxxc1 and Kmt2b knockdown with those sensitive to H3K4me3 loss (Fig. 5), revealing an overlap across all four conditions (Fig. S7o). This supports an impact of early H3K4me3 marking on gene expression programs.

We then addressed whether Kmt2b and Cxxc1 depletion affected development. Both Cxxc1 and Kmt2b knockdown embryos showed delayed development compared to control and wildtype embryos, as well as morphological defects, including aberrant gut formation, an overall twisted, shorter body axis, and a "bent tail" phenotype (Fig. 6g). Tadpoles with methyltransferase knockdown did not survive past the feeding tadpole stage (4 days post fertilization), suggesting that Cxxc1 and Kmt2b are each essential for development (Fig. 6h). It is yet unclear if their function is important for successful development only during pre ZGA stages or also during later stages.

To test phenotype specificity, we generated and validated MO-resistant *cxxc1* mRNAs (Fig. S7k). Injection of regular *cxxc1* mRNAs or Cxxc1 asMOs alone caused poor embryonic survival (Fig. S7l–n). However, co-injection of MO-resistant *cxxc1* mRNA with the Cxxc1 asMO resulted in partial rescue of the phenotype, indicating asMO specificity for Cxxc1 (Fig. S7l–n). Rescue experiments for Kmt2b could not be performed due to technical problems. Nevertheless, a substantial overlap between genes downregulated upon Cxxc1 and Kmt2b knockdown and those sensitive to H3K4me3 loss (Fig. S7o) supports a shared role of these enzymes in regulating gene expression and accentuates the importance of H3K4me3 marking on proper embryonic gene expression.

In summary, our results reveal that the methyltransferases Cxxc1 and likely Kmt2b regulate accurate activation of zygotic genes marked by H3K4me3 during embryogenesis and are important for embryonic development. These findings underscore the importance of H3K4me3 reader–writer enzymes in ensuring proper H3K4me3 patterns, facilitating accurate zygotic gene expression, and supporting successful development.

## Discussion

This study shows that early embryonic chromatin is pre-marked with H3K4me3 and reveals its crucial role in proper ZGA and development. We show that H3K4me3 is found at common promoter loci in gametes, transcriptionally silent pre-ZGA embryos, and during ZGA when transcription begins. We show that pre-marking chromatin with H3K4 methylation supports correct ZGA and is facilitated by H3K4 methyltransferases Cxxc1 and likely Kmt2b, whose disruption results in reduced H3K4me3 levels, aberrant ZGA, and developmental failure.

Thus, these findings support a role of H3K4 methylation in mediating epigenetic memory of active chromatin states.

We find that numerous histone modifications are present on pre-ZGA chromatin, including active modifications. This is unexpected given the dynamic nature of early embryonic chromatin, which replicates approximately every 30 min with repressed transcription. We observe that transcription-independent Cxxc1 contributes to H3K4me3 deposition[42,50] and that Kmt2b may also deposit canonical promoter H3K4me3 in pre-and post-ZGA *Xenopus* embryos, in addition to its known role in establishing broad non-canonical domains in mouse oocytes[51]. These enzymes likely recognize CpG-rich, hypomethylated promoters and reinstate H3K4me3 after DNA replication, which may be enhanced by broad H3K4me3 domains across several nucleosomes. However, evaluating H3K4me3 at discrete stages cannot resolve continuous transmission at SHARED promoters, and additional contributions from TFs or DNA sequence remain possible.

We find that H3K4me3 occurs at pre-ZGA despite global transcription repression and the absence of the original inducing signal. While current interference methods cannot distinguish between H3K4me2/3, loss of H3K4 methylation impairs gene activation and embryonic development, consistent with H3K4 methylation contributing to epigenetic memory of active chromatin states[13,14,52]. Whether other active histone modifications detected at pre-ZGA show similar dynamics remains unexplored.

H3K4me3 and DNAme are mutually antagonistic, as H3K4me3 impedes de novo DNAme by inhibiting DNMT3 binding[34,53]. Conversely, hypomethylated DNA recruits H3K4 methyltransferases via the CXXC domain[50,51,54]. This raises the question whether early H3K4me3 reflects underlying DNAme patterns, or vice versa. Previous work in *Xenopus tropicalis* hypothesized that H3K4me3 is absent in cleavage-stage embryos and acquired during ZGA based on DNAme, favoring hypomethylated, CpG-rich regions[11]. Our analyses support a distinct model for H3K4me3 in *Xenopus laevis*, consistent with recent hybrid *Xenopus* findings[18]. We demonstrate that H3K4me3 is present in gametes and at both pre- and post-ZGA. These findings suggest that the two frog species have diverged in ZGA regulatory mechanisms. Nevertheless, promoter hypomethylation in *Xenopus laevis* pre-ZGA embryos mirrors trends reported in *Xenopus tropicalis* mid-ZGA embryos[38].

What is the role of H3K4me3 during early development? In previous experiments in zebrafish and mouse addressing the roles of histone modifications and variants such as H3K4me3, H3K27ac, H3K9me3, H3K27me3, H2AK119ub and H2A.Z in supporting ZGA, embryos were typically subjected to interference from fertilization onwards[24,44,55–68]. Unfortunately, this fails to distinguish whether the propagation of the histone mark before ZGA or the accumulation of the mark throughout early development is important for establishing accurate gene expression patterns at ZGA. We show that H3K4 methylation perturbation within a time window before ZGA has a gene expression phenotype almost as strong as persistent depletion, suggesting that pre-ZGA H3K4me3 marking is important for proper onset of ZGA and successful development.

Canonical pre-ZGA H3K4me3 patterns in *Xenopus* closely resemble those of human oocytes and embryos[69], in contrast to non-canonical patterns in mouse oocytes[58]. However, *Xenopus* and human embryos may transiently exhibit mouse-like H3K4me3 patterns at an earlier developmental stage. Mll2 is expressed in early human embryos and depleted as ZGA begins, much like the expression patterns of its ortholog Kmt2b in *Xenopus*[69]. In context with other species-similarities, such as the higher degree of paternal histone retention after fertilization in both species and multiple transcriptionally silent cell divisions before ZGA[12,70–72], *Xenopus* may provide unique insights into mechanisms supporting epigenetic memory of active states in pre-ZGA human embryos.

We find that early gene reactivation during ZGA correlates with H3K4me3 pre-marking. Because promoter and gene-body H3K4me3 are important for high RNA Pol II processivity[26], pre-ZGA H3K4me3 at promoters may aid in rapid gene expression early during ZGA. Interestingly, TFs reported to facilitate ZGA in *Xenopus*, Sox3 and Pou5f3.2[47,48], are amongst those genes. Their combinatorial knockdown causes developmental arrest at gastrulation[47,48] in *Xenopus*. Both persistent and window disruption of H3K4 methylation before ZGA result in significant downregulation of *pou5f3.2* and, to a lesser extent, *sox3*, likely causing decreased protein levels and downstream ZGA effects.

In summary, our findings highlight that H3K4me3 pre-marking of promoter regions in early embryos is essential for subsequent accurate ZGA and development, with Cxxc1 ensuring proper deposition at hypomethylated, CpG-rich loci.

## Methods

### Xenopus laevis culture

Adult *Xenopus Laevis* were obtained from Nasco (901 Janesville Avenue, P.O. Box 901, Fort Atkinson, WI 53538-0901, USA) and Xenopus1 (Xenopus1, Corp. 5654 Merkel Rd. Dexter, MI. 48130). All frog maintenance and care were conducted according to the German Animal Welfare Act. Research animals were used following guidelines approved and licensed by ROB-55.2-2532.Vet_02-23-126.

### Embryo collection

Sexually mature females were pre-primed for ovulation by injection of 150 IU of Ovogest 300 I.E./ml into their dorsal lymph sac 4–5 days prior to experiments. 500 IU of the same hormone at 1000 I.E./ml was injected on the day before experiments, and the females were kept overnight at 14 °C in 1x MMR (Marc's Modified Ringer's buffer: 100 mM NaCl, 2 mM KCl, 1 mM MgSO4, 2 mM CaCl2, 0.1 mM EDTA, 5 mM HEPES pH 7.8). Females were then moved to room temperature to lay eggs. Eggs were collected in a timely manner, and in vitro fertilization was performed.

### In vitro fertilization

In vitro fertilization was performed on eggs by shredding a piece of freshly dissected testes between the two arms of a riffled pair of forceps to release the sperm in a glass dish containing eggs in a minimal amount of 1x MMR. After a 2-min incubation, the 1x MMR was diluted approximately 10-fold with distilled water, and the fertilized eggs were kept at room temperature for 20–30 min. Subsequently, the jelly coating on the fertilized eggs was chemically removed using a dejellying solution (2% L-Cysteine in milliQ $H_2O$, pH 7.8; adjusted by NaOH), followed by washes in 0.1x MMR.

### Microinjections

For labeling nascent RNA, embryos were injected with 9.2 nl of 50 μM 5-EU at the 1-cell stage using a NANOINJECT II microinjector (Drummond Scientific Company). For transcription inhibition, embryos were co-injected with 4.6 ng α-amanitin. For all injection experiments, embryos were kept in 0.5x MMR during and up to 2 h post-injections, after which they were transferred to 0.1x MMR for further development.

### Chromatin-bound protein isolation of pre-ZGA embryos

Two hundred 256-cell stage embryos per sample were collected at the desired stage. Buffer was removed, and embryos were gently washed thrice using embryo extraction buffer (10 mM HEPES pH 7.7, 100 mM KCl, 50 mM Sucrose, 1 mM MgCl2, 0.1 mM CaCl2) and then spun down for 1 min at 700 g. All excess liquid was removed, and samples were stored at −80 °C until further processing. Frozen embryos were thawed on ice and spun at $17,000 \times g$ for 10 min at 4 °C. This results in the formation of three layers: the upper lipid layer and the bottom

layer of cellular debris are avoided, and the middle gray layer of nuclei is collected using a P200 pipette. The nuclear fraction is washed twice with SuNASP buffer (250 mM Sucrose, 75 mM NaCl, 0.5 mM Spermidine, 0.15 mM Spermine) with a 5-min spin at 4 °C at 3500 g and then twice with RIPA buffer (50 mM Tris-HCl pH 7.4, 1% NP40, 0.25% Na-deoxycholate, 150 mM NaCl, 1 mM EDTA, 0.1% SDS, 0.5 mM DTT, 5 mM Na-butyrate) supplemented with protease inhibitors, spun for 10 min at $14,000 \times g$ at 4 °C. A Kimwipe is used to aid the removal of yolk and debris caught on the walls of the tube while removing the supernatant between washes. For SDS-PAGE, the pellet was resuspended in 2x E1 buffer (50mM HEPES-KOH, pH 7.5, 140mM NaCl, 0.1mM EDTA, pH 8.0, 10% Glycerol, 0.5% Igepal CA-630, 0.25% Triton X-100, 1% β-mercaptoethanol) and 4x Laemmli sample buffer (BIO-RAD, #1610747) supplemented with 10% β-mercaptoethanol. An equivalent of 50 embryos was loaded per well.

### Western blotting

For testing histone modifications in gastrula (NF stage 12) embryos, 1–3 embryos were collected and resuspended in E1 buffer, followed by centrifugation at 4 °C for 2 min at 3500 g. The supernatant containing the cytoplasmic fraction was collected, and the pellet was dislodged with a pipette tip, resuspended in Tris/SDS solution (50 mM Tris-HCl, pH6.8 + 2% SDS) supplemented by protease inhibitors and vortexed well. 4x Laemmli buffer was added to both chromatin and non-chromatin bound fractions, chromatin was fragmented by manual disruption and heated for 15 min at 95 °C. 1/4th of an embryo was loaded per well.

For testing cytoplasmic proteins in early cleavage stage embryos, embryos were resuspended in RIPA buffer (50 mM Tris-HCl, pH 7.4, 1% NP40, 0.25% Na-deoxycholate, 150 mM NaCl, 1 mM EDTA, 0.1% SDS, 0.5 mM DTT, 5 mM Na-butyrate) supplemented with protease inhibitors and spun down at 14,000 g for 10 min at 4 °C. The supernatant containing the cytosolic fraction was collected, and 4x Laemmli buffer was added. The samples were manually disrupted and vortexed well, followed by heating at 95 °C for 15min. 1/4th of an embryo was loaded per well.

SDS-PAGE was performed on 4–20% gradient gels from (Bio-Rad, Mini-PROTEAN TGX #4561096) followed by transfer on PVDF membranes and blocking in 5% BSA in TBS. Depending on the experiment, antibodies were used at 1:1000 dilutions and were incubated overnight at 4 °C (see Table S4 for antibody details). Membranes were imaged using the Licor Odyssey CLx system, and image analysis was performed using ImageJ.

### Histone tail mass spectrometry and data analysis

Pre-ZGA (256-cell stage i.e., 4.5 hpf) and mid-ZGA (4000-cell stage i.e., 6.5 hpf) embryos were prepared using the chromatin-bound protein sample preparation method described above ($N = 150$ embryos per time point, $n = 3$). An equivalent of 50x embryos was loaded per well on a 4–20% gradient gel for SDS-PAGE to resolve histones. Gels were stained using a Coomassie R-250 staining solution (0.2% Coomassie R-250 (w/v), 30% ethanol, 10% glacial acetic acid) at room temperature for 30–40 min. Gels were then destained (30% ethanol, 10% glacial acetic acid) at room temperature every few hours for approximately 24 h until distinct histone bands were achieved against a transparent background. Gels were excised between the 10–20 kDa range, and in-gel sample preparation was performed for mass spectrometry.

Gel pieces containing histones were washed with 100 mM ammonium bicarbonate, dehydrated with acetonitrile, chemically propionylated with propionic anhydride and digested overnight with trypsin. Tryptic peptides were extracted sequentially with 70% acetonitrile/0.25% TFA and acetonitrile, filtered using C8-StageTips, vacuum concentrated and reconstituted in 15 μl of 0.1% FA.

For LC-MS/MS purposes, desalted peptides were injected in an Ultimate 3000 RSLCnano system (Thermo) and separated in a 25 cm

analytical column (75 μm ID, 1.6 μm C18, IonOptics) with a 50 min gradient from 2 to 37% acetonitrile in 0.1% formic acid. The effluent from the HPLC was directly electrosprayed into an Orbitrap Exploris 480 HF (Thermo) operated in data-dependent mode to automatically switch between full scan MS and MS/MS acquisition with the following parameters: survey full scan MS spectra (from m/z 250–1200) were acquired with resolution $R = 60{,}000$ at m/z 400 (AGC target of $3 \times 10^6$). The 15 most intense peptide ions with charge states between 2 and 6 were sequentially isolated to a target value of $2 \times 10^5$ ($R = 15{,}000$) and fragmented at 27% normalized collision energy. Typical mass spectrometric conditions were: spray voltage, 1.5 kV; no sheath and auxiliary gas flow; heated capillary temperature, 250 °C; ion selection threshold, 33.000 counts.

Data analysis was performed with the Skyline (version 21.2) by using doubly and triply charged peptide masses for extracted ion chromatograms. Automatic selection of peaks was manually curated based on the relative retention times and fragmentation spectra with results from Proteome Discoverer 1.4. Integrated peak values were exported for further calculations. The relative abundance of an observed modified peptide was calculated as a percentage of the overall peptide.

## Immunofluorescence and image analysis

Immunofluorescence and confocal imaging of nascent transcripts in whole-mount embryos was performed as previously described[17,43] with minor modifications. Embryos were fixed at mid-ZGA, stained for 5-EU incorporation and/or histone marks, cleared, and imaged by confocal microscopy. Nuclear segmentation was performed using a custom-trained Cellpose model applied to the DNA channel. Mean nuclear intensities were extracted for downstream quantification. Detailed experimental and imaging parameters and computational workflow are provided in the Supplementary Methods.

## MBD-seq sample collection, library preparation and sequencing

250x pre-ZGA (256-cell stage, i.e., 4.5 hpf) embryos were collected per replicate and prepared using the chromatin-bound protein sample preparation method as described above. Methylated DNA capture was performed using the Methylated DNA Capture kit (Diagenode: #C02020010) following the manufacturer's instructions. The experimental protocol, library preparation and sequencing parameters are explained in detail in the Supplementary Methods.

## MBD-seq and ChIP-seq data processing

Sequencing reads were aligned to the *Xenopus laevis* genome (v10.1) with bowtie (v4.2). The alignment process excluded discordant and mixed reads, with paired-end reads mapped to the reference genome. Aligned reads were converted to BAM format and filtered to retain only properly paired alignments. HOMER (v4.11) was used for peak identification against corresponding input controls. Broad peaks were called using histone-style parameters, while narrow peaks were called with default settings. Identified peaks were converted to BED format for downstream analysis. Unless stated otherwise, all analysis on both raw signal intensities and peak calling data was performed on gene promoters. This was done by quantifying the signal intensities, or called peaks, in a ±1kb window around the TSS of each gene.

## CATaDa sample collection, library preparation and sequencing

Chromatin accessibility of pre-ZGA embryos was profiled using CATaDa (Chromatin Accessibility Profiling Using Targeted DamID) as described previously[36]. Embryos were injected at the 1-cell stage with 2.3 ng pRN3P_AID-Dam-only construct (a kind gift by Maria-Elena Torres-Padilla; Addgene plasmid # 136065)[73]. 500x injected or non-injected control 256-cell stage embryos were collected per replicate and prepared using the chromatin-bound protein sample preparation method described above. Pellets were resuspended in 400 μl gDNA

Tissue Lysis Buffer from the Monarch Genomic DNA Purification Kit (#T3010S); instructions for lysis of animal tissue and genomic DNA binding steps were followed as described. Elution was performed with 100 μl preheated gDNA Elution Buffer. Sample volumes were decreased to 15 μl using a SpeedVac instrument for 25 min.

All samples were processed for CATaDa as described previously[74]. Extracted genomic DNA was digested overnight at 37 °C with DpnI (NEB, R0176S), and column-purified with the QIAquick PCR Purification Kit (Qiagen, 28104). Adapters were blunt-end ligated to DpnI-digested fragments using T4 DNA ligase (NEB, M0202S; 2 h at 16 °C, heat inactivation at 65 °C for 20 min). Fragments were digested with DpnII (NEB, R0543S), purified with a 1:1.5 ratio of Seramag beads (Fisher Scientific, 65152105050250), and PCR amplified using DreamTaq HS DNA polymerase (Thermo Fisher Scientific, EP1703). PCR product was column-purified with the QIAquick PCR Purification Kit (Qiagen, 28104), sonicated and digested with AlwI (NEB, R0513S) before sequencing library preparation. CATaDa fragments were prepared for Illumina sequencing according to a modified TruSeq protocol[74]. Sequencing was performed as paired-end 50 bp reads by the CRUK Genomics Core Sequencing facility on a NovaSeq 6000.

## CATaDa data processing

Analysis of fastq files from CATaDa experiments was performed with the damidseq pipeline script[75]. Preprocessed fastq-files were mapped to the Xl.v10.1 genome assembly using bowtie2, and mapped reads were binned into fragments delineated by 5′-GATC-3′ motifs. Files were converted to the bigwig file format with bedGraphToBigWig (v4) for visualization with the Integrative Genomics Viewer (v2.13.0). MACS (v3.0.1)[76] was used to call broad peaks on the mapped read (.bam) files generated by the damidseq_pipeline.

## Signal intensity calculation for ChIP-seq, MBD-seq, and CATaDa

The resulting coverages from the pre-processing explained above were log2-transformed after adding a pseudo-count of 1. Final signal intensities for Meta plots and Heatmaps were calculated by adding up the coverage of all replicates.

Meta plots (line plots)—Aggregate line plots, depicting the mean signal intensity of gene sets of interest with bands representing standard error of the mean, were generated using ggplot2 (v3.4.4).

Heatmaps—Heatmaps were generated using the EnrichedHeatmap package (v1.32.0) in R. Each row represents the signal intensity of one gene.

Violin plots—Violin plots show the enrichment calculated with the enriched_score function of the EnrichedHeatmap package within the promoter regions of the genes. Promoter regions were defined as ±1 kb windows centered on the TSS. The violins are scaled to the same maximum width to improve readability of the density distributions. A one-sided Wilcoxon rank-sum test was used to calculate statistical significance. Four stars (****) denote a *p*-value smaller or equal than 0.0001. Three stars (***) denote a *p*-value smaller than or equal to 0.001. Two stars (**) denote a *p*-value smaller than or equal to 0.01. One star (*) denotes a *p*-value smaller than or equal to 0.05. *P*-values larger than 0.05 are denoted with n.s.

## Statistical testing for association of gene groups

To quantify the significance of the association between multiple gene lists, such as H3K4me3 dynamics groups, RNA dynamics groups and ZGA timing groups, Fisher's exact test was used. The results are visualized in Balloon plots. Each dot represents the result of a Fisher's exact test on a $2 \times 2$ contingency table. The table is created by taking one group of the first list of groups of interest and merging all other groups of the list into a comparison group. The same is repeated for the second list of groups of interest. The one-sided Fisher's test is performed for the positive association of the two gene lists of interest.

The dot size and color show the FDR-adjusted *p*-value to account for multiple testing.

## Gene ontology analysis

GO enrichment was calculated with the clusterProfiler package (v4.10.0) in R. The visualization of the similarities of the enriched GO terms (Benjamini-Hochberg adjusted $p < 0.01$) was done with the simplifyEnrichment package (v1.12.0).

## mRNA production

mRNAs for all overexpression, AID, and rescue experiments were produced by in vitro transcription from linearized plasmids (full cloning and transcription details in Supplementary Methods).

## Auxin experimental setup

For the "treated" condition, embryos were co-injected at the 1-cell stage with 11 ng Kdm5b[wt], 345 pg H3.3[K4M] and 5 ng TIR1 mRNA. "Control" condition embryos were co-injected at the 1-cell stage with 4 ng Kdm5b[ci], 345 pg H3.3[wt] and 5 ng TIR1 mRNA. mRNA amounts were determined on the basis of equal expression levels of the over-expressed proteins in pre-ZGA stage embryos. For "window" depletion treatment, embryos were grown until the pre-ZGA stage (4.5 hpf/256-cell stage) and transferred to an auxin solution (500 µM auxin in 0.1 × MMR) to develop further. "Persistent" depletion treatment embryos were grown in 0.1x MMR throughout development. To validate protein expression, pre-ZGA embryos were collected for Western blotting. To validate auxin-induced protein degradation, embryos were collected at the mid-ZGA stage (6.5 hpf/4k-cell stage) and collected for Western blot and immunofluorescence. To assess the extent of H3K4me3 depletion, embryos were collected either at the pre-ZGA stage for chromatin-bound protein isolation and Western blot, or at NF12 for Western blot.

## Morpholino-mediated knockdown experimental setup

Custom antisense morpholinos (asMOs) targeting the translation start sites of both L and S homeologs of *Xenopus laevis kmt2b* and *cxxc1* were designed with the help of Gene Tools, LLC. The recommended standard control morpholino from Gene Tools LLC was used as a control. The morpholino sequences are shown in Supplementary Table 3.

1-cell stage embryos were injected with 9.2 nl of 1 mM morpholino solution and developed in 0.5x MMR solution for 2 h before moving to 0.1x MMR. Embryos were grown at 18 °C until the desired stage. For analysis of developmental phenotypes, embryos were imaged and counted at periodic intervals with daily replacement of 0.1x MMR.

## CUT&RUN

CUT&RUN was performed as described previously[47,77] with minor modifications. 45 whole embryos were collected per sample at mid-ZGA stage (6.5 hpf). The complete CUT&RUN experimental protocol, conditions, library preparation and sequencing parameters are provided in detail in the Supplementary Methods.

## CUT&RUN data processing and visualization

Sequencing reads were aligned to the *Xenopus laevis* genome (v10.1) and a spike-in genome (Escherichia coli, GCA_001606525) using Bowtie2 (v2.4). The alignment process excluded discordant and mixed reads, with paired-end reads mapped to the reference genome. Aligned reads were converted to BAM format and filtered to retain only properly paired alignments. Normalization scaling factors were computed using deepTools (v3.5.3) based on the read distribution across genomic regions. HOMER (v4.11) was used for peak identification. Narrow peaks were called using histone-style parameters and

converted to BED format for downstream analysis. Differential peak analysis for H3K4me3 signal was performed on a consensus peak set defined based on narrow H3K4me3 peaks called in at least two out of three independent biological replicates from wildtype embryos, using the findOverlaps function from GenomicRanges (v1.58.0) with maxgap = 150 bp.

Meta plots (line plots) depicting the mean signal intensity of gene sets of interest with bands representing the standard error of the mean were generated using normalizeToMatrix from EnrichedHeatmap (v1.36.0) and *E. coli* spike-in normalized signal. For statistical comparison of H3K4me3 signal across gene sets of interest, the Kolmogorov-Smirnov test was used.

## RNA-seq experimental setup

For RNA-sequencing experiments, independent experiments were performed on two different frogs, serving as two biological replicates. Within each biological replicate, three sets of three embryos each were collected per time point, corresponding to three replicates within each independent experiment. Embryos were collected as described at hourly intervals after the pre-ZGA stage (i.e., 4.5 hpf/256-cell stage): at 5.5 hpf, 6.5 hpf and 7.5 hpf. All liquid was removed from the tube, and samples were snap-frozen on dry ice and then stored at −80 °C until further processing. Gene expression was analyzed for morpholino knockdown experiments and H3K4me3 window-depletion experiments.

## Total RNA extraction and ribosomal depletion

Three embryos were collected per sample and stored at −80 °C. Total RNA was isolated using the RNeasy kit (Qiagen, #74106) according to the manufacturer's instructions and ribosomal RNA was depleted using custom-made oligomer mixes for *Xenopus laevis* rRNA as described previously[78] with minor modifications. Complete protocols are provided in the Supplementary Methods.

## RNA-seq library preparation and sequencing

Sequencing libraries were generated using the NEBNext Ultra II Directional RNA Library Preparation Kit for Illumina (NEB, #E7760) as per the manufacturer's instructions using 12–13 PCR amplification cycles. The quality of cDNA libraries was assessed using the Agilent High Sensitivity D5000 ScreenTape System (Agilent, #5067-5592) in the Agilent 4150 TapeStation System or using the Agilent DNF-474 HS NGS Fragment Kit (Agilent, DNF-474-1000) in the Fragment Analyzer System (Agilent). Libraries were multiplexed, and sequenced was performed as paired-end 50 bp reads on the Illumina NovaSeqX+ by the Helmholtz Core Facility Genomics (CF-GEN).

## RNA-seq data processing

Paired sequencing reads were processed using Kallisto (v0.48) for pseudoalignment and quantification of transcript abundance. Transcript and annotation files were downloaded from Xenbase (v10.1) for *Xenopus laevis* (transcripts and annotation). After quantification, transcript-level abundances were imported using "tximport" (v1.32.0) and converted to a "SummarizedExperiment" (v1.34.0) object in R (v4.3.2). Transcript isoforms are summed up for the respective gene count. Datasets of two independent batches were subjected to differential expression analysis on each of their three respective replicates performed with DESeq2 (v1.42.1). Reported *p*-values are FDR adjusted for multiple testing. Transcript abundances were normalized to transcripts per million and log2-transformed with a pseudo count of 1 for further analysis.

## Reporting summary

Further information on research design is available in the Nature Portfolio Reporting Summary linked to this article.

## Data availability

Original data produced in the experiments have been deposited on GEO. The MBD-seq data are available under the accession number GSE286525. The CATaDa data is available under the accession number GSE286524. The RNA-seq data from the auxin experiment are available under the accession number GSE286887. The RNA-seq data from the morpholino treatments are available under the accession number GSE286886. The CUT&RUN data is available under accession number GSE303651. The mass spectrometry proteomics data have been deposited to the ProteomeXchange Consortium via the PRIDE[79] partner repository with the dataset identifier PXD058937. Publicly available data were obtained from GEO with the following accession numbers: GSE125982, GSE75164, GSE76059, GSE201835. All other relevant source data supporting the key findings of this study are provided in this paper. Source data are provided with this paper.

## Code availability

All original code can be accessed at Zenodo (https://doi.org/10.5281/zenodo.17667122) or GitHub (https://github.com/ScialdoneLab/h3k4me3_maintenance)[80]. The preprocessed data for the analysis can be downloaded from Zenodo (https://doi.org/10.5281/zenodo.14648394).

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

## Acknowledgements

M.S.O. and E.H. were supported by the HO 6864/2-1 project grant and the Chromatin Dynamics CRC1064, Project-ID 213249687, both DFG; Recognition Award of the Helmholtz Association; Project grant of the MRC MR/P00479/1; S.H. by ERC starting grant 852798. Work in the labs of E.H. and A.S. was funded by the Helmholtz Association. A.E. received support from the Helmholtz Association, awarded to Maria-Elena Torres-Padilla. M.S. was supported by the "Joachim Herz Stiftung" fellowship and the "Munich School for Data Science—MUDS". JvdA is supported by a Wellcome Clinical Research Career Development Fellowship (219615/Z/19/Z), Wellcome Discovery Award (226653/Z/22/Z), a UKRI BBSRC Grant (BB/X00256X/1) and Medical Research Council (MC_UU_00028/

8). The authors are grateful to the members of the CAM LMU Munich and G. Eckstein and I. de la Rosa Velazquez from the Helmholtz Genomics Core Facility (CF-GEN) for technical support. We thank all present and past lab and institute members, especially Nemanja Vasovic, as well as Ralph Rupp and Kikue Tachibana, for reagents, their critical input and support. For the purpose of open access, the author has applied a Creative Commons Attribution (CC BY) license to any Author Accepted Manuscript version arising from this submission.

## Author contributions

M.S.O. performed experimental work with assistance from M.M., A.J. analyzed the CUT&RUN experiments, I.F. and A.I. performed and analyzed the mass spectrometry experiments, A.H.M. and J.v.d.A performed and analyzed the CATaDa experiments, and A.E. provided expertise in confocal imaging. S.H. provided guidance on experimental design through scientific discussions. M.S., T.S. and M.S.O. performed computational analyses with input from A.S. and E.H. The manuscript was written by M.S.O. and E.H. with input from A.S. and M.S., and E.H. designed and supervised the study.

## Funding

## Competing interests

The authors declare no competing interests.
