## [Peer Review file · Nature Communications]

Pre-marking chromatin with H3K4 methylation is required for accurate zygotic genome activation and development

Corresponding Author: Dr Eva Hörmanseder

Version 0:

Reviewer comments:

Reviewer #1

(Remarks to the Author)

Oak & Stock et al present evidence that gamete chromatin status is transmitted to the *Xenopus laevis* embryo to influence which genes are expressed during embryonic genome activation -- specifically, H3K4me3 at promoters, which is associated with DNA hypomethylation and high GC content. Their results provide an important perspective on how genome activation specificity may be achieved. However, as appealing as the authors' model may be, the data do not definitively support their claim (or, strong implication) that promoter H3K4me3 is maintained continuously from gametogenesis through to the blastula. The pre-ZGA stage (256 cell) is comparatively quite late -- based on other studies, we would indeed expect that histone marks do emerge in the stages prior to ZGA comparable to 256-cell. What about earlier cleavage stages? Apart from a very weak Western signal at 32cell, the authors do not present any data that would indicate there could be early promoter marks. So, still consistent with the model in other animals that histone marks are wiped in early embryos, then re-emerge later. Even if H3K4me3 is detected in very early embryo, the H3K4me3 chromatin patterns may look very different than in sperm or blastula, e.g. analogous to the mouse broad domains in the zygote that only later focus on promoters. Thus, in this reviewer's opinion, the language and framing of the manuscript need to be moderated accordingly, to avoid over-reaching.

There are also potentially ambiguities with the bioinformatics analyses that affect the interpretation of the scale of H3K4me3 recapitulation from gamete to embryo. Some analyses were performed on peaks while others on gene promoters -- and, it is additionally unclear whether multiple transcript isoforms for the same genes were treated individually (if so, that could lead to double-counting the same genomic locus). As much as possible, the authors should unify their analyses on genes (one transcript isoform per gene for the chromatin/epigenetics, pooled transcripts for the RNA-seq).

The authors should strive to better relate their findings to other work, especially in *Xenopus*, especially if their results/interpretations disagree. For example, in *X. tropicalis*, Bogdanovic et al 2011 (which is not cited) already found that TSSs are largely DNA hypomethylated in the blastula/gastrula (are their proportions of hyper/hypomethylated promoters similar to the current manuscript's?). And Hontelez et al 2015 also found that transcriptionally inhibited embryos still gain H3K4me3.

Finally, to enhance accessibility of the manuscript, the authors could consider curating their main figure panels, which are numerous and perhaps not all equally contributing to the narrative.

Specific points:

- Figure panel placement/ordering is a bit irregular -- following the grid would improve readability
- Fig 1b, the color scale is not intuitive to this reader. What does "max log scaled per peptide" mean? Is this simply low to high across the entire dataset? Also, is % abundance relative to unmodified?
- Fig 1b, should "H3K9me1" in row 4 be "H3K4me1"?
- Fig 1d, were equal numbers of embryos used for the different stages? The H4 controls seem to indicate this. There is an interpretation challenge then. Ideally, H3K4me3 intensity should be quantified relative to H3 (not H4) over replicates to facilitate comparison between stages
- Fig 1f, g -- the micrographs suggest that H3K4me3 loss in α -Amanitin is binary, where some nuclei have strong signal while others are completely absent (compared to the H4 panel, which presumably labels all the nuclei). This pattern does

not seem to be reflected in the distributions in panel g. The authors should clarify and also overlay the H4 and H3K4me3 staining to better appreciate the variability between nuclei.

- Fig 1i, j -- ideally these should have the same sort order and be clustered. It would be valuable to see the actual signal intensities grouped similarly to Fig 1k, especially since the relative proportions of, e.g. hypo vs hypermethylated promoters don't obviously correspond between the panels
- Fig 1j, k -- the lack of correspondence between H3K4me3 and promoter accessibility is highly unusual. Are there mappability biases with the CATaDa (e.g., presence of a GATC) that could lead to under estimation of accessibility? Are the accessibility profiles consistent with later stage ATAC-seq? Do the H3K4me3 non accessible promoters gain accessibility later?
- Fig 2b, the legend says "promoter peaks" but is this just counting promoters? As opposed to peaks in promoters?
- Fig 2e (also 3d), are there equal numbers of TSSs across each of the four panel sets? The violin plots for Pre-ZGA and ZGA look like they have far less density than for sperm. If not, why would the number of genes change between stages? Also, heatmaps similar to Fig 1i,j would be valuable here as they would show the variability within groups and between stages relative to the TSS position
- Fig 3b,c the counts here seem quite low for genes activated during ZGA, and even lower than what we understand to be the size of the maternal contribution. Moreover, the total is much lower than the gene total in Fig 2b.
- Fig 3 generally combines sperm and oocyte/egg for the GS and GZ categories, but they probably should be considered separately. For example, oocyte-specific genes are included in the GS curves for 3d, which is likely lowering the average in the sperm H3K4me3 graphs. And in general, genes represented in the maternal contribution are an important category distinct from spermatid-active genes, since maternal but not (much?) paternal mRNA is inherited in the embryo.
- Fig 4, Since the Window and Persistent treatments are extremely similar, there is some concern that the AID degradation would be too slow to specifically implicate pre-ZGA H3K4me3 loss as causal for the gene expression defects. It could be that ZGA proper is still being impacted, while H3K4me3 levels slowly recover. If the authors' model is maintenance throughout early embryogenesis, wouldn't disrupting H3K4me3 at only the very early stages be sufficient to abolish propagation of the gamete patterns? The lack of ChIP-seq profiles after recovery to compare patterns to untreated strongly diminishes the conclusions we can make here.
- Wouldn't increasing H3 levels affect the nuclear-cytoplasmic ratio and potentially delay ZGA? Are the control embryos activation profiles strongly different from untreated (or mock injected)?
- Fig 4c, H3 needs to be shown also
- Fig 4g, the GS and ZS colors are too similar to distinguish
- Fig 4k (and 6h), a log scale should be used for fold change, otherwise equivalent magnitude increases/decreases will not have the same color intensities. Also, comparison with normal zygotic activation levels (wt vs alpha amanitin?) would be helpful to compare the magnitude of the effect.
- Fig 4i/Lines 404-405, sox3 is more strongly maternal, so that would potentially account for not being able to detect a change.
- Fig 5, are the categories independent of activation status? According to Fig 4k, the vast majority of activated genes are in the Gained category. But the analyses here suggest the Kept group is once again large. Additionally, only a subset of Kept loci have accessible chromatin according to Fig 1. So, it seems like a more stratified analysis would be informative here that also takes gene activation into account (and perhaps sensitivity to H3K4me3 loss as determined in Fig 4). Otherwise, Fig 5 seems out of place in the narrative
- Fig 6, what is the overlap with the H3K4me3 affected genes from Fig 4?
- Fig 6b, is H3K4me3 reduced at earlier stages e.g. 256cell?
- Fig 6d, the volcano plots look odd -- particularly the 7.5hpf timepoints, where there seem to be two distinct groups of downregulated genes (mostly blue and mostly orange). Please verify that the plots are correct
- Throughout, note that two-tailed (sided) hypothesis tests should be used, rather than one-tailed, when changes can be in either direction
- Throughout, please check the Western kDa size labels. E.g., Fig 4b shows HA-H3.3 is >55 kDa, Fig 6b shows H3K4me3 is <15 kDa
- Line 134, "modifications on globally transcriptionally silent chromatin" -- this is a presumption, the methods suggest that we can only know the histones are nuclear as opposed to explicitly DNA bound (chromatin). If this is not the case, please clarify
- Line 154, "small number of microRNAs" grossly understates the scale and impact that miRNA activation has during early ZGA. In zebrafish, the high copy number miR-430 cluster spans a hundred kilobases (Hadzhiev Dev Cell 2023) and has a high density of marked chromatin (Chan Dev Cell 2019). The Xenopus miR-427 cluster is similarly large, per the Owens paper.
- Line 283, the analysis doesn't really address whether genes are continuously marked, given the lack of early-stage and oocyte/egg data
- Line 304-307, the authors could evaluate this claim with available public data
- Line 420, no functional link has been established between ZGA and "maintenance of H3K4 methylation *from the gametes to the embryo**"
- Line 510, "we show that these H3K4me3 reader-writer enzymes acting independently of transcription..." -- I don't think the authors have demonstrated that these enzymes are acting independent of transcription for Xenopus genome activation. This statement should be tempered

(Remarks on code availability)

Reviewer #2

(Remarks to the Author)

The manuscript by Oak & Stock et al. aims to provide evidence for the faithful propagation of epigenetic memory of active chromatin states via H3K4 methylation in *Xenopus* in the absence of transcription or instructive transcription factors. What it does achieve is to conclusively support that, as in zebrafish, there is indeed H3K4me before ZGA and that it is maintained from gametes to embryos across a transcriptionally silent window. This finding contradicts earlier claims by the Veenstra lab that H3K4me₃ is acquired around ZGA. The ms goes on to characterize and segment H3K4me bound loci using previously published datasets. They also manipulate H3K4me levels using various tools/antisense morpholinos and monitor differentially expressed genes (DEGs) and embryonic development. However, these results are at best suggestive, but not conclusive. With regard to the physiological relevance, the results are inconclusive because *Kmt2b* and *Cxxc1* Morpholino-induced deficiencies persist into later developmental stages that compromises attributing phenotype to pre-ZGA stage effects. Also with regard to the main question, does H3K4me provide epigenetic memory in the absence of INSTRUCTIVE TFs, the results are only suggestive, not conclusive, as such factors cannot be ruled out. Moreover, while some new pre-ZGA H3K4me datasets are established, the study is to a large extent relying on reanalysis of published data. So, do the conclusive insights and new data sets of this study warrant a Nat Commun paper? I think it is borderline. In *Xenopus*, the Gurdon lab already provided good evidence for a memory role of H3K4me. This includes PMID 18066050 (not cited), which already showed epigenetic memory of an active gene state depends on histone H3.3 incorporation into chromatin in the absence of transcription. In Fish, H3K4me is also instructive for later developmental gene expression PMID: 22137762. Given these caveats and the many experimental shortcomings (see below) the ms is probably a REJECT at this point.

Main points:

1. Fig1b: "We detected a wide range of active and repressive histone modifications on globally transcriptionally silent chromatin of pre-ZGA embryos (256-cell stage). These include the active marks H3K4me_{1/2/3}." H3K4me seems absent in ZGA. Also: LC MS/MS quantification without isotopic standards is at best semi-quantitative. Bottom line: The mass-spec data add little to the main ms, even contradict it re H3K4me levels, but the other histone mark profiles may be interesting for the community and may be difficult to publish otherwise. So one may keep them in.
2. Furthermore, the color-coding for the percent abundance does not match the scale bar for some of the modifications (e.g. H3K9me₃ and H3K27me₃ to name a few). It is not clearly stated in the text whether H3K4me₃ was followed up due to pre-ZGA level and mid-ZGA level differences. All subsequent results impinge on this experiment, so a clear outcome and inference is important. Fig.1e is clearer in this regard, so the Fig.1b could be moved to the supplement.
3. Fig1c: Y axis is "% occurrence", i.e. linear scale; Fig. 1b says "log scaled", i.e. translating into orders of magnitude differences that are not reflected in 1c. Confusing.
4. Line 144: "these results reveal for the first time that the highly dynamic chromatin of rapidly dividing pre-ZGA embryos is decorated by both repressive and active chromatin marks." Early *Xenopus* embryos have large amounts of free histone proteins unbound to DNA that may contaminate the nuclear fractions. In *Drosophila*, PMID: 30639105 concluded that most nuclear histone H3 is not DNA-bound in the early cleavage cycles. How do you know that spurious amounts of pre-ZGA H3K4me₃ detected by LC MS/MS are DNA-bound and not free modified histones?
5. #148 "H3K4me₃...were detectable on mostly transcriptionally silent chromatin". How do you know on what TYPE of chromatin, if any, H3K4me₃ locates at this point of the ms?
6. Fig.1f-g: What about Pol I inhibition?
7. Fig. 1i-k: what is the nature of the genes with prominent H3K4me₃? Those that are induced at ZGA? Protein coding? Others ?
8. #197 " this shows that in *Xenopus* embryos at stages that precede ZGA, H3K4me₃-marked promoters of transcriptionally silent genes". How do we know these loci are silent? Even if there were silent (i.e. H3K4me₃ deposition is Amanitin-insensitive, which would be nice to show), what is the Pol II binding profile at this stage? Is there paused / stalled /other Pol II at these same loci?
9. Fig.S1c. pre ZGA H3K4me goes against Veenstra. How many H3K4me peaks were obtained? How does this number compare to the number of post ZGA peaks?
10. Fig. 3 does not add much beyond Fig 2, as highlighted in the repetitive conclusion (#342-345). Can go to supplement.
11. Fig.3d: The authors do not comment on the gamete specific group of genes that they identify from the transcriptome analysis and the H3K4me₃ signatures associated with promoters of GS group of genes at the spermatid and sperm time points. There is no explanation/speculation as to why H3K4me₃ peak intensity is low for gamete specific genes even in gametes as well as the reason for GZ H3K4me₃ peak being higher than ZS H3K4me₃ peak during pre-ZGA and ZGA time points.
12. Fig. 4 Very elegant approach: *Kdm5bci* and H3.3wt as the control, were co-injected with TIR1 – which degrades AID-fused proteins upon addition of auxin – into embryos at the one-cell stage; why not use this approach for phenotypic analysis? Too weak effects?
13. Fig4c: "Treated" and "Control" labels seem to be swapped.
14. #391 "Mean expression levels of the KEPT group of genes were significantly reduced" is not visible in 4i, unlike for GAINED group. It even says "n.s." at 6.5hpf. To the contrary, the mean expression is even increased in persistent treated at 6.5hpf and at 7.5hpf (4j). In fact, only a handful of genes seems to be affected in 4k. Could this be due to rather moderate effect of *Kdm5b*/H3 o.e.?
15. Fig.4l and Fig.6i: The effect of maintenance of H3K4me₃ and its role in regulating the expression of *sox3* as a ZGA transcription factor is not convincing.
16. Fig. 6: *kmt2b* and *cxxc1* Morpholino validation? Rescue with non-targeted wt construct? Two independent MOs?
17. 6b: WB should quantified and normalized to H4. Likely, *cxxc1* Mo achieves not even 50% reduction of H3K4me₃ compared to Ctrl Mo. *kmt2b* Mo looks more convincing.
18. #491: " Additionally, we observed a significant overlap of downregulated zygotic genes in individual *Cxxc1* and *Kmt2b* knockdown conditions" Shown where? Also, what is the overlap between DEGs with the analogous experiment in Fig. 4?

19. #500: "We then addressed whether the observed depletion of Kmt2b and Cxxc1 in embryos was affecting embryonic development. Both Cxxc1 and Kmt2b knockdown embryos showed a delay in development." Phenotype interpretation is compromised by the Mos having effects into later development. Should be mentioned. The Tir1 approach with Auxin degradation should not suffer from this problem and may be more interesting to study. Same for #541: "we can conclude that loss of H3K4 methylation results in defective gene activation and embryonic death."

20. # 542: "our data suggests that H3K4me3 maintenance occurs independently of transcription and of the signal that initially induced the state during gamete maturation." Pioneer maternal TFs cannot be excluded to program H3K4me3 sites during each cell cycle.

Minor points:

1. Text wise the Results is not an easy read, multiple, complex figures are lumped together in the Results section for the reader to make of them, without being helped to interpret them individually (e.g. #281,, 221, 271, 273, 389, 409). If their messages is no different, why show all the panels without text-wise distinguishing them?
2. The figure legends do not provide complete information, which would make the figure self-explanatory.
3. Fig.1d: H3K4me3 seems to be detected at 256 cell stage and not 32 cell stage.
4. Fig.1f: The authors check H3K4me3 and H4 staining in mid-ZGA embryos but point out (lines 154-156) that some transcription occurs in pre-ZGA stages in other species. The staining should likewise be performed in pre-ZGA embryos to test the hypothesis that early transcription reinforces H3K4 promoter methylation. This is especially required because the genomic context of H3K4me3 occurrence is later explored in pre-ZGA embryos
5. Fig.2e and 2f: The "gained" and "lost" group nomenclature is rather unclear as to when exactly the gain and loss of signal takes place. There are three stages at which loss can take place and two stages at which gain can take place. However, there is one line that corresponds to the gain and loss groups each in the graphs. Some clarification in the legend or text might be helpful.
6. Fig.S3: It is unclear whether the network with the central "mRNA processing" node belongs to the GZ genes (Fig.S3b) or GS genes (Fig.S3d), mentioning this in the legend would be helpful.
7. Fig.3h: It is important to check the expression of GS group of genes at various time points monitored, as a negative control. It should be mentioned in the legend that the heat map only includes GZ and ZS group of genes.
8. Fig.4d, 4e: The reduction in fluorescent signal of H3K4me3 in treated persistent samples cannot be attributed only to overexpression of Kdm5bwt and H3.3K4M mRNAs as the H4 signal that should otherwise remain constant also seems to be reduced (lines 374-375). The description of this observation is also a bit vague in the main text without alluding to the specific panels. Also, explain "persistent"
9. Fig.4 and Fig.S4: The results of experiments shown correspond to two biological replicates that are not in agreement. The interpretations in the text however describe one replicate over the other, therefore the claims cannot be generalized. E.g. expression of KEPT group genes is significantly reduced only in treated embryos at 7.5hpf and not 6.5hpf as is claimed in lines 391-392 as seen in Fig.4i for one biological replicate. The disagreement between biological replicates is also observed in the number of DEGs identified upon Cxxc1 and Kmt2b in Fig.S6f.
10. Fig5: Figs 2-3 deal with metanalysis, Fig 4 introduces Kdm5b: why return to metanalysis in this figure after the Kdm5b chapter? The figure does not even recur to the Fig 4 Kdm5b -DEGs. The logic of the ms Figure flow escapes me. Figure can also go to supplement.
11. Lines 415-417 should be corrected towards the end of the sentence.
12. Lines 430-432, Fig.S5a only depicts lower DNA methylation at promoters of KEPT group of genes in gametes but not higher CG content and increased DNA accessibility

(Remarks on code availability)

Version 1:

Reviewer comments:

Reviewer #1

(Remarks to the Author)

In their revised manuscript, Oak & Stock et al have clarified many of the points raised previously, generally to the improvement of the manuscript. At this point, I do not think any additional experiments are warranted. However, I still feel that the framing of the narrative as illustrating epigenetic memory is too strong relative to the results. Additionally, there are points where a more objective acknowledgement of the actual results would be desirable, as well as more rigorous statistical approaches, as described below.

FRAMING:

- The title is claiming a phenomenon of epigenetic memory (though rephrased as "transcriptional memory") despite the interpretation caveats
- The use of the terms "retained" versus "maintained" throughout the manuscript doesn't really change the implication of continuous marking. In the plausible scenario where K4me3 is passively lost during early cleavage stages but then reappears at Stage 7-8 at the same locations, one would not call that either "retained" or "maintained" because neither of those terms acknowledges the dynamics. Similarly, the categories "KEPT," "GAINED," and "LOST" strongly imply continuity; alternative terms such as "SHARED," "EMBRYO-SPECIFIC," and "GAMETE-SPECIFIC" would be more objective and appropriate.
- The similarity of 32-cell and 256-cell states of H3K4me3/H4 only supports the absence of global erasure between those stages (with the caveat that the Western quantification is not very precise), not between the zygote and 32-cell. The Long et al 2023 result is difficult to interpret since their Western shows surprisingly high and equivalent H3K4me3 throughout St 1-9 (and highly reduced at St10), which would seem to contradict the current manuscript Fig 1c (if there's a free vs chromatin-bound distinction, then I don't know how the Long et al result could be used as support here).
- The authors strongly emphasize transcription independence and the transcriptionally silent early embryo, but this should also be moderated somewhat. We cannot exclude the possibility that low-level, non-specific, and/or abortive Pol II engagement are occurring prior to major ZGA, which may contribute to chromatin state, especially as the embryo approaches Stage 8/9. miR-427 activation is already detectable prior to Stage 8 in trop (Owens) and laevis (Phelps). Also, amanitin and triptolide are elongation inhibitors, which does not necessarily prevent association of transcription machinery to chromatin, and certainly would have no effect on pioneer transcription factor binding and their capacity to recruit histone modifying enzymes.

ADDITIONAL POINTS:

- The CATaDa is indeed showing a noticeable bias against GATC-poor regions in Fig 1g, suggesting that the category of "High H3K4me3 / low accessibility" may indeed be artifactual. Curiously, the authors do not seem to discuss this in the manuscript, just showing the heatmap with no explanation. Response Plot 4 shows no apparent differences in GATC promoter content between groups, but we might presume the groups are all equivalently composed of some GATC-containing and non GATC-containing promoters. Given that the proposed existence of K4me3/low accessibility promoters has little to no bearing on the rest of the manuscript, it's unclear why the authors would not acknowledge the ambiguity.
- Response Plot 5 should be included in the manuscript, as it lends support to the defined categories that is otherwise harder to appreciate from the existing figures. The comparison of signal intensity across timepoints is not a concern; the within-timepoint comparisons between groups are what is valuable here.
- Response Plot 6 when compared to Fig 3d seems to show subtly different dynamics, but in a surprising direction -- indeed, sperm genes have profiles that more closely match zygotic-specific (the curves are closer together in Response Plot 6 than in Fig 3d). But this would then indicate that egg-specific genes have stronger H3K4me3 signal in the embryo as well as in sperm, which is an odd result. I do not agree with the choice to exclude this analysis and the discussion from the manuscript.
- The H3K4me3 CUT&RUN was a valuable addition, but the Fig S5A PCA plot seems to show some poorer replicate

correlation, and the intermingling of treatment and control conditions in PCA space, calling into question the significance of the results. Heatmaps across all promoters, stratified according to gene group, would likely be more informative.

- Fig 5h, S5J, 6E appear to be versions of RNA-seq differential expression tests, but not performed in a way consistent with the field. Control vs treatment should be compared using something like DESeq, then the number of significantly differentially expressed genes for each of the KEPT and GAINED groups can be reported. The issues with the manuscript's current approach are: a) only one replicate is used for the stat test, but all replicates need to be considered; b) a Wilcoxon rank sum test is used, which again is not an appropriate test to use for a two-condition RNA-seq comparison, but even so it is an unpaired test when in fact the data are paired (you're measuring the change of the same genes between control and treated); c) a one-tailed test is used despite the possibility of changes in either direction and thus allows results to be reported as non-significant even if there is a significant change in the opposite direction (which may be the case in Fig 5h persistent-kept). The argument about testing a specific directional hypothesis, while arguably technically valid, is not in the spirit of the scientific endeavor here.

- Fig 5i and 6e likewise seem to be non-standard stat tests for RNA-seq data -- what test specifically isn't stated, and how the replicates are used is unclear.

- Fig 6b and S7b lack statistics. Inspection of the points suggest poor replicability between Western blots (indeed, Fig 6b Kmt2b MO lane seems problematic with loading), which does raise caveats about the strength of the MO effects -- stating the average % loss is probably not realistic.

- Fig S7M uses an inappropriate stat test (Fisher's) that does not take the replicates into account. Pooling the replicates can obscure large magnitude deviations in trials with smaller N, for example. Indeed, the plot shows high variability in a couple of the conditions

- Fig S1A: Please indicate the y axis is on a log scale

- Fig S6A - I'm struggling to interpret the western relative to the legend; additional labeling may help

(Remarks on code availability)

Reviewer #2

(Remarks to the Author)

I appreciate the clarifications, edits and additional experiments. However, several substantive issues remain unresolved and limit the strength of the conclusions:

1. Stage-restricted functional tests: While the auxin-inducible system is an improvement, there is still no phenotypic analysis confined strictly to the pre-ZGA window. Morpholino knockdowns persist into later stages, making it impossible to exclude indirect effects on gene expression and development.
2. Independence from maternal TFs: The study still cannot rule out that maternal pioneer factors re-establish H3K4me3. The claim of transcription-independent epigenetic memory remains suggestive.
3. Chromatin-bound validation: The authors' extraction protocol may enrich for chromatin-bound histones, but no direct biochemical validation is provided to exclude contamination by free histones. This is critical in early *Xenopus* embryos, where free histone pools are abundant.
4. Quantitative ChIP-seq comparability: The numbers of peaks pre- vs. post-ZGA are not normalized with spike-ins, so apparent "maintenance" could reflect global differences in ChIP efficiency or background rather than true retention.
5. Definition of transcriptional silence: For loci designated "silent," there is no Pol II occupancy analysis to exclude paused or engaged polymerase; α -Amanitin assays at mid-ZGA do not address the earlier cleavage stages.
6. Specificity of Kmt2b knockdown: Unlike Cxxc1, there is no rescue experiment or second MO for Kmt2b, leaving potential off-target effects unresolved.

These points are relevant to the paper's main mechanistic claims. While the data support the presence of H3K4me3 prior to ZGA and suggest it may influence genome activation, the mechanistic link to epigenetic "memory" remains insufficiently established.

(Remarks on code availability)

Reviewer #1 (Remarks to the Author)

We thank the reviewer for their assessment of our manuscript and the helpful feedback. We greatly value both the overall perspective and the specific suggestions, which have enabled us to strengthen our arguments and present our findings with improved clarity and precision. Please find our comments below:

“Oak & Stock et al present evidence that gamete chromatin status is transmitted to the *Xenopus laevis* embryo to influence which genes are expressed during embryonic genome activation – specifically, H3K4me3 at promoters, which is associated with DNA hypomethylation and high GC content. Their results provide an important perspective on how genome activation specificity may be achieved. However, as appealing as the authors’ model may be, the data do not definitively support their claim (or, strong implication) that promoter H3K4me3 is maintained continuously from gametogenesis through to the blastula.”

Thank you for highlighting this important point. Our work aims to propose mechanisms ensuring the presence of H3K4me3 at promoters prior to and during ZGA independently of transcription, as evidenced by functional disruptions causing gene expression defects at ZGA (revised manuscript: Fig. 5), and involvement of *Kmt2b* and *Cxxc1* in establishing H3K4me3 (revised manuscript: Fig. 6).

However, we do **not** claim or exclude continuous or unbroken maintenance of promoter H3K4me3 from gametogenesis through to blastula. We agree that our current Western blot data cannot rule out transient loss of chromatin H3K4me3 during a brief moment in early cleavage stages or immediately post-replication. Nonetheless, we observe that many promoters marked by H3K4me3 in spermatids remain marked in sperm, 256-cell embryos (which are transcriptionally repressed), and at ZGA. This pattern supports a model in which promoter H3K4me3 is established *before* ZGA and our functional analyses support that it plays a role in enabling gene activation, even if the mark may be subject to a brief loss during early development.

To reflect this, we now use “retained” instead of “maintained” in the revised manuscript to mean that the H3K4me3 mark is *observed at gene promoters at consecutive developmental stages* (e.g., gametes, 256-cell, ZGA), without implying nor excluding unbroken continuity or direct molecular propagation of the mark. We now also explicitly mention and discuss limitations regarding continuity of the mark.

The pre-ZGA stage (256 cell) is comparatively quite late -- based on other studies we would indeed expect that histone marks do emerge in the stages prior to ZGA comparable to 256-cell. What about earlier cleavage stages? Apart from a very weak Western signal at 32cell, the authors do not present any data that would indicate there could be early promoter marks. So, still consistent with the model in other animals that histone marks are wiped in early embryos, then re-emerge later. Even if H3K4me3 is detected in very early embryo, the H3K4me3 chromatin patterns may look very different than in sperm or blastula, e.g. analogous to the mouse broad domains in the zygote that only later focus on promoters.

Previous *Xenopus* studies suggesting erasure and re-establishment of H3K4me3 used blastula stage 8 embryos treated with transcription inhibitors as proxies for pre-ZGA (Hontelez et al., 2015) but did not directly analyze earlier stages, as we do here.

In addition, comparing pre-ZGA chromatin states directly across species is inherently difficult due to substantial developmental differences. For example, mouse ZGA occurs at the 2-cell stage after just one division post-fertilization, while *Xenopus* embryos undergo ~12 rapid cleavage cycles before ZGA. In zebrafish and *Drosophila*, histone marks such as H3K4me3 also begin accumulating before ZGA, suggesting re-emergence precedes transcriptional onset and may prime chromatin in a species-dependent manner.

In our study, we deliberately chose the 256-cell stage because it represents dynamic chromatin due to rapid cell division with transcriptional repression. After 8 cell cycles with no transcription, H3K4me3 would be diluted 256-fold if not actively (re-)established. Our data show H3K4me3 is present on chromatin already at the 32-cell stage by Western blot, with a reproducible albeit weaker signal (Figure 1c). The relative H3K4me3/H3 ratio remains similar between 32-cell and 256-cell stages, increasing only at ZGA, arguing against global erasure (Fig. S1b). Furthermore, western blotting of whole-protein extract in recent studies suggests that H3K4me3 is present in *Xenopus laevis* 1-cell zygotes and 4-cell embryos at levels comparable to those in 256-cell embryos (Long Q et al., 2023; PMID37027476). This is consistent with early establishment of H3K4me3, however, with the caveat that its association with chromatin is unclear from their analyses. We added this information in the manuscript (line 145-147).

We acknowledge that H3K4me3 chromatin patterns may be broader early on (similar to mouse zygotes), focusing on promoters later. We have now clarified these points and limitations in the revised manuscript.

Thus, in this reviewer's opinion, the language and framing of the manuscript need to be moderated accordingly, to avoid over-reaching.

In summary, we understand this criticism and corrected or rephrased potentially overreaching statements in our text. We have carefully moderated language throughout the manuscript to avoid overstatements and clearly present the limitations of our data, while preserving the significance of our findings.

There are also potentially ambiguities with the bioinformatics analyses that affect the interpretation of the scale of H3K4me3 recapitulation from gamete to embryo. Some analyses were performed on peaks while others on gene promoters

Thank you for pointing this out. To clarify, all analyses were done around promoters, but depending on the comparisons, we looked at enrichments or at called peaks. For comparisons of different promoters within a single time point, the promoter H3K4me3 ChIP intensity was quantified and compared. Direct comparisons across time points are not appropriate, given the variation in overall H3K4me3 levels. Thus, peaks are called independently at each time point, and instead of comparing signal intensities directly, we assess the binary presence or absence of a peak within the promoter region. We have reported these differences in analyses more transparently in the Methods section of the manuscript (line #881-882).

-- and, it is additionally unclear whether multiple transcript isoforms for the same genes were treated individually (if so, that could lead to double-counting the same genomic locus). As much as possible, the authors should unify their analyses on genes (one transcript isoform per gene for the chromatin/epigenetics, pooled transcripts for the RNA-seq).

To clarify, transcript counts from all isoforms were summed as it was done for the RNA-seq analysis. We have now included an explicit description of how transcript isoforms for the same genes were treated in the methods.

The authors should strive to better relate their findings to other work, especially in *Xenopus*, especially if their results/interpretations disagree. For example, in *X. tropicalis*, Bogdanovic et al 2011 (which is not cited) already found that TSSs are largely DNA hypomethylated in the blastula/gastrula (are their proportions of hyper/hypomethylated promoters similar to the current manuscript's?). And Hontelez et al 2015 also found that transcriptionally inhibited embryos still gain H3K4me3.

We have further incorporated discussion of these important studies, noting their observations of promoter DNA hypomethylation and transcription-independent H3K4me3 establishment.

Please find citations of Bogdanovic et al., 2011 (line #196, #647) and Hontelez et al., 2015 (line #82, #639).

Although *X. tropicalis* (used in their studies) and *X. laevis* (our study) are closely related, they differ in ploidy, which limits direct comparisons between generated datasets (e.g. enrichment analyses, gene set overlaps). Nevertheless, their results align well with our conclusions. This comparative perspective has been added to the Discussion.

Finally, to enhance accessibility of the manuscript, the authors could consider curating their main figure panels, which are numerous and perhaps not all equally contributing to the narrative.

We have curated the main figure panels and removed redundant ones to improve accessibility of the manuscript. For example, in Fig 1, we condensed Fig. 1i-k of the original submission into a single heatmap (Fig. 1g).

Specific points:

- Figure panel placement/ordering is a bit irregular -- following the grid would improve readability

We have rearranged the figures so they follow the grid and are more accessible to the reader.

- Fig 1b, the color scale is not intuitive to this reader. What does "max log scaled per peptide" mean? Is this simply low to high across the entire dataset? Also, is % abundance relative to unmodified?

We acknowledge that the scaling of the heatmap may be unintuitive to the reader. We have now adjusted the heatmap to a simple, linear scale that denotes the percent abundance of each modification, relative to the total abundance of the corresponding peptide in all its unmodified and modified forms and have also included the abundance of the unmodified peptides (Fig. 1b). Each box has been outlined and labelled clearly with its respective peptide and the sum of all possible forms of the peptide equals 100% for each individual replicate. We have also edited the figure legend and ensured that the explanation is self-sufficient.

- Fig 1b, should "H3K9me1" in row 4 be "H3K4me1"?

This has been corrected.

- Fig 1d, were equal numbers of embryos used for the different stages? The H4 controls seem to indicate this. There is an interpretation challenge then. Ideally, H3K4me3 intensity should be quantified relative to H3 (not H4) over replicates to facilitate comparison

While care was taken to use an equal number of cells by scaling the number of embryos for the early cleavage stages (32-cell, 256-cell and 4000-cell), the efficiency of chromatin extraction is lower in 32 cell embryos due to the high yolk content. Nevertheless, the intention behind this plot is simply to demonstrate that H3K4me3 can be detected as early as the 32-cell stage (Fig. 1c). Quantification of the H3K4me3 intensity normalized to H3 revealed that this mark could be detected at comparable levels at 32-cell and 256-cell (pre-ZGA) stages, with a sharp increase in intensity at 4000-cell stage (mid-ZGA), in line with our mass spectrometry observations (Fig. S1b).

We have also repeated the experiment to target H3 in addition to H3K4me3 and H4 (**Plot 1**). We observe that both H3 and H4 intensities progressively increase across time, with similar trends. This supports the validity of using H4 for quantifying relative intensity of H3K4me3.

Plot 1

-Fig 1f, g -- the micrographs suggest that H3K4me3 loss in α -Amanitin is binary, where some nuclei have strong signal while others are completely absent (compared to the H4 panel, which presumably labels all the nuclei). This pattern does not seem to be reflected in the distributions in panel g. The authors should clarify and also overlay the H4 and H3K4me3 staining to better appreciate the variability between nuclei.

We would like to clarify a potential misunderstanding: The lack of nascent transcripts in the α -amanitin-treated condition causes a loss of the 5-EU signal and results in a “binarized” signal. The loss of H3K4me3 upon transcription inhibition, however, is more variable, as is reflected in the violin plots. We have now represented the individual panels in grayscale and added an overlay of H4 and H3K4me3 staining channels for better visualization (Fig. 1e).

- Fig 1i, j -- ideally these should have the same sort order and be clustered. It would be valuable to see the actual signal intensities grouped similarly to Fig 1k, especially since the relative proportions of, e.g. hypo vs hypermethylated promoters don't obviously correspond between the panels

We apologize; the description of the plotted genes was missing in the figure. We point out that in Figure 1J of the first submission, we show only H3K4me3 marked promoters, and not all genes. We have now added gene set information to all heatmaps in the revised manuscript. We thank the reviewer for pointing this out.

We further agree with the reviewer that the different representations in Figures 1I, 1J and 1K can be confusing, and thus the clarity for the reader can be increased by visualizing the information from these figures in a single more detailed heatmap, represented in the revised manuscript (Fig. 1g).

- Fig 1j, k -- the lack of correspondence between H3K4me3 and promoter accessibility is highly unusual. Are there mappability biases with the CATaDa (e.g., presence of a GATC) that could lead to under estimation of accessibility? Are the accessibility profiles consistent with later stage ATAC-seq? Do the H3K4me3 non accessible promoters gain accessibility later?

We appreciate the reviewer's comments regarding the potential influence of mappability biases in CATaDA, specifically related to the presence of GATC sites. To address this, we applied a linear normalization of the CATaDA signal based on local GATC density. While this approach reduces bias, it cannot fully eliminate it. To ensure transparency, we have added GATC density tracks to Fig. 1g, allowing readers to visually assess regions with sparse GATC coverage.

To further assess the reliability of CATaDA-measured accessibility, and test the robustness of our claims, we performed several validation steps:

1. **Comparison with later-stage ATAC-seq:** We compared CATaDA signals to publicly available ATAC-seq data from NF10 and NF12 ectoderm tissue (PMID: 32119833), as well as unpublished CATaDA data from whole NF11 embryos. These comparisons reveal a general concordance in accessibility profiles across techniques and stages, supporting the robustness of the CATaDA signals (see new **Plot 2**).
2. **Temporal dynamics of H3K4me3-positive, low-accessibility promoters:** For promoters in Cluster 2 of Fig. 1g, which are marked by H3K4me3 but show low accessibility at the pre-ZGA stage, we tested whether these regions become accessible at later stages using CATaDA. Indeed, approximately half of these promoters gain accessibility, indicating that their initial low accessibility is not an artifact of the method but reflects true biological dynamics (see new **Plot 3**).
3. **Assessment of GATC content across groups:** Finally, we examined GATC density across all promoter groups analyzed for RNA expression and H3K4me3 levels. We observed comparable GATC distributions among these groups, indicating that our intergroup comparisons are not confounded by significant differences in GATC site densities (see new **Plot 4**).

Together, these analyses support the validity of our CATaDA-based accessibility measurements and indicate that the divergence between H3K4me3 enrichment and accessibility at early stages is a biologically meaningful phenomenon rather than a technical artifact.

- Fig 2b, the legend says "promoter peaks" but is this just counting promoters? As opposed to peaks in promoters?

The reviewer is correct; the plot in question refers to a binarized representation of H3K4me3 peak presence across timepoints at all promoter regions (TSS +/-1kb), resulting in 16 possible groups, which were subsequently combined into five major groups (Kept, Gained, Lost, Absent and Fluctuating), as described in the Methods section. For simplicity, genes within the Fluctuating group were excluded from further analysis. The legend for Fig.2b has been corrected to clarify that all promoter regions are included.

- Fig 2e (also 3d), are there equal numbers of TSSs across each of the four panel sets?

The violin plots for Pre-ZGA and ZGA look like they have far less density than for sperm. If not, why would the number of genes change between stages? Also, heatmaps similar to Fig 1i,j would be valuable here as they would show the variability within groups and between stages relative to the TSS position

Yes, each panel has the same number of genes.

The difference in violin width comes from scaling all violins in each panel to the same area. We agree with the reviewer that this may affect the readability of the density distribution in the pre-ZGA and ZGA plots. Therefore, we switched all violin plots to be scaled to the same maximum width instead of the same area. We have also added a description to the methods section.

We have produced a heatmap similar to Fig. 1g including all timepoints as columns that reveal the change of the H3K4me3 position between stages and groups (Plot 5). However, the H3K4me3 signal intensities themselves cannot be directly compared across timepoints due to the different overall levels of H3K4me3 between timepoints.

- Fig 3b,c the counts here seem quite low for genes activated during ZGA, and even lower than what we understand to be the size of the maternal contribution. Moreover, the total is much lower than the gene total in Fig 2b.

Stringent definitions based on nascent RNA-seq led to smaller gene sets compared to Fig 2b: In Fig. 3b and 3c, we aimed to curate a high-confidence, rather than exhaustive, list of gene groups based on nascent RNA-seq data during 6–9 hpf. This approach prioritizes specificity to avoid misclassification, particularly considering potential technical artifacts. We have addressed the reasoning behind our strategy in the Methods section under “Gene expression group definitions”.

For example, as the original study mentioned (Chen et al., 2022/PMID: 36007528), maternal transcripts can non-specifically bind to beads during nascent RNA pulldown. This may lead to incorrect assignment of gamete-specific (GS) genes as gamete+zygotic (GZ). To circumvent this, we followed the strategy described in the original nascent RNA-seq study (Chen et al., 2022/PMID: 36007528), calculating nascent transcription as the net increase in reads over the 5 hpf background. Genes were only classified as zygotic if they exceeded the TPM threshold (>5) in all three replicates at any timepoint (6-9 hpf). For GS classification, we further required that transcripts be undetectable (≤ 5 TPM) in all replicates during 6-9hpf timepoints.

As a result of these stringent filters, genes that did not meet strict detection criteria across replicates were excluded from further analysis. This explains the lower gene counts in Figure 3 compared to the broader gene set analyzed in Figure 2b, as well as the overall lower counts of maternally contributing and zygotically expressed genes.

Although this approach may underestimate the total number of expressed genes, it ensures that the group definitions reflect confidently assigned expression dynamics, minimizing misinterpretation due to technical noise.

- Fig 3 generally combines sperm and oocyte/egg for the GS and GZ categories, but they probably should be considered separately. For example, oocyte-specific genes are included in the GS curves for 3d, which is likely lowering the average in the sperm H3K4me3 graphs. And in general, genes represented in the maternal contribution are an important category distinct from spermatid-active genes, since maternal but not (much?) paternal mRNA is inherited in the embryo.

We agree that distinguishing between oocyte- and spermatid-specific transcripts is conceptually important, particularly given the maternal bias in RNA inheritance during early development. Unfortunately, H3K4me3 data from oocytes is not available in *X. laevis*, and generating such data is currently technically too challenging and to our knowledge has not been performed successfully.

To address this limitation, we re-analyzed our data after removing oocyte specific genes from the GS and GZ groups (**Plot 6**). The resulting H3K4me3 profiles remained consistent with our original findings: spermatid-specific genes still show lower average signal than spermatid+zygotic genes.

This strongly supports the claim that the inclusion of oocyte-specific transcripts is not driving the observed differences in H3K4me3 profiles.

- Fig 4, Since the Window and Persistent treatments are extremely similar, there is some concern that the AID degradation would be too slow to specifically implicate pre-ZGA H3K4me3 loss as causal for the gene expression defects. It could be that ZGA proper is still being impacted, while H3K4me3 levels slowly recover. If the authors' model is maintenance throughout early embryogenesis, wouldn't disrupting H3K4me3 at only the very early stages be sufficient to abolish propagation of the gamete patterns?

Our data support that transient pre-ZGA loss of H3K4me3 is sufficient to cause gene expression defects at ZGA, even after global restoration of the mark.

While we agree that introducing auxin earlier would further test our model, the developmental window between fertilization and ZGA is extremely short—approximately 4.5 hours at 23°C. Within this time, injected mRNA must be translated, and the resulting proteins must act on chromatin to deplete H3K4me3 before auxin can be administered. Western blot data show that ectopic proteins are mildly detectable by the 4-cell stage (2 hpf), and auxin induction at 4.5 hpf results in near-complete ectopic protein loss within one hour (1000-cell stage; **Plot 7**).

Plot 7

Importantly, we show by immunofluorescence that H3K4me3 levels are globally restored by 6.5 hpf (old manuscript: Fig. 4d, 4e), coinciding with robust transcription in wild-type embryos. To strengthen this point, and as suggested by the reviewer, we performed CUT&RUN at 6.5 hpf and confirmed complete recovery of H3K4me3 at this stage (Fig. 5d-e, Fig. S5a), yet transcriptional defects persist in this condition (Fig. S5d). This supports a specific requirement for H3K4me3 prior to ZGA and argues against slow degradation or incomplete recovery as explanations for the observed gene expression defects.

The lack of ChIP-seq profiles after recovery to compare patterns to untreated strongly diminishes the conclusions we can make here.

To confirm recovery of H3K4me3 levels after inducing degradation of the overexpressed proteins, we performed CUT&RUN on 6.5hpf embryos (Fig. 5d-e, S5a). Using principal component analysis and differential peak analysis, this data revealed that embryos with transient (“window”) H3K4 methylation depletion largely restored H3K4me3 levels genome-wide, clustering closely with controls and wildtype, while embryos with persistent depletion showed widespread loss of H3K4me3 and diverged significantly from controls.

- Wouldn't increasing H3 levels affect the nuclear-cytoplasmic ratio and potentially delay ZGA? Are the control embryos activation profiles strongly different from untreated (or mock injected)?

We acknowledge that the nuclear-to-cytoplasmic ratio of histones is an important regulator of ZGA (Hamm & Harrison, 2018). However, we believe the levels of ectopic H3.3 used in our study are unlikely to significantly increase histone or histone octamer concentrations to perturb this balance for the following reasons:

1. We inject less than 1 ng of ectopic wildtype H3.3 mRNA, which is well below the threshold shown to affect embryonic development in *Xenopus* (PMID: 23318639).
2. Western blot analysis using a pan-H3 antibody showed no detectable increase in total H3 levels following 3xHA-H3.3 overexpression. For full transparency, the exogenous 3xHA-

H3.3 band (expected at ~55–60 kDa) is only visible when probed with an anti-HA antibody. In contrast, pan-H3 antibodies—although effective in detecting endogenous H3—are not sufficiently sensitive to detect the comparatively low levels of exogenous HA-H3.3, even when applied to samples with identical conditions. These observations indicate that the ectopically expressed H3.3 constitutes only a minor fraction of the total H3 pool (**Plot 8**).

3. Our RNA-seq data shows that mean zygotic-specific gene activation is not delayed in control conditions, indicating that ZGA occurs with normal timing despite the mRNA injection (Fig. S5f).
4. While there is a modest reduction in expression of zygotic transcripts in both control and treated conditions, we believe this reflects either mRNA injection-related stress or a role for Kdm5b that is independent of its catalytic function. We now include the profiles of untreated embryos in our heatmap to allow a direct comparison between conditions (Fig. 5g, S5i).

- Fig 4c, H3 needs to be shown also

We appreciate the reviewer's suggestion. However, we believe that H4 serves as a robust and appropriate loading control in this context. As demonstrated in **Plot 1** above, both H3 and H4 levels increase proportionally across early cleavage stages. Including H3 in Fig. 4c would therefore offer limited additional insight from our point of view.

Given the substantial effort required to repeat these experiments in Fig. 4 with additional H3 blots and the minimal expected impact on interpretation, we respectfully maintain the current presentation.

- Fig 4g, the GS and ZS colors are too similar to distinguish

To improve clarity, we made the following adjustments: We changed the point shapes for GS and ZS groups to visually distinguish them regardless of color and increased point size in the legend for better visibility.

These changes improve interpretability while preserving the integrity of the original data.

- Fig 4k (and 6h), a log scale should be used for fold change, otherwise equivalent magnitude increases/decreases will not have the same color intensities. Also, comparison with normal zygotic activation levels (wt vs alpha amanitin?) would be helpful to compare the magnitude of the effect.

We now use the fold change for the differential expression heatmap in previous Figures 4 and 6, and thus also in the violin plots in the same figures. For comparison with normal zygotic activation levels, we also added the wildtype expression to the heatmaps, as well as the log fold changes from a published transcription inhibition experiment with triptolide (Phelps et al., 2022/PMID: 37787392).

- Fig 4i/Lines 404-405, *sox3* is more strongly maternal, so that would potentially account for not being able to detect a change.

We have now included this consideration in the revised manuscript (Fig. 5i and Fig.6f).

- Fig 5, are the categories independent of activation status? According to Fig 4k, the vast majority of activated genes are in the Gained category. But the analyses here suggest the Kept group is once again large.

We thank the reviewer for spotting this- the labels of the heatmaps were switched and are now corrected.

Additionally, only a subset of Kept loci have accessible chromatin according to Fig 1. So, it seems like a more stratified analysis would be informative here that also takes gene activation into account (and perhaps sensitivity to H3K4me3 loss as determined in Fig 4). Otherwise, Fig 5 seems out of place in the narrative

We thank the reviewer for this helpful feedback on the manuscript's flow. The original intent of Fig. 5 was to provide additional rationale for selecting *Cxxc1* and *Kmt2b* as candidate reader/writers of H3K4me3 during early embryonic cell cycles. However, we agree that the way this analysis was initially presented disrupted the narrative and lacked stratification based on chromatin accessibility, gene activation, or sensitivity to H3K4me3 loss.

In response, we have now integrated these analyses more cohesively into the manuscript, placing them in clearer context with the transcriptional and chromatin state data presented earlier. This restructuring improves the logical flow and better supports the relevance of *Cxxc1* and *Kmt2b* in the framework of our model.

- Fig 6, what is the overlap with the H3K4me3 affected genes from Fig 4?

To address this question, we generated a Venn diagram showing the overlap between genes affected in Fig. 6 and those impacted by H3K4me3 loss in Fig. 4, using the union of replicates (Fig. S7o). This analysis highlights the relationship between early chromatin regulation and later transcriptional outcomes.

- Fig 6b, is H3K4me3 reduced at earlier stages e.g. 256cell?

Regarding H3K4me3 levels at earlier stages: Yes, we observe a 30–40% reduction in H3K4me3 relative to H4 by Western blot at the 256-cell stage. This result has been added to the manuscript as Fig. 6b to provide additional support for early depletion prior to ZGA.

- Fig 6d, the volcano plots look odd -- particularly the 7.5hpf timepoints, where there seem to be two distinct groups of downregulated genes (mostly blue and mostly orange). Please verify that the plots are correct

We have carefully rechecked the volcano plots and can confirm that they are correct. We believe that the distinct separation of GZ and ZS genes at the 7.5 hpf timepoint reflects the underlying biological differences between these two groups. ZS (Zygotic-Specific) genes are first transcribed during ZGA, whereas GZ (Gamete+Zygotic) genes also show expression in one or both gametes, resulting in maternally deposited transcripts. We believe that this maternal contribution dampens the apparent fold-change in expression during ZGA stages, causing the observed clustering.

To support this interpretation, we calculated the ratio of unspliced to spliced transcripts in the two groups (**Plot 9**).

GZ genes show significantly lower unspliced/spliced ratios, consistent with a higher proportion of mature, maternally loaded transcripts. Conversely, ZS genes show higher ratios, reflecting the transcription of nascent transcripts during ZGA

- Throughout, note that two-tailed (sided) hypothesis tests should be used, rather than one-tailed, when changes can be in either direction

We would like to clarify that we intentionally used one-sided tests where appropriate, based on prior directional hypotheses derived from our model and previous results. For most of the comparisons (e.g., neighboring violin plots in Figures 2e-h, S2e-g, 3d, S3c-f), the test direction is biologically motivated and the violins are already ordered by their median values, further clarifying the expected trend.

To prevent any confusion, we have now explicitly stated the direction of each test in the figure legends and clarified the rationale in the Methods section.

- Throughout, please check the Western kDa size labels. E.g., Fig 4b shows HA-H3.3 is >55 kDa, Fig 6b shows H3K4me3 is <15 kDa

We have carefully re-checked all molecular weight markers and size labels across the Western blots and have corrected any inaccuracies. The revised figures now reflect the correct kDa annotations for all relevant protein bands.

- Line 134, "modifications on globally transcriptionally silent chromatin" -- this is a presumption, the methods suggest that we can only know the histones are nuclear as opposed to explicitly DNA bound (chromatin). If this is not the case, please clarify

We thank the reviewer for this important point. Our method involves nuclear isolation followed by detergent-based extraction designed to remove soluble and free-floating nuclear proteins, leaving only chromatin-bound proteins. Therefore, the proteins analyzed are enriched for those physically associated with chromatin, including DNA-bound histones.

To ensure clarity, we have revised Line 128 to accurately reflect that the histones analyzed are from chromatin-bound nuclear fractions rather than general nuclear proteins.

- Line 154, "small number of microRNAs" grossly understates the scale and impact that miRNA activation has during early ZGA. In zebrafish, the high copy number miR-430 cluster spans a hundred kilobases (Hadzhiev Dev Cell 2023) and has a high density of marked chromatin (Chan Dev Cell 2019). The *Xenopus* miR-427 cluster is similarly large, per the Owens paper.

The role of microRNAs during early ZGA is important, and we agree that describing the miRNA activation as involving a "small number" understates their scale and impact. To better reflect this, we have revised the manuscript to describe the activation as involving "a defined cluster of microRNAs with significant regulatory impact." We also added references to the zebrafish miR-430 cluster (Hadzhiev et al., 2023; Chan et al., 2019) and the *Xenopus* miR-427 cluster (Owens et al. 2016) to underscore the large genomic span and dense chromatin marking associated with these important regulatory loci.

- Line 283, the analysis doesn't really address whether genes are continuously marked, given the lack of early-stage and oocyte/egg data

To address this, we have carefully revised the manuscript language to avoid suggesting uninterrupted or continuous H3K4me3 presence throughout these stages. While direct data from oocytes and eggs are limited in our study, we have clearly stated this limitation and framed our interpretations accordingly.

- Line 304-307, the authors could evaluate this claim with available public data

Using published bulk RNA-seq data for later developmental timepoints (NF10, NF12, NF15) (Session et al., 2016/PMID: 27762356), we observed that overall gene expression levels of GS genes did not drastically increase from ZGA stages to post-ZGA stages, suggesting that the retention of H3K4me3 at these gene promoters during ZGA is unlikely to be solely explained by later gene expression (**Plot 10**). However, as nascent RNA-seq data beyond 9 hpf is not available, we cannot fully exclude the possibility of later activation.

- Line 420, no functional link has been established between ZGA and "maintenance of H3K4 methylation *from the gametes to the embryo*"

We have revised the statement on line 420 to soften the claim and avoid implying a direct functional link between ZGA and the maintenance of H3K4 methylation from the gametes to the embryo, reflecting the current state of evidence more accurately.

- Line 510, "we show that these H3K4me3 reader-writer enzymes acting independently of transcription..." -- I don't think the authors have demonstrated that these enzymes are acting independent of transcription for *Xenopus* genome activation. This statement should be tempered

To clarify, our original statement referred to Cxxc1 and Kmt2b as H3K4me3 reader-writer enzymes previously shown in other systems to act independently of transcription. We have revised the phrasing to avoid any misleading implication that we have directly demonstrated this independence during *Xenopus* genome activation.

Reviewer #2 (Remarks to the Author)

We appreciate the reviewer's thorough evaluation and the candid feedback offered. Their observations—both broad and specific—have been invaluable in helping us refine the manuscript for greater precision and coherence. Our detailed responses are outlined below:

The manuscript by Oak & Stock et al. aims to provide evidence for the faithful propagation of epigenetic memory of active chromatin states via H3K4 methylation in *Xenopus* in the absence of transcription or instructive transcription factors. What it does achieve is to conclusively support that, as in zebrafish, there is indeed H3K4me before ZGA and that it is maintained from gametes to embryos across a transcriptionally silent window. This finding contradicts earlier claims by the Veenstra lab that H3K4me3 is acquired around ZGA.

The ms goes on to characterize and segment H3K4me bound loci using previously published datasets. They also manipulate H3K4me levels using various tools/antisense morpholinos and monitor differentially expressed genes (DEGs) and embryonic development. However, these results are at best suggestive, but not conclusive. With regard to the physiological relevance,

the results are inconclusive because *Kmt2b* and *Cxxc1* Morpholino-induced deficiencies persist into later developmental stages that compromises attributing phenotype to pre-ZGA stage effects.

Also with regard to the main question, does H3K4me provide epigenetic memory in the absence of INSTRUCTIVE TFs, the results are only suggestive, not conclusive, as such factors cannot be ruled out.

Moreover, while some new pre-ZGA H3K4me datasets are established, the study is to a large extent relying on reanalysis of published data.

So, do the conclusive insights and new data sets of this study warrant a Nat Commun paper? I think it is borderline. In *Xenopus*, the Gurdon lab already provided good evidence for a memory role of H3K4me. This includes PMID 18066050 (not cited), which already showed epigenetic memory of an active gene state depends on histone H3.3 incorporation into chromatin in the absence of transcription. In Fish, H3K4me is also instructive for later developmental gene expression PMID: 22137762. Given these caveats and the many experimental shortcomings (see below) the ms is probably a REJECT at this point.

We appreciate the recognition that “*what it does achieve is to conclusively support that, as in zebrafish, there is indeed H3K4me before ZGA and that it is maintained from gametes to embryos across a transcriptionally silent window.*” This important observation challenges earlier claims, such as those from the Veenstra lab, which suggested that H3K4me₃ is acquired only around ZGA.

Regarding the physiological relevance and timing of *Kmt2b* and *Cxxc1* knockdowns: Our study provides, for the first time in *Xenopus*, functional evidence directly testing the requirement for H3K4me₃ prior to ZGA. Knockdown of the H3K4me₃ writer (*Kmt2b*) and reader (*Cxxc1*) enzymes leads to reduced H3K4me₃ levels in the transcriptionally repressed cleavage stages before ZGA and subsequently results in defective genome activation. We acknowledge that the phenotypic consequences appear later and may include indirect effects, due to current methodological limitations (e.g. absence of antibodies useful for TimAway) that prevent precise temporal restriction of knockdown exclusively to pre-ZGA stages. To tackle this, we implemented an auxin-inducible system to transiently interfere with H3K4me₃ during cleavage cycles prior to ZGA, providing stronger functional evidence that early H3K4me₃ dynamics influence later embryonic transcription and, included in the new manuscript, have an effect on development.

Regarding the question of H3K4me providing epigenetic memory independent of instructive transcription factors: We agree this remains a complex and unresolved question. We have moderated the manuscript language accordingly to avoid overinterpretation.

Regarding reliance on published datasets: While we have incorporated analyses of some published datasets that have not been explored in this context, our study also presents novel, in-depth analyses of early embryonic chromatin. We introduce histone PTM mass spectrometry, DNA methylation profiling via MBD-seq, and chromatin accessibility mapping using CATaDa to provide new insights into a developmental stage that has remained largely inaccessible to traditional profiling methods. These approaches, along with targeted temporal degradation experiments, together provide an original foundation to our conclusions.

Regarding previous work, including the Gurdon lab’s study (Ng and Gurdon, 2008/PMID: 18066050) and zebrafish studies (Aanes et al., 2011/PMID: 22137762): We note that

Hörmanseder et al., 2017 (Cell Stem Cell, PMID: 28366589; also from the Gurdon lab) demonstrated that active transcriptional states are inherited in nuclear transfer embryos from donor cells, correlating with H3K4me3 marking in donor chromatin, but did not directly prove that H3K4me3 itself propagates this memory. Ng and Gurdon (2008) indicated increased H3.3 incorporation enhances memory of a candidate gene in nuclear transfer embryos, yet no disruption experiments or genome-wide analyses were performed. Both Gurdon lab studies were performed in reprogramming conditions, i.e. NT-embryos, and not in wild type embryos generated from gametes. Thus, these studies investigate the propagation of somatic cell memory during cell fate reprogramming and do not tackle the role of H3K4me3 for epigenetic memory during normal development. Zebrafish studies highlight an instructive role of H3K4me3 for later developmental gene expression but do not investigate early functional disruption during rapid cleavage cycles.

In summary, by combining genome-wide profiling, biochemical analyses, and functional knockdowns with temporal degradation tools in *Xenopus*, our work complements and significantly extends prior studies, providing new insights into the role of early H3K4me3 in epigenetic priming of embryonic genome activation. We hope these clarifications address the reviewer's concerns and underscore the novelty and significance of our contributions.

Main points:

1. Fig1b: "We detected a wide range of active and repressive histone modifications on globally transcriptionally silent chromatin of pre-ZGA embryos (256-cell stage). These include the active marks H3K4me1/2/3." H3K4me seems absent in ZGA. Also: LC MS/MS quantification without isotopic standards is at best semi-quantitative. Bottom line: The mass-spec data add little to the main ms, even contradict it re H3K4me levels, but the other histone mark profiles may be interesting for the community and may be difficult to publish otherwise. So one may keep them in.

We show here mass spec analysis of histone modifications in pre-ZGA vertebrate embryos for the first time. Our data indicate with high confidence that H3K4me is not absent at ZGA and H3K4 (mono-, di- and tri-) methylated states are indeed present. While these marks are of low abundance, their presence is consistent and reproducible. We have provided the reviewer with a detailed presentation of the mass spectrometry data analyses in support of our claim as an additional document within this submission. To aid interpretation, we have now revised the figure legend to clarify this point and marked modifications not detected in our assay in dark red text.

As is well appreciated, the use of isotopic standards enables more precise quantification; however, in this analysis, we determined the abundance of each modification by calculating its relative abundance in comparison to other modifications present on the same peptide. As a result, the reported values represent the relative abundance of each modification on a given peptide at a specific timepoint, rather than allowing direct comparisons across timepoints. Although absolute comparisons across developmental stages are limited by peptide-specific ionization efficiency, we believe that the relative enrichment of certain modifications on the same peptide remains informative and valuable for the field. Such data can contribute meaningfully to ongoing efforts to understand the regulatory environment of chromatin in the transcriptionally silent embryo.

2. Furthermore, the color-coding for the percent abundance does not match the scale bar for some of the modifications (e.g. H3K9me3 and H3K27me3 to name a few). It is not clearly stated in the text whether H3K4me3 was followed up due to pre-ZGA level and mid-ZGA level differences. All subsequent results impinge on this experiment, so a clear outcome and inference is important. Fig.1e is clearer in this regard, so the Fig.1b could be moved to the supplement.

To improve clarity and interpretability, we have updated Fig. 1b as follows:

- A linear color scale is now used to represent the percent abundance of each histone modification relative to all modified and unmodified forms of the same peptide.
- Each box is clearly labeled and outlined by peptide, with individual replicates normalized to 100%.
- Modifications that were undetected in our assay are indicated in dark red.
- The figure legend has been revised to be fully self-explanatory.

Regarding the rationale for focusing on H3K4me3: its detection in pre-ZGA embryos, contrary to earlier reports, prompted us to examine it further. While its abundance relative to later stages was not a criterion, the presence of H3K4me3 at this transcriptionally silent stage raised the possibility of a memory function. This hypothesis is further supported by Hörmanseder et al., 2017 (Cell Stem Cell/PMID: 28366589), who showed that transcriptional states can be inherited in nuclear transfer embryos and correlate with H3K4me3 in donor chromatin. This rationale is introduced in the beginning of the manuscript.

3. Fig1c: Y axis is “% occurrence”, i.e. linear scale; Fig. 1b says “log scaled”, i.e. translating into orders of magnitude differences that are not reflected in 1c. Confusing.

We have now incorporated a simple, linear scale for Fig1b that represents the percent abundance of each modification, relative to the total abundance of the corresponding peptide in all its unmodified and modified forms. This representation now matches the linear representation of the occurrence of H3K4me3 in Fig1c.

4. Line 144: “these results reveal for the first time that the highly dynamic chromatin of rapidly dividing pre-ZGA embryos is decorated by both repressive and active chromatin marks.” Early *Xenopus* embryos have large amounts of free histone proteins unbound to DNA that may contaminate the nuclear fractions. In *Drosophila*, PMID: 30639105 concluded that most nuclear histone H3 is not DNA-bound in the early cleavage cycles. How do you know that spurious amounts of pre-ZGA H3K4me3 detected by LC MS/MS are DNA-bound and not free modified histones?

As the reviewer has highlighted, early *Xenopus laevis* embryos have large amounts of free-floating histone proteins, which indeed also include H3K4 trimethylated histones, as has been shown in a recent study using standard whole-embryo protein lysates (PMID: 37027476). We would like to clarify that we have purposely designed a method to isolate chromatin-bound proteins from the cell lysate for the specific focus of our study. The term “nuclear isolation” was imprecise, as the protocol in question is in fact a chromatin isolation method. We have corrected this terminology throughout the manuscript to avoid any ambiguity.

Briefly, our chromatin isolation protocol first enriches nuclei using washes in the nuclear suspension SuNaSp buffer, which simultaneously preserves chromatin integrity and minimizes cytoplasmic and yolk protein contamination. This step is crucial for early embryos due to their excessive yolk protein contamination. Importantly, this is **followed by nuclear membrane lysis** using a strong RIPA buffer, which is supplemented with sodium butyrate to preserve histone modifications. This step is designed to **solubilize and remove free-floating histones**, after which the nuclear chromatin and chromatin-bound proteins are enriched in the pellet and used for downstream analysis.

In our study, this protocol was implemented for sample preparation of selected pre-ZGA stage Western blots (Fig 1c, 1d and 5c) and all mass spectrometry, MBD-sequencing and CATaDa-sequencing experiments.

5. #148 “H3K4me3...were detectable on mostly transcriptionally silent chromatin”. How do you know on what TYPE of chromatin, if any, H3K4me3 locates at this point of the ms?

We have revised the statement in the manuscript to more accurately reflect the data.

6. Fig. 1f-g: What about Pol I inhibition?

Poll is not inhibited by α -amanitin, which primarily inhibits RNA Polymerase II, and at higher concentrations, also inhibits RNA Polymerase III. It has been widely used as a tool to inhibit transcription in *Xenopus laevis*. Ribosomal RNA transcription, largely mediated by RNA Polymerase I, has been reported to be absent during early cleavage stages, with the earliest detection of rRNA occurring at the mid- to late-blastula stage, approximately 4 hours post-fertilization (hpf) (Shiokawa et al., 1981, Newport & Kirschner, 1982). Given this timeline, our use of α -amanitin effectively targets the majority of transcriptional activity during early cleavage cycles and remains suitable for assessing whether early transcription contributes to the maintenance of H3K4 methylation for the developmental stages examined in our study.

7. Fig. 1i-k: what is the nature of the genes with prominent H3K4me3? Those that are induced at ZGA? Protein coding? Others?

We found that genes with prominent H3K4me3 (from cluster 1 and 2 of Fig. 1g) mostly belong to the KEPT genes, consistent with our findings in Figures 2, with a bias of cluster 1 genes towards the GZ group (**Plot 11**). Further GO analysis revealed that cluster 1 genes (H3K4me3-marked and higher accessibility) enriched for translation and protein modification-related terms, while genes from cluster 2 (H3K4me3-marked and lower accessibility) enrich for mRNA processing terms in addition to translation-related functions (**Plot 12**).

Plot 11

Plot 12

8. #197 “this shows that in *Xenopus* embryos at stages that precede ZGA, H3K4me3-marked promoters of transcriptionally silent genes”. How do we know these loci are silent? Even if there were silent (i.e. H3K4me3 deposition is Amanitin-insensitive, which would be nice to show), what is the Pol II binding profile at this stage? Is there paused / stalled /other Pol II at these same loci?

We would like to clarify that our intention here was to convey that this section shows that during the pre-ZGA stage of *Xenopus* embryos, which is characterized by global transcriptional quiescence, promoters exhibit distinct chromatin configurations. However, we recognize that

the original phrasing may have overstated the directness concerning transcriptional status of specific loci and have now corrected it (Line #176).

9. Fig.S1c. pre ZGA H3K4me goes against Veenstra. How many H3K4me peaks were obtained? How does this number compare to the number of post ZGA peaks?

We assume that the reviewer is referring to Hontelez et al., 2015. In this study, the authors performed H3K4me3 ChIP-seq embryos earliest at stage 9, at which time embryonic transcription has already begun, unlike the pre-ZGA timepoint in Fig.S1e. Moreover, the study was performed on *Xenopus tropicalis* embryos, limiting the direct comparisons of peak numbers that we can make across the two studies, as *Xenopus tropicalis* is a diploid organism and has 20 chromosomes, in contrast to the allotetraploid *Xenopus laevis* with 36 chromosomes.

Within our own dataset, the total number of H3K4me3 peaks pre-ZGA and post-ZGA is on average 28,500 and 22,000 across replicates, respectively. However, as the ChIP-seq datasets were not normalized for cross-stage comparisons, these numbers should not be interpreted quantitatively.

10. Fig. 3 does not add much beyond Fig 2, as highlighted in the repetitive conclusion (#342-345). Can go to supplement.

We appreciate the reviewer's comment but respectfully disagree. While Fig. 2 demonstrates the maintenance of H3K4me3 through early development, Fig. 3 provides an essential extension by correlating these chromatin states with transcriptional dynamics at the onset of genome activation. This connection underscores the functional significance of early H3K4me3 marking and adds critical support to our central hypothesis. We therefore believe it belongs in the main figures.

11. Fig.3d: The authors do not comment on the gamete specific group of genes that they identify from the transcriptome analysis and the H3K4me3 signatures associated with promoters of GS group of genes at the spermatid and sperm time points. There is no explanation/speculation as to why H3K4me3 peak intensity is low for gamete specific genes even in gametes as well as the reason for GZ H3K4me3 peak being higher than ZS H3K4me3 peak during pre-ZGA and ZGA time points.

While all actively transcribed genes in our datasets are generally marked by H3K4me3 at their promoters in the corresponding tissue, we speculate that the elevated H3K4me3 signal at promoters of certain gene sets compared to others may reflect a more persistent or reinforced modification. This could support the propagation of the mark and facilitate early transcriptional reactivation. Although we did not comment on this aspect in the manuscript, as the interpretation remains speculative, we acknowledge its potential relevance.

12. Fig. 4 Very elegant approach: Kdm5bci and H3.3wt as the control, were co-injected with TIR1 – which degrades AID-fused proteins upon addition of auxin – into embryos at the one-cell stage; why not use this approach for phenotypic analysis? Too weak effects?

We thank the reviewer for the positive feedback. We agree that the co-injection system is powerful for the analyses of effects on ZGA. In response to this comment, we performed survival assays (Fig. S6) which show overall reduced viability in both control (H3.3^{wt}/Kdm5b^{ci}) and treated (H3.3^{K4M}/Kdm5b^{wt}) groups. We speculate that these phenotypes may result either from effects of the control-injected protein independent of Kdm5b's catalytic activity or from the high levels of injected mRNA required for robust H3K4me3 demethylation prior to ZGA. For context, injection of 5-EU, which is commonly used to label nascent transcripts during ZGA, also leads to later embryonic lethality in frog embryos. Elevated nucleic acid levels may impair development beyond the effect of H3K4 demethylation prior to ZGA.

However, when we titrated mRNA amounts, we find that survival is improved in control embryos, while treated embryos of both window and persistent conditions consistently exhibit specific developmental defects (Fig. S6).

13. Fig4c: "Treated" and "Control" labels seem to be swapped.

The "Treated" and "Control" labels in Fig. 4c were indeed swapped and have now been corrected in the revised figure.

14. #391 "Mean expression levels of the KEPT group of genes were significantly reduced" is not visible in 4i, unlike for GAINED group. It even says "n.s. " at 6.5hpf. To the contrary, the mean expression is even increased in persistent treated at 6.5hpf and at 7.5hpf (4j). In fact, only a handful of genes seems to be affected in 4k. Could this be due to rather moderate effect of Kdm5b/H3 o.e.?

We thank the reviewer for the careful reading. In old Fig. 4i, the KEPT group shows no significant change at 5.5 and 6.5 hpf but becomes significantly reduced at 7.5 hpf in the window treatment, whereas the GAINED group is significantly affected at all time points. In old Fig. 4j, expression levels of the KEPT group are more widely distributed under treatment at 7.5 hpf, but this difference is not significant. We apologize for the label swap between KEPT and GAINED groups in Fig. 4k, which has now been corrected.

We agree that the effect size is moderate but consistent and statistically significant. To clarify this, we have added fold-change expression data upon triptolide treatment (PMID: 37787392) to the revised figures for better visualization and context (Fig. 5g, S5i).

15. Fig.4l and Fig.6i: The effect of maintenance of H3K4me3 and its role in regulating the expression of *sox3* as a ZGA transcription factor is not convincing.

Both *sox3.S* and *sox3.L* are gamete+zygotic (GZ) genes with strong maternal contribution. We expect that the relatively mild reduction in fold change of *sox3* expression stems from the remaining, non-degraded maternal transcripts. We have adjusted the language in the manuscript referring to Fig.4l and Fig.6i (revised manuscript references: Fig. 5i and Fig. 6f) to address this possibility.

16. Fig. 6: *kmt2b* and *cxxc1* Morpholino validation? Rescue with non-targeted wt construct? Two independent MOs?

For *Cxxc1*, we performed a rescue experiment by co-injecting the *Cxxc1* morpholino with a synthetic mRNA encoding a mismatch version of *Cxxc1* bearing a modified translation start site (to render it morpholino-insensitive) and a C-terminal FLAG tag. We observe a mild but consistent rescue of the phenotype, supporting the specificity of the morpholino-induced effect (Fig. S7k-n).

The *kmt2b* gene is large (~9,180 bp coding sequence; ~300 kDa protein), making cloning and in vitro transcription of full-length rescue mRNA difficult. Additionally, due to limited and inconsistent 5' UTR annotations in *Xenopus laevis*, we were only able to design a single morpholino that targets both the L and S homeologs of *kmt2b*. For these reasons, rescue experiments for *kmt2b* could not be pursued. Nonetheless, the specificity of the observed phenotypes is supported by the consistent results across multiple embryos and controls and is further corroborated by complementary data presented elsewhere in the study.

17. 6b: WB should be quantified and normalized to H4. Likely, *cxxc1* Mo achieves not even 50% reduction of H3K4me3 compared to Ctrl Mo. *kmt2b* Mo looks more convincing.

We have added the quantification of H3K4me3 relative to H4 at the NF12 stage (Fig. S7b). We found that *Cxxc1* and *Kmt2b* knockdown conditions resulted in approximately 30%

reduction of H3K4me3 levels compared control conditions. Additionally, we observe a 30–40% reduction at the 256-cell stage (Fig. 6b).

18. #491: " Additionally, we observed a significant overlap of downregulated zygotic genes in individual Cxxc1 and Kmt2b knockdown conditions" Shown where?

The corresponding plot reflecting the overlapping downregulated genes in Cxxc1 and Kmt2b knockdown conditions is now referenced in the text (Fig. S7g).

Also, what is the overlap between DEGs with the analogous experiment in Fig. 4?

We have now included a Venn diagram representing the overlap of downregulated genes across Cxxc1 knockdown, Kmt2b knockdown, window H3K4 methylation depletion and persistent H3K4 methylation depletion conditions (Fig. S7o).

19. #500: "We then addressed whether the observed depletion of Kmt2b and Cxxc1 in embryos was affecting embryonic development. Both Cxxc1 and Kmt2b knockdown embryos showed a delay in development." Phenotype interpretation is compromised by the Mos having effects into later development. Should be mentioned.

We agree that morpholino-mediated knockdown causes sustained depletion of Cxxc1 and Kmt2b beyond the ZGA window, which may contribute to developmental defects observed at later stages. This potential confounder is now explicitly mentioned.

Importantly, our gene expression analyses focus on the ZGA timepoints, capturing the impact of Cxxc1 and Kmt2b depletion during the transcriptionally quiescent window and onset of ZGA. While both enzymes have known transcription-independent functions relevant to early development, we do not claim that early knockdown alone fully accounts for later phenotypes. The persistence of MO effects likely influences subsequent development. Furthermore, maternal protein contributions may allow embryos to initiate ZGA despite knockdown, but partial depletion of these critical H3K4 methyltransferases still leads to measurable developmental delays.

We have now added a clarifying statement in the results and discussion to acknowledge this limitation and caution in interpreting later developmental defects (Line #572-574).

The Tir1 approach with Auxin degradation should not suffer from this problem and may be more interesting to study. Same for #541: "we can conclude that loss of H3K4 methylation results in defective gene activation and embryonic death."

We agree, and as mentioned above, we have now incorporated the survival assay of the Kdm5b + H3.3 overexpression experiments into the manuscript (see Figure S6).

20. # 542: "our data suggests that H3K4me3 maintenance occurs independently of transcription and of the signal that initially induced the state during gamete maturation." Pioneer maternal TFs cannot be excluded to program H3K4me3 sites during each cell cycle.

We agree that maternal pioneer transcription factors may contribute to programming or reinforcing H3K4me3 at specific sites during each cell cycle. We have mentioned this aspect in our manuscript. At the same time, our data indicate that the maintenance of H3K4me3 prior to ZGA occurs independently of ongoing transcription and does not require the continuous presence of the original signals that established these marks during gamete maturation.

Minor points:

1. Text wise the Results is not an easy read, multiple, complex figures are lumped together in the Results section for the reader to make of them, without being helped to interpret them

individually (e.g. #281,, 221, 271, 273, 389, 409). If their messages is no different, why show all the panels without text-wise distinguishing them?

We acknowledge that the Results section contained a dense presentation of multiple complex figures. To improve clarity and ease of reading, we have carefully reorganized and simplified the Results, ensuring that each figure panel is clearly introduced and interpreted in turn. This restructuring guides the reader through the key messages without redundancy.

2. The figure legends do not provide complete information, which would make the figure self-explanatory.

In response to the concern about figure legends, we have revised and expanded them to provide complete, self-contained explanations. This will allow the figures to stand independently and improve accessibility for readers.

3. Fig.1d: H3K4me3 seems to be detected at 256 cell stage and not 32 cell stage.

Regarding Fig. 1d (Fig. 1c in the revised manuscript), we have included quantification of H3K4me3 intensity relative to total H3 (Fig. S1b). These new data confirm the presence of low but detectable levels of H3K4me3 already at the 32-cell stage, followed by an increase at the mid-ZGA stage (4000-cell), consistent with our mass spectrometry findings.

4. Fig.1f: The authors check H3K4me3 and H4 staining in mid-ZGA embryos but point out (lines 154-156) that some transcription occurs in pre-ZGA stages in other species. The staining should likewise be performed in pre-ZGA embryos to test the hypothesis that early transcription reinforces H3K4 promoter methylation. This is especially required because the genomic context of H3K4me3 occurrence is later explored in pre-ZGA embryos

Regarding the direct examination of H3K4me3 in pre-ZGA embryos (revised manuscript Fig. 1e): We attempted immunofluorescence staining for H3K4me3 at pre-ZGA stages, however, due to technical limitations such as yolk protein interference at this very early timepoint, we were unable to obtain interpretable or reproducible results.

To address this indirectly, we performed immunofluorescence on α -amanitin-injected embryos at mid-ZGA as a functional proxy. This approach allowed us to assess the dependency of H3K4me3 on transcription in embryos during the onset of zygotic transcription. Our observation of only a modest reduction in global H3K4me3 levels in α -amanitin-treated embryos suggests that a substantial portion of H3K4me3 is retained even when transcription is broadly inhibited, supporting our model that H3K4me3 is retained, at least in part, independently of ongoing transcription.

5. Fig.2e and 2f: The “gained” and “lost” group nomenclature is rather unclear as to when exactly the gain and loss of signal takes place. There are three stages at which loss can take place and two stages at which gain can take place. However, there is one line that corresponds to the gain and loss groups each in the graphs. Some clarification in the legend or text might be helpful.

The subgroups have been grouped together for the sake of simplicity and the reasoning is explained in the Methods section. However, we have now elaborated on this also in the text.

6. Fig.S3: It is unclear whether the network with the central “mRNA processing” node belongs to the GZ genes (Fig.S3b) or GS genes (Fig.S3d), mentioning this in the legend would be helpful.

We have rearranged the figures to make the association of these networks clear (Fig. S3b).

7. Fig.3h: It is important to check the expression of GS group of genes at various time points monitored, as a negative control. It should be mentioned in the legend that the heat map only includes GZ and ZS group of genes.

We thank the reviewer for the suggestion. The GS group of genes was specifically defined based on consistently low gene expression levels across all monitored time points, as described in the Methods section and presented in Fig.3c. As such, these genes have not been included in the zygotically expressed gene group analysed in Fig.3h (revised: Fig. 4).

We have now adjusted the figure legend to clarify that only zygotic (GZ+ZS genes) were considered for this analysis.

8. Fig.4d, 4e: The reduction in fluorescent signal of H3K4me3 in treated persistent samples cannot be attributed only to overexpression of Kdm5bwt and H3.3K4M mRNAs as the H4 signal that should otherwise remain constant also seems to be reduced (lines 374-375).

The description of this observation is also a bit vague in the main text without alluding to the specific panels. Also, explain “persistent”

The apparent reduction in H4 signal in the representative image was due to a slightly lower number of nuclei in that particular Z-stack plane and the quantification shown was performed using consistent segmentation conditions across multiple embryos.

However, we have taken the reviewer’s comment into consideration and have now replaced the immunofluorescence panel in this figure with CUT&RUN on 6.5hpf embryos to assess the levels of H3K4me3 in the different treated and control conditions (Fig. S5a-c). Using principal component analysis and differential peak analysis, we revealed that embryos with persistent depletion of H3K4 methylation indeed showed widespread loss of H3K4me3 and diverged significantly from controls.

The text has been revised to be more specific and “window” and “persistent” have been introduced more clearly.

9. Fig.4 and Fig.S4: The results of experiments shown correspond to two biological replicates that are not in agreement. The interpretations in the text however describe one replicate over the other, therefore the claims cannot be generalized. E.g. expression of KEPT group genes is significantly reduced only in treated embryos at 7.5hpf and not 6.5hpf as is claimed in lines 391-392 as seen in Fig.4i for one biological replicate. The disagreement between biological replicates is also observed in the number of DEGs identified upon Cxxc1 and Kmt2b in Fig.S6f.

We have adjusted the language in the manuscript to ensure that all interpretations hold true for each individual replicate.

10. Fig5: Figs 2-3 deal with metanalysis, Fig 4 introduces Kdm5b: why return to metanalysis in this figure after the Kdm5b chapter? The figure does not even recur to the Fig 4 Kdm5b - DEGs. The logic of the ms Figure flow escapes me. Figure can also go to supplement.

We have rearranged the figures to improve the flow and clarity of the manuscript.

11. Lines 415-417 should be corrected towards the end of the sentence.

This has been corrected.

12. Lines 430-432, Fig.S5a only depicts lower DNA methylation at promoters of KEPT group of genes in gametes but not higher CG content and increased DNA accessibility.

This panel was removed to improve flow and clarity of the manuscript.

Response on reliability of H3K4me detection via mass spectrometry

Fig1b: “We detected a wide range of active and repressive histone modifications on globally transcriptionally silent chromatin of pre-ZGA embryos (256-cell stage). These include the active marks H3K4me1/2/3.” H3K4me seems absent in ZGA.

As a proof of identification and quantification of the peptide containing the H3K4me2 and me3 modifications, please find below the Skyline TIC profiles of the H3K4 related peptides (unmodified, mono-, di- and trimethylated on K4), together with their quantification and RT distribution, and a representative annotated MS/MS spectra from a Proteome Discoverer 1.4 (Sequest/Mascot) search.

Skyline results for H3K4unmodified peptide: Upper left panel shows the selected peptide (TKQTAR, propionylation on K₄ i.e. unmodified, MH²⁺= 380.7192), lower left panel shows the m/z values corresponding to the apex of the extracted ion chromatogram (XIC) displayed in the middle panel for sample 01 (red dot), where we also show the XICs of the other five samples (02-06). Upper right panel shows the quantification of the area under the XIC peaks in the middle panel, and lower right panel shows the distribution of the retention times (RT) of the quantified peaks.

Skyline result for H3K4me1 peptide: Upper left panel shows the selected peptide (TKQTAR, methylation-propionylation on K₄ i.e. methylation, MH²⁺= 387.7271), lower left panel shows the m/z values corresponding to the apex of the XIC displayed in the middle panel for sample 01 (red dot), where we also show the XICs of the other five samples (02-06). Upper right panel shows the quantification of the area under the XIC peaks in the middle panel, and lower right panel shows the distribution of the RT of the quantified peaks.

Skyline result for H3K4me2 peptide: Upper left panel shows the selected peptide (TKQTAR, dimethylation on K₄, MH²⁺= 366.7218), lower left panel shows the m/z values corresponding to the apex of the XIC displayed in the middle panel for sample 01 (red dot), where we also show the XICs of the other five samples (02-06). Upper right panel shows the quantification of the area under the XIC peaks in the middle panel, and lower right panel shows the distribution of the RT of the quantified peaks.

Skyline result for H3K4me3 peptide: Upper left panel shows the selected peptide (TKQTAR, trimethylation on K₄, MH²⁺= 373.7296), lower left panel shows the m/z values corresponding to the apex of the XIC displayed in the middle panel for sample 01 (red dot), where we also show the XICs of the other five samples (02-06). Upper right panel shows the quantification of the area under the XIC peaks in the middle panel, and lower right panel shows the distribution of the RT of the quantified peaks.

Quantification of the forms of peptide 3-8 from H3 containing K₄. The abundance of each peptide as quantified by Skyline was normalized to the sum of abundances of all the H3 3-8 forms and displayed as percentage.

MS/MS spectrum with assigned fragments to the peptide sequence of the H3 3-8 peptide containing a propionylated K4 residue. The presence of the propionylation is consequence of the sample treatment required for bottom-up histone PTM analysis, K4 should be considered as biologically unmodified. Please refer to "Völker-Albert MC, Schmidt A, Forne I, Imhof A. Analysis of Histone Modifications by Mass Spectrometry. Curr Protoc Protein Sci. 2018 92(1):e54. doi: 10.1002/cpps.54." for more details.

MS/MS spectrum with assigned fragments to the peptide sequence of the H3 3-8 peptide containing a propionylated and methylated K4 residue. The presence of the propionylation is consequence of the sample treatment required for bottom-up histone PTM analysis, K4 should be considered as biologically monomethylated. Please refer to "Völker-Albert MC, Schmidt A, Forne I, Imhof A. Analysis of Histone Modifications by Mass Spectrometry. Curr Protoc Protein Sci. 2018 92(1):e54. doi: 10.1002/cpps.54." for more details.

MS/MS spectrum with assigned fragments to the peptide sequence of the H3 3-8 peptide containing a dimethylated K4 residue.

MS/MS spectrum with assigned fragments to the peptide sequence of the H3 3-8 peptide containing a trimethylated K4 residue.

We are aware that the MS/MS spectrum of the H3K4me3 peptide is not as informative as the spectra for the other three forms. This is due to the low intensity of the peptide (<0.05% of the total H3 3-8 peptide) and to the fact that the RT at which the MS/MS was triggered (22.44 min, red dot in the left panel below) is not at the apex of the elution peak (22.42 min) but almost at the end (22.45min). Therefore, the amount of isolated and fragmented peptide is low and the presence of other co-occurring signals (right panel below) introduced unrelated signals in the MS/MS spectrum.

Despite these considerations, we have several reasons to be confident that the peak assigned to H3K4me3 in this study is the correct one:

1.- Based on the long-term experience in analyzing histone PTMs and previous benchmark analysis of the LC-MS conditions with synthetic labelled peptides including the different modification of the H3 3-8 peptide among others, it is known that H3K4me2 and H3K4me3 elute at very similar RT as we can observe in our data (see the upper panel in the figure below for H3K4me2, and lower panel for H3K4me3).

2.- Considering the accuracy of the peptide measurement (>1.5ppm), the output from Proteome Discoverer 1.4 (Sequest/Mascot) did only consider the H3K4me3 as a possible candidate of the MS/MS spectra for the m/z 373.7296.

3.- The biological samples were first acid-extracted, after which the gel bands corresponding to the histone region were excised prior to propionylation and tryptic digestion, and then analysed by LC-MS. These histone enrichment steps provide extra specificity and confidence in the identification of signals where the MS/MS is not completely conclusive.

POINT-BY-POINT RESPONSE:

We thank the reviewers for the time and effort that went into the revision of this manuscript. We have addressed the comments in the responses below.

Reviewer #1

In their revised manuscript, Oak & Stock et al have clarified many of the points raised previously, generally to the improvement of the manuscript. At this point, I do not think any additional experiments are warranted. **However, I still feel that the framing of the narrative as illustrating epigenetic memory is too strong relative to the results.** Additionally, there are points where a more objective acknowledgement of the actual results would be desirable, as well as more rigorous statistical approaches, as described below.

While we had toned down the language in the manuscript according to the reviewer's first round of comments, we understand that this term might still be a concern for the reviewer. Our use of the term "transcriptional memory" was based on widely cited definitions of epigenetic memory, such as "epigenetic memory is defined as the maintenance of gene expression states through cell generations in the absence of the initiating signals" (PMID: 24755934).

However, we believe that our use of terminology should not be grounds for rejection. We further adjusted the and **removed the terms "epigenetic memory" and "transcriptional memory" from the title, abstract and results/conclusions sections of the manuscript. We also no longer use the verbs "retain" or "maintain" in these sections.** We now only discuss that our results are consistent with a role of H3K4me3 for epigenetic memory.

1. The title is claiming a phenomenon of epigenetic memory (though rephrased as "transcriptional memory") despite the interpretation caveats

We now understand the reservations of the reviewer and changed the title.

2. The use of the terms "retained" versus "maintained" throughout the manuscript doesn't really change the implication of continuous marking. In the plausible scenario where K4me3 is passively lost during early cleavage stages but then reappears at Stage 7-8 at the same locations, one would not call that either "retained" or "maintained" because neither of those terms acknowledges the dynamics. Similarly, the categories "KEPT," "GAINED," and "LOST" strongly imply continuity; alternative terms such as "SHARED," "EMBRYO-SPECIFIC," and "GAMETE-SPECIFIC" would be more objective and appropriate.

We understand the residual concerns of the reviewer. We thus **removed the terms "retained" and "maintained" as well as "retention" and "maintenance" throughout the manuscript.**

We changed the term "KEPT" to "SHARED", ask however to use "LOST" or "GAINED", instead of "embryo specific" and "gamete specific", as the latter overlap with labels of the gene expression groups.

3. The similarity of 32-cell and 256-cell states of H3K4me3/H4 only supports the absence of global erasure between those stages (with the caveat that the Western quantification is not very precise), not between the zygote and 32-cell. The Long et al 2023 result is difficult to interpret since their Western shows surprisingly high and equivalent H3K4me3 throughout St 1-9 (and highly reduced at St10), which would seem to contradict the current manuscript Fig 1c (if there's a free vs chromatin-bound distinction, then I don't know how the Long et al result could be used as support here).

Please note that in our study, there is no claim of continuous presence and there is no claim of an “absence of global erasure” after fertilization. In fact, we specifically highlight this limitation in our manuscript.

We reiterate that we do not conclude direct propagation of H3K4me3 at gene promoters across mitotic cycles but rather document its detectable presence prior to ZGA. In this study, we report for the first time the existence of chromatin-bound H3K4me3 in pre-ZGA embryos using Western blot and mass-spectrometry, and we also extended the Western blot technique to 32-cell stage embryos. We conclude that the mark is found on chromatin at developmental time points prior to ZGA. The presence of H3K4me3 on chromatin so early in development has not been shown before in *Xenopus* (PMID: 19758566, 26679111).

We refer to the study by Long et al, which analyzed whole cell extracts, as supportive evidence showing that even at the level of total histones, there are no major changes in H3K4me3 levels until ZGA begins, even in the zygote (st1). Moreover, H3K4 trimethylation is deposited by Set1/COMPASS that bind chromatin during catalysis (PMID: 11687631, 12667454, 22663077). Considering this alongside the relatively constant H3K4me3 levels between 32-cell and 256-cell stage embryos, with chromatin-bound ChIP-seq profiles observed at 256-cell embryos, it is reasonable to assume that H3K4me3 detected in early embryos is at least in part chromatinized. While this does not constitute direct proof of chromatin-bound H3K4me3 in the zygote, we emphasize that this represents the current technical limit of the field.

Please note that our study does not examine chromatin at developmental stage 10, so it is not possible for the reviewer to conclude that the reduction Long et al. report at stage 10 contradicts our observations.

In addition, since we do not want to imply continuity, we will remove “epigenetic memory” and “transcriptional memory” from the title, abstract and results section.

4. The authors strongly emphasize transcription independence and the transcriptionally silent early embryo, but this should also be moderated somewhat. We cannot exclude the possibility that low-level, non-specific, and/or abortive Pol II engagement are occurring prior to major ZGA, which may contribute to chromatin state, especially as the embryo approaches Stage 8/9. miR-427 activation is already detectable prior to Stage 8 in trop (Owens) and laevis (Phelps). Also, amanitin and triptolide are elongation inhibitors, which does not necessarily prevent association of transcription machinery to chromatin, and certainly would have no effect on pioneer transcription factor binding and their capacity to recruit histone modifying enzymes.

We apologize if this was not entirely clear from our manuscript, and we moderated our language where this was not obvious.

To the first point- we cannot fully exclude the possibility that low-level, non-specific, or abortive Pol II engagement occurs prior to major ZGA, which could in principle contribute to chromatin state. However, if this were the primary driver of H3K4me3, one would expect the marks to be more stochastic or randomly positioned. Instead, we observe H3K4me3 at highly reproducible, promoter-specific loci across biological replicates and all assayed timepoints. Furthermore, these loci correlate with CpG density and DNA hypomethylation, indicating that chromatin features, rather than transcription alone, largely define H3K4me3 positioning. While low-level Pol II activity cannot be completely ruled out, the **data strongly suggest that**

H3K4me3 marking at these promoters reflects specific chromatin architecture rather than non-specific or abortive transcription.

For the next point, please refer to the manuscript in lines 161-163: “In *Xenopus*, as well as in *Drosophila*, zebrafish and mouse, a defined cluster of microRNAs and β -catenin target genes with significant regulatory impact are transcribed before the major wave of ZGA. We hypothesized that such underlying early transcription may reinforce H3K4 promoter methylation through transcription-coupled mechanisms mediated by RNA Polymerase II and III, as RNA Polymerase I is absent in pre ZGA embryos.” **Importantly, we do not exclude this possibility and additionally demonstrate that the majority of H3K4me3 marks are retained on the chromatin of mid-ZGA embryos even in the absence of RNA Pol II and III-mediated transcription (Fig.1e).**

Finally, the reviewer suggests that our transcription inhibition experiments do not necessarily prevent the association of transcription machinery (e.g. RNA Pol II and associated machinery and TFs). We are afraid we cannot entirely follow this argument of the reviewer:

1. The statement of the reviewer that amanitin and triptolide does not affect Pol II–DNA association is inconsistent with existing published data showing that **both drugs destabilize Pol II and prevent its recruitment to chromatin**: α -amanitin not only inhibits elongation but also destabilizes the RPB1 subunit, leading to a rapid loss of Pol II in mammalian cells within minutes (PMID: 11904382, 8760875). The rate of this degradation is concentration-dependent and given the high dosage of α -amanitin we inject, and the extended period of exposure, residual Pol II–DNA association is an unlikely occurrence in our system. In contrast, triptolide targets the TFIIH complex and blocks transcription initiation itself, thereby preventing Pol II recruitment. It has also been reported to trigger proteasome-dependent degradation of Rpb1 (PMID: 21931633). Thus, while we cannot exclude the possibility that maternal pioneer factors contribute to chromatin state, our inhibitor treatments strongly reduce the likelihood that Pol II binding or transcriptional activity accounts for the observed promoter H3K4me3.
2. Our direct labeling of nascent RNA followed by immunofluorescence shows no detectable nascent transcripts in pre-ZGA embryos. Importantly, even if low-level transcription occurs at this stage, Pol II occupancy alone does not provide direct evidence of productive transcription and can represent a stable, paused state (PMID: 24184211).
3. Although Pol II ChIP-seq has not yet been successfully performed in *X. laevis* pre-ZGA embryos, Pol II ChIP-seq data from *X. tropicalis* show minimal Pol II occupancy on pre-MBT chromatin (PMID: 31537794), and H3K4me3 profiling after transcription inhibition in this model is consistent with our results (PMID: 26679111).

While our manuscript does not claim independence from maternal transcription factor binding, we concede that the deposition of a certain proportion of H3K4me3 at promoters may stem from such mechanisms. We specifically discuss this possibility in our manuscript. Based on some literature references (e.g. PMID: 24755934), this potentially parallel deposition mechanism may still be considered an epigenetic mechanism. However, we appreciate that there is at this moment no uniform definition of epigenetic memory across the field and thus we will remove “epigenetic memory” and “transcriptional memory” from the title, abstract and results section.

5. The CATaDa is indeed showing a noticeable bias against GATC-poor regions in Fig 1g, suggesting that the category of "High H3K4me3 / low accessibility" may indeed be artifactual. Curiously, the authors do not seem to discuss this in the manuscript, just showing the heatmap

with no explanation. Response Plot 4 shows no apparent differences in GATC promoter content between groups, but we might presume the groups are all equivalently composed of some GATC-containing and non GATC-containing promoters. Given that the proposed existence of K4me3/low accessibility promoters has little to no bearing on the rest of the manuscript, it's unclear why the authors would not acknowledge the ambiguity.

The “bias” of CATaDa towards GATC motifs is an inherent feature of the method, which is why the **signal is now normalized to the frequency of underlying GATC sites in Fig.1g**, as is standard practice in Dam-ID based studies (PMID: 30877125, PMID: 40273908). We thank the reviewer for this comment and have added clarification in the manuscript and figure legend to make this explicit.

Importantly, we would like to point the reviewer again to Response Plot 3, which directly addresses and alleviates the concern that the “High H3K4me3 / low accessibility” category might be artificial: **further clustering within this group reveals a substantial subset of promoters that remain less accessible at later stages, as independently confirmed by “non-biased” ATAC-seq datasets** (PMID: 32119833).

6. Response Plot 5 should be included in the manuscript, as it lends support to the defined categories that is otherwise harder to appreciate from the existing figures. The comparison of signal intensity across timepoints is not a concern; the within-timepoint comparisons between groups are what is valuable here.

We agree and are happy to include this in the manuscript (Fig.S2c)

7. Response Plot 6 when compared to Fig 3d seems to show subtly different dynamics, but in a surprising direction -- indeed, sperm genes have profiles that more closely match zygotic-specific (the curves are closer together in Response Plot 6 than in Fig 3d). But this would then indicate that egg- specific genes have stronger H3K4me3 signal in the embryo as well as in sperm, which is an odd result. I do not agree with the choice to exclude this analysis and the discussion from the manuscript.

We agree and we currently do not have an easy explanation for this data; the results look quite interesting and might prove to be a valuable area for future research, provided that the current significant technical limitations of working on egg and oocyte chromatin in *X. laevis* can be overcome. However, we believe that this plot does not add information and does not challenge or contradict any of our conclusions. Moreover, if included, there is potential to confuse and distract from the main message of the manuscript, which is that the genes expressed in either of the gametes and the embryo are both marked by H3K4me3 at their promoters and that this is important for proper ZGA and embryonic development. Upon both reviewers' recommendations, we removed many accompanying plots from the manuscript to make it concise and adding this plot would be contradictory to this exercise.

Nevertheless, we have included this information in the supplementary figure (Fig.S3d-e, Lines #318-327).

8. The H3K4me3 CUT&RUN was a valuable addition, but the Fig S5A PCA plot seems to show some poorer replicate correlation, and the intermingling of treatment and control conditions in PCA space, calling into question the significance of the results. Heatmaps across all promoters, stratified according to gene group, would likely be more informative.

There is a misunderstanding of the experimental design. There are two conditions, each with their own treated and control samples: the “window” and “persistent” conditions. Based on our hypothesis, only the treated embryos from the persistent condition are expected to show reduced H3K4me3 levels and therefore to cluster separately from the other three groups (“treated” control, “persistent” treated and “persistent” control). This pattern is indeed observed in the PCA plot.

In our dataset, differential peak analysis was performed using a DESeq2-like approach, which focuses on shared genomic regions and normalizes for replicate-to-replicate variability. Unfortunately, the suggested genome-wide heatmaps do not provide additional resolution, as local changes at promoters are subtle and not easily visible by eye. Instead, to address the reviewer's suggestion, we generated promoter-centered (TSS \pm 1 kb) metaplots for each gene group, shown both as averages (Plot 1) and as individual replicates (Plot 2). These plots demonstrate that, despite minor variation between replicate 1 vs replicates 2 and 3 for the KEPT/SHARED group of genes, all are consistent: They show reductions of H3K4me3 signal in the persistent depletion condition and overall restoration in the window depletion condition, confirming the reproducibility of the observed trend across replicates.

For completeness, we also provide a PCA restricted to promoter peaks only (Plot 3), which recapitulates the clustering pattern observed in the genome-wide data and reinforces that the primary source of variation is biological rather than technical. Small developmental and clutch-dependent variability is common in *Xenopus* embryos, as discussed in the previous revision round, and does not affect the interpretation of the consistent treatment-associated changes observed here.

Plot 1

Plot 3

Plot 2

9. Fig 5h, S5J, 6E appear to be versions of RNA-seq differential expression tests, but not performed in a way consistent with the field. Control vs treatment should be compared using

something like DESeq, then the number of significantly differentially expressed genes for each of the KEPT and GAINED groups can be reported.

We apologize for the confusion- **we have indeed performed DESeq2 on the RNA-seq data and compared treatment vs control using all replicates; please refer to the “RNA-seq data processing” part of the Materials and Methods section** for clarification. The number of so identified differentially expressed genes are represented in Fig.S5b and Fig.S7c. Following this initial analysis, we have zoomed in on zygotic genes and used additional approaches to test the validity of our claims.

The issues with the manuscript's current approach are:

a) only one replicate is used for the stat test, but all replicates need to be considered; In the plots mentioned above (Fig.5h, S5J, 6E) we have calculated log2 fold change of gene expression levels of all zygotic genes calculated over mean TPM of the respective control condition independently for each biological replicate.

For each experiment we used embryos from a different frog clutch, with three biological replicates per time point (each a bulk of 3 embryos) analyzed by DESeq2. Because ZGA is highly dynamic and development can vary slightly between clutches, we analyzed each clutch independently and showed reproducibility by presenting the intersection of differentially expressed genes (Fig.S5c, Fig.S7d). This approach of separate clutch representation allows us to maintain statistical confidence and is consistent with the field (PMID: 26774488, 24757007).

b) a Wilcoxon rank sum test is used, which again is not an appropriate test to use for a two-condition RNA-seq comparison, but even so it is an unpaired test when in fact the data are paired (you're measuring the change of the same genes between control and treated); We agree with the reviewer that a paired test, or even more so, a one-sample test against the expected mean logFC of 0 may be more appropriate for this analysis. We have rechecked the data using this approach and can confirm that the p-value ranges remain unchanged.

c) a one-tailed test is used despite the possibility of changes in either direction and thus allows results to be reported as non-significant even if there is a significant change in the opposite direction (which may be the case in Fig 5h persistent-kept). The argument about testing a specific directional hypothesis, while arguably technically valid, is not in the spirit of the scientific endeavor here.

Our use of a one-tailed test reflects the directional nature of our hypothesis: namely, that H3K4me3 levels would decrease in the persistent condition. In such cases, a one-tailed test is a statistically appropriate and widely accepted approach, as it increases power to detect changes in the predicted direction without inflating false-positive risk.

While we recognize that a two-tailed test could capture unexpected changes in the opposite direction, our study is designed to test a specific prediction, rather than to explore all possible outcomes. For this reason, we believe that the use of a one-tailed test is justified and in line with the goals of the work. Nevertheless, we have reanalyzed the data and can confirm that all of the significant p-value ranges that have been reported did not change upon the use of a two-tailed test.

10. Fig 5i and 6e likewise seem to be non-standard stat tests for RNA-seq data -- what test specifically isn't stated, and how the replicates are used is unclear.

We think the reviewer is referring to Fig 5i and Fig 6f. We can optimize the figure legend here to include the statistical test. Both replicates have been combined in this plot and this is clearly stated in the figure legend for both plots in question.

11. Fig 6b and S7b lack statistics. Inspection of the points suggest poor replicability between Western blots (indeed, Fig 6b Kmt2b MO lane seems problematic with loading), which does raise caveats about the strength of the MO effects -- stating the average % loss is probably not realistic.

The point of this plot is to demonstrate an approximation of the magnitude of the effect on global H3K4me3 levels in pre-ZGA embryos. As morpholinos do not deplete maternally deposited protein levels and the targeted H3K4 methyltransferases compensate for each other to a certain extent, we do not expect a complete loss of H3K4me3. Moreover, these Western blot measurements are inherently semi-quantitative and can vary depending on morpholino diffusion and embryo-to-embryo variability, making exact percentage estimates indeed difficult. However, the magnitude of the effect can be safely estimated and compared. **Importantly, we do observe a reduction consistently across embryos and experiments, which is sufficient to cause reproducible developmental phenotypes, supporting the functional relevance of the knockdowns.**

12. Fig S7M uses an inappropriate stat test (Fisher's) that does not take the replicates into account. Pooling the replicates can obscure large magnitude deviations in trials with smaller N, for example. Indeed, the plot shows high variability in a couple of the conditions

We kindly disagree and provide the following reasoning that the test is appropriate here: The data in question is categorical survival data, and each embryo technically represents an independent biological outcome. Also, even though we acknowledge that in principle replicate-to-replicate (or expt-to-expt) variability across embryo clutches might be masked with this method, in our case the number of embryos was comparable across replicates (included in the source data file) and the trend of each independent experiment was consistent. Nevertheless, to address this concern directly, we checked with additional statistical analysis that considers the stratified data by the different replicates, such as Cochran–Mantel–Haenszel (CMH) test and mixed-effects logistic regression (MELR) and can confirm the reported significance with both approaches (CMH p-value: 0.00636; MELR p-value: 0.00489).

13. Fig S1A: Please indicate the y axis is on a log scale

We have made this addition.

14. Fig S6A - I'm struggling to interpret the western relative to the legend; additional labeling may help

We have adjusted the labeling of the plot and the corresponding figure legend to increase clarity.

Reviewer #2

I appreciate the clarifications, edits and additional experiments. However, several substantive issues remain unresolved and limit the strength of the conclusions:

1. Stage-restricted functional tests: While the auxin-inducible system is an improvement, there is still no phenotypic analysis confined strictly to the pre-ZGA window.

We apologise if this was not clear from our revised manuscript. The contrary is actually the case: we have done phenotypic analysis specifically for this experiment (Fig.5, Fig.S5, Fig.S6). The auxin system is confined strictly to the pre-ZGA window. Embryos, in which we performed interference with H3K4m3 strictly in the pre-ZGA window, were analyzed 1) by **WB at pre-ZGA stage 7 showing a global reduction of H3K4me3 (Fig.5c)**, 2) RNA-seq during ZGA timepoints revealing effects on ZGA (Fig.5, Fig.S5) and 3) a survival assay with a titration

of the dosage for H3K4 methylation depletion (Fig.S6) showing reduced developmental success when compared to their respective controls. Hence, phenotypical analyses were performed in the pre-ZGA window, at ZGA and after ZGA.

Morpholino knockdowns persist into later stages, making it impossible to exclude indirect effects on gene expression and development.

Although morpholino knockdowns persist beyond the cleavage stages, the molecular effects we report were assayed at ZGA, when the earliest transcriptional consequences manifest, and show similar effects as we observed with the stage-restricted auxin-inducible degradation.

2. Independence from maternal TFs: The study still cannot rule out that maternal pioneer factors re-establish H3K4me3. The claim of transcription-independent epigenetic memory remains suggestive.

While classical studies in *Xenopus* have demonstrated that maternal transcription factor binding in early cleavage stages is majorly outcompeted by transcriptional repressors including abundant free-floating histones, until the increasing nuclear-to-cytoplasmic ratio hits a threshold for ZGA initiation (PMID: 6183003, 25713373), we would like to clarify that we do not claim H3K4me3 establishment to be completely independent of maternal pioneer factors and agree that it is an interesting mechanistic possibility. Indeed, as noted in the manuscript (line #629), we cannot exclude their potential contribution. Pioneer transcription factor binding prior to ZGA is well established across species as a mechanism to establish chromatin competence for future transcription, for example by opening regulatory elements and rendering them accessible. Consistent with this, in our study we observe pre-ZGA accessible promoters that lack H3K4me3 (Fig.1g). However, our focus here is on transcriptional independence, demonstrating the presence and function of promoter H3K4me3 prior to major ZGA. While maternal TF involvement is an important question, addressing it in detail is outside the scope of this study. We further appreciate that there is at this moment no uniform definition of epigenetic memory across the field (i.e. does it have to be independent of pioneer TFs or not to be called epigenetic?) and thus we will remove “epigenetic memory” and “transcriptional memory” from the title, abstract and results section.

3. Chromatin-bound validation: The authors’ extraction protocol may enrich for chromatin-bound histones, but no direct biochemical validation is provided to exclude contamination by free histones. This is critical in early *Xenopus* embryos, where free histone pools are abundant.

We have provided a Western blot to demonstrate the effectiveness of chromatin fractionation. The strong enrichment of histone H3 in the chromatin-bound protein isolate, combined with the depletion of cytoplasmic tubulin and the nuclear pore complex (NPC), indicates that our protocol successfully isolates chromatin-bound material while minimizing contamination from non-chromatin-associated nuclear proteins. This provides biochemical validation that the H3 detected in our pre-ZGA chromatin samples predominantly represents nucleosome-bound histones rather than free histone pools.

4. Quantitative ChIP-seq comparability: The numbers of peaks pre- vs. post-ZGA are not normalized with spike-ins, so apparent “maintenance” could reflect global differences in ChIP efficiency or background rather than true retention.

We have not compared the signal across timepoints and have limited our conclusions to observations made strictly from peak calling that has been performed against the respective background of the sample in question.

5. Definition of transcriptional silence: For loci designated “silent,” there is no Pol II occupancy analysis to exclude paused or engaged polymerase; α -Amanitin assays at mid-ZGA do not address the earlier cleavage stages.

We emphasize here that transcriptional silence in pre-ZGA *Xenopus* embryos is well established by multiple independent approaches, including direct nascent RNA labeling (PMID: 31211992), classical studies showing that ZGA timing is controlled by a sizer mechanism whereby the increasing nuclear-to-cytoplasmic ratio titrates out repressors, including free-floating histones (PMID: 6183003, 25713373), and work demonstrating that inhibitory chromatin prevents transcription factor binding, in particular TBP, until this developmental threshold is reached (PMID: 10567523). Together, these findings provide strong evidence that the pre-ZGA genome is globally silent, rather than merely paused.

In line with this, we have shown direct labelling of nascent transcripts in pre-ZGA embryos followed by immunofluorescence in Fig.S1d, which revealed no detectable signal. Consistently, the transcription-associated histone marks H3K36me3 and H3K79me3 were absent from pre-ZGA embryo chromatin as assessed by mass spectrometry and Western blots (Fig.1b, 1d). The absence of these elongation-associated marks strongly supports widespread transcriptional quiescence. While Pol II ChIP-seq has not been successfully performed on pre-ZGA embryos in *X. laevis* embryos till date, *X. tropicalis* Pol II ChIP has shown minimal Pol II binding to DNA in pre-ZGA stages (PMID: 31537794). As noted, Pol II can occupy chromatin in a paused or engaged state in the absence of active transcription (PMID: 31537794, PMID: 17632051). Taken together, our data provide a more direct measure of globally silent chromatin rather than Pol II occupancy alone.

Regarding the reviewer’s point on amanitin, we note that embryos are injected at the 1-cell stage, providing several hours for amanitin to inhibit the RPB1 subunit and trigger its degradation before transcription is assayed (PMID: 11904382, 8760875), making residual Pol II binding under these conditions highly unlikely. We also previously addressed the issue of loci designated “silent” and adjusted the manuscript language accordingly (Line #168).

We apologize for making these points not clear enough in our manuscript. We have included further explanation and description of previously published data clearly showing the global transcriptional silence during pre-ZGA development in the frog *Xenopus* (Lines #125-128).

6. Specificity of Kmt2b knockdown: Unlike Cxxc1, there is no rescue experiment or second MO for Kmt2b, leaving potential off-target effects unresolved.

We acknowledge that this is an important experiment that would add further confidence to our Kmt2b knockdown results. Unfortunately, we could not provide this due to technical limitations. However, we see that the effects of Kmt2b depletion on transcription are not only in line with the known role of Kmt2b as a major H3K4 methyltransferase, but are also comparable to the effect of Cxxc1 depletion on transcription (which has overlapping functions), and different from the Ctrl MO condition, supporting a target-specific effect rather than general MO toxicity.

While we show a range of supporting data, we recognize that the missing definitive rescue control limits our conclusions, and we propose to tone down the involvement of Kmt2b by removing it from the abstract and temper our language regarding the importance of this enzyme.

These points are relevant to the paper’s main mechanistic claims. **While the data support the presence of H3K4me3 prior to ZGA and suggest it may influence genome activation, the mechanistic link to epigenetic “memory” remains insufficiently established.**

We appreciate the reviewer's acknowledgement that our data support the presence of H3K4me3 prior to ZGA and suggest its potential role in influencing genome activation. We agree that this represents a novel and important contribution of our study.

Regarding the concern that the mechanistic link to epigenetic "memory" is insufficiently established, we appreciate that there is at this moment no uniform definition of epigenetic memory across the field and thus we will remove "epigenetic memory" and "transcriptional memory" from the title, abstract and results section and discuss the implications of our results in the Discussion section of the manuscript.